# Transport Coefficients of Relativistic Matter: A Detailed Formalism with a Gross Knowledge of Their Magnitude

Ashutosh Dwibedi [1],*, Nandita Padhan [2], Arghya Chatterjee [2] and Sabyasachi Ghosh [1]

1   Indian Institute of Technology Bhilai, Kutelabhata, Khapri, Durg 491001, India; sabya@iitbhilai.ac.in
2   National Institute of Technology Durgapur, Mahatma Gandhi Avenue, Durgapur 713209, India; np.23ph1105@phd.nitdgp.ac.in (N.P.); achatterjee.phy@nitdgp.ac.in (A.C.)
*   Correspondence: ashutoshd@iitbhilai.ac.in

**Abstract:** The present review article has attempted a compact formalism description of transport coefficient calculations for relativistic fluid, which is expected in heavy ion collision experiments. Here, we first address the macroscopic description of relativistic fluid dynamics and then its microscopic description based on the kinetic theory framework. We also address different relaxation time approximation-based models in Boltzmann transport equations, which make a sandwich between Macro and Micro frameworks of relativistic fluid dynamics and finally provide different microscopic expressions of transport coefficients like the fluid's shear viscosity and bulk viscosity. In the numeric part of this review article, we put stress on the two gross components of transport coefficient expressions: relaxation time and thermodynamic phase-space part. Then, we try to tune the relaxation time component to cover earlier theoretical estimations and experimental data-driven estimations for RHIC and LHC matter. By this way of numerical understanding, we provide the final comments on the values of transport coefficients and relaxation time in the context of the (nearly) perfect fluid nature of the RHIC or LHC matter.

**Keywords:** heavy ion collisions; relativistic fluid; Boltzmann transport equation; transport coefficients; relaxation time approximation; Chapman–Enskog approximation

## 1. Introduction

The presence of both asymptotic freedom and confinement properties in the theory of Quantum Chromodynamics (QCD) makes two distinct phases of the nuclear matter possible: the quark-gluon plasma (QGP) phase in high energy and the hadron gas phase in low energy [1–3]. The heavy ion collision (HIC) facilities available in the Relativistic Heavy-Ion Collider (RHIC) located at Brookhaven National Laboratory (BNL), USA and the Large Hadron Collider (LHC) at the European Organization for Nuclear Research (CERN), Geneva accelerate the ion beams (containing many nuclei) to a relativistic speed and let them collide. The collision of each nuclei pair is referred to as an event. Each event could be either head-on (or central), i.e., the two positively charged nuclei collide with the center of them lying on a common line, or it could be peripheral (or off-central), i.e., the two positively charged nuclei center does not lie on a common line. In such an event, a medium is formed that lasts for $\sim10^{-23}$ s ($\sim1$ fm), and the temperature of the system is around $\sim10^{12}$ K ($\sim100$ MeV). At such a high temperature, on account of the asymptotic freedom of QCD, the bound (confined) quarks in the initial nucleons of the heavy ions are expected to become free, and the system is expected to be characterized by deconfined quarks, which interact with themselves with the exchange of gluons. It is widely believed that the medium formed is QGP. Gradually, this medium undergoes an expansion, resulting in a decrease in the system's temperature. In this process of cooling, when a quark–hadron phase transition temperature is reached (also known as the hadronization temperature), the free quarks start to form hadrons, and we have a state of hadronic matter where the

quarks are confined within their respective hadrons. This state of hadronic matter can be modeled by the hadron resonance gas (HRG) phase as it can have all hadrons like pion, kaon, nucleon, and many more hadron resonances. Finally, the HRG expands again, and the individual hadrons become free particles and travel toward the detector; this stage is called kinetic freeze-out. The particle detectors measure the energy and momentum of the final state particles emitted from the kinetic freeze-out hypersurface. The experimental determination of the thermodynamic and out-of-equilibrium transport properties of the nuclear matter formed in the early stage of HIC is challenging since the medium is very short-lived and, therefore, not directly observable. To extract the thermodynamic and transport properties of this matter from the particle detectors, which measure only the energy and momentum of the final state particles, one needs to model the whole-time evolution of the matter formed, starting from the initial stage of the collision of two nuclei to the kinetic freeze-out hypersurface. Relativistic viscous hydrodynamics serve as a useful tool to model the evolution of this medium [4–6].

This review is planned to introduce the basic concepts of relativistic fluid dynamics and the calculation of transport coefficients of relativistic matter to the beginning-level researchers of HIC. We have introduced the basic concepts and added useful references in the article. The more sophisticated concepts and the recent developments in the fields can be obtained in the references cited. Before discussing relativistic fluid dynamics, which is essentially a many-body phenomenon, we will first recall some of the concepts associated with many-body systems, and then we will briefly address the historical development of both non-relativistic and relativistic fluid dynamics. To describe a many-body system, one needs to keep track of numerous degrees of freedom, which becomes increasingly difficult and complicated as the number of particles increases. For example, to completely specify the state of a classical gas inside a balloon, one needs to determine the time evolution of $\sim 10^{23}$ variables, which is a practically impossible job. The principles of equilibrium statistical mechanics can be applied to such many-body systems in thermodynamic equilibrium, where the quantities of interest are certain macroscopic variables fixing the corresponding macro-state of the system. Similarly, large-time ($\sim$macroscopic time) and long-distance ($\sim$macroscopic length) descriptions of a non-equilibrium many-body system can be effectively made by averaging the relevant degrees of freedom over the macroscopic length and time scales. The theory describing many-body systems over such macroscopic length and time scales may be called an effective field theory. In this sense, fluid dynamics is a classical effective field theory that describes systems that are not very far from equilibrium [7].

The non-relativistic theory of fluid dynamics started back in the days of Leonhard Euler, who gave the equation of ideal fluid dynamics around the mid-18th century. After this monumental work carried out by Euler, the dissipative effects were included progressively, mainly by Claude-Louis Navier and George Gabriel Stokes, and theory was put on a firm footing around the middle of the mid-19th century. The non-relativistic dissipative theory of fluid dynamics, which goes by the name of Navier and Stokes, has a broad range of applicability, from aerodynamics [8] to the theory of living matter [9]. The equation of motion for non-relativistic fluid mechanics (the non-relativistic Navier–Stokes equation) and its validity is well established and forms a textbook material [10]. Nevertheless, when the macroscopic fluid velocity or the microscopic velocity of particles comprising the system becomes comparable with the speed of light, one needs a relativistic theory of fluid dynamics [10]. In contrast to the non-relativistic theory of fluid dynamics, the theory of relativistic fluid dynamics and its regime of validity is still under active research [11–17]. Historically, the standard relativistic dissipative theory of fluid dynamics was developed by Eckart [18]. In the textbook of Landau and Lifshitz [10], a chapter is dedicated to the formulation of relativistic fluid dynamics where a different definition of fluid 4-velocity was taken as that of Eckart [18]. Both of these formulations arrive at equations that are known as relativistic Navier–Stokes equations (NS equations). Similar to the case of its non-relativistic counterparts, the equations of motion for relativistic NS theory are not closed by themselves. One uses linear constitutive relations between thermodynamic gradients and dissipative

fluxes/flows to close the equation of motion. From a phenomenological standpoint, one may regard the thermodynamic and fluid field gradients as the thermodynamic forces that drive dissipative flows in the system, like shear flow, heat flow, and diffusive flows [19]. The proportionality constants in these linear constitutive relations are known as transport coefficients. The theory of relativistic fluids, where the NS equations govern the dynamics, contains first-order gradients of thermodynamic variables and fluid velocity. One can, in principle, go beyond Navier–Stokes type theories with the gradient expansion technique and introduce higher-order gradients of the fluid velocity and thermodynamic variables in the equation of motion [7,20].

Despite having a plethora of success, the non-relativistic NS equations have a well-known problem: the heat propagation speed can become unboundedly high. The same unattractive feature is also present in the relativistic NS equations, making the theory acausal by superluminal heat propagation. To fix this issue, a new theory of fluid dynamics was developed, which is known as the transient theory of fluid dynamics, where the dissipative fluxes were promoted to independent dynamical variables [21–23]. This theory goes by the name of Werner Israel and John Stewart and is also known as the Israel–Stewart (IS) theory. During the 1980s, by careful analysis, William A Hiscock and Lee Lindblom showed that the relativistic NS equations are neither casual nor stable [24–27]. The effect of hydrodynamic frame choice (see Section 2 for definition) on the causality and stability of the equations of motion was also studied during the same period [28]. The conclusion of the above analysis is that the IS equations are almost always preferable for studying the hydrodynamic behavior of any relativistic system. Recently, researchers have found a way to make the first-order fluid dynamics casual and stable. They have achieved it carefully by adding more terms to the constitutive relations of the Navier–Stokes theory. One can see the articles [16,17,29–34] to obtain comprehensive knowledge of the first-order casual hydrodynamics.

Having said this so far, one may wonder whether it is possible to derive the equation of fluid dynamics from the underlying microscopic dynamics. A starting point for the equation of microscopic dynamics can be Newton's equations of motion or the Schrodinger equation, depending on whether the system under consideration is a non-relativistic classical system or a non-relativistic quantum system. Suppose the energy scales are high (particles moving with velocities close to $c$). In that case, the starting of the equation of microscopic dynamics can be classical relativistic equations of motion or quantum field theoretical equations of motion, depending on whether the system under consideration is a relativistic classical system or a relativistic quantum system. Among all the methods, one popular method is to start from the transport equation of Boltzmann and arrive at the fluid dynamical equations with the help of certain approximation schemes. The Boltzmann equation, or the Boltzmann transport equation (BTE), was first devised by Ludwig Boltzmann in 1872, where the underlying microscopic dynamics are non-relativistic classical mechanics. This equation determines the time evolution of a single-particle distribution function, and the single-particle distribution function is defined in such a way that a suitable averaging of it leads to the determination of number density, energy density, energy current, etc., of the system. The BTE equation has two parts: the LHS of the equation corresponds to a gradual change of the particle distribution function in the presence or absence of external force, and the RHS (also known as collision kernel) represents an abrupt change in the particle distribution due to localized momentary collisions (usually, binary collisions are considered). After nearly sixty years, the collision kernel of the equation was modified by changing the collision term (RHS of the equation) with the inclusion of Bose enhancement and Pauli blocking factors to capture some of the aspects of quantum mechanics. This modified BTE goes by the name of the Uehling–Uhlenbeck equation or quantum BTE [35,36]. In the 1950s, the pioneering work on the covariant BTE both in flat and curved space–time was started by the authors of the Refs. ([37–49]), where the layout of covariant kinetic theory was provided.

Once we have the BTE at our disposal, we can derive the fluid dynamic equations and transport coefficients from it. Historically, of course, it started first for the non-relativistic BTE, and the equations of fluid dynamics and transport coefficients were derived in various approximate ways, from which two qualitatively different methods stand out: the method of Chapman–Enskog–Hilbert [20,50–52] and the method of Grad [53]. Even though the methods of Hilbert [50] and Chapman–Enskog [20] have certain similarities, they have important differences, while the truncation of the Chapman–Enskog perturbative series in the lowest and leading order leads to the Euler equation and Navier–Stokes equation, respectively; the truncation of the Hilbert perturbative series in the leading order does not lead to Navier–Stokes equation. Different than the Chapman–Enskog–Hilbert method, Grad, on the other hand, converted the BTE into an infinite tower of coupled differential equations for the moments of the distribution function, from which he derived hydrodynamic equations by truncating the infinite tower of equations in terms of the lowest order moments. All these methods of deriving transport coefficients and fluid dynamics were also pursued in the relativistic domain, where the commencing point was the relativistic BTE. The extension of the Chapman–Enskog method and moment method for the relativistic BTE and the calculation of transport coefficients in these approximations was carried out by Chernikov, Israel, Stewart, Kelly, Anderson, and Marle [43,48,54–58]. We feel that the history of relativistic Boltzmann equation based kinetic theory would remain incomplete without citing the theoretical physics group of Amsterdam where comprehensive work on many topics of kinetic theory has been carried out; their work spans from the validity of the second law of thermodynamics to the calculation of transport coefficients under arbitrary interaction of particles to the kinetic theory of gas mixtures [59–86].

The traditional methods of obtaining transport coefficients and fluid dynamic equation of motion described in the previous paragraphs were mostly implemented to the BTE with the binary $2 \leftrightarrow 2$ collision kernel. In such cases, one can notice that in the Champan–Enskog method, even in the first order, only the formal expressions of transport coefficients can be given. Usually, one obtains a power series that has to be truncated to obtain some actual estimate of them. Even to obtain the formal expression, one has to simplify the collision integral, which is an involved task [87]. The calculation carried out with any of the traditional methods becomes complex and involved because of the $2 \leftrightarrow 2$ collision kernel. Now, the following question comes to mind: is it possible to model the collision term in a way that will be simple enough for mathematical calculation but should also be able to capture the underlying physics? The answer to this question is yes. It is indeed possible to model the collision kernel with the help of the system's relaxation time scale, which is a measure of the average time between the collisions. All the models that use the average collision time of the system's particles to approximate the collision kernel go by the name of relaxation time approximation (RTA). In 1954, Bhatnagar, Gross, and Krook put forward a relaxation time model in the setting of studying the small oscillations of single-component ionized and neutral non-relativistic gases [88,89]. Since then, the relaxation time model has become popular and has also been used in the context of calculating transport coefficients of relativistic gases by Marle in 1969 [55,56], and by Anderson–Witting in 1974 [90]. The Anderson–Witting model overcomes the difficulties present in Marle's model in the extreme relativistic limit. More recently, the RTA has also been used to derive fluid dynamics and transport coefficients where the system under concern is the QGP formed in the heavy ion collision experiments [91–99]. The calculation of thermodynamic parameters and the transport coefficients of relativistic matter can be found in Refs. [100–111]. Comprehensive knowledge on shear viscosity of nucleonic matter can be found in Ref. [112]. As an aside, we should also mention that the time evolution of a multiparticle system is associated with multiple timescales. In these time hierarchies, a relaxation time determines the scale of one-particle relaxation, and a correlation time determines the scale of two-particle relaxation. These time scales for the systems formed in HIC are collision-energy-dependent and can be found out from the experimental data of $P_T$ distribution and multiplicity distribution [113].

With all of that being said, we can ask how the external forces change the system's transport properties. As we have already mentioned, the information of external forces enters the LHS of BTE and thereby modifies the transport equation. For illustration, let us consider a plasma made up of electrically charged particles; in such systems, one may not be able to ignore the electromagnetic fields produced by the electrically charged plasma constituents. The constituents of plasma are affected by their own electromagnetic fields, and one may write down the BTE with the Lorentz force term to describe such plasma. The readers can see the classic Refs. ([87,114–122]), where many properties of such relativistic plasma, starting from the entropy production to hydromagnetic waves, have been discussed in detail. In off-central HIC, the spectators moving with a relativistic speed produce electromagnetic fields [123–132], which affect the medium constituents of the QGP created [128,133–193]; apart from that, there can also be a significant magnitude of the electromagnetic field produced by the medium constituents of the QGP itself [146,194,195]. Inspired by this, the study of the properties of QGP under electromagnetic fields has gained considerable attention, and researchers working on HIC phenomenology have carried out extensive work on the subject, which spans from deriving magnetohydrodynamics equations from BTE to the calculation of the multi-component structure of transport coefficients of QGP [196–205]. A detailed review on the topic of magnetohydrodynamics can be found in Refs. ([206,207]).

In recent times, beyond the year 2000, transport coefficients like shear viscosity earned abrupt attention from the different domains of physics when people came to know about the possibility of their lower limits. It was pointed out by P. Kovtun, D. T. Son, and A. O. Starinets that the value $1/(4\pi) \approx 0.08$ may be considered as a lower bound of shear viscosity to entropy density ($\eta/s$) for a wide class of systems. This is popularly known as KSS bound, although a quantum lower bound possibility has been indicated in very early time [208]. The ratio of shear viscosity to entropy density for most of the fluids we encounter daily lies well beyond this lower bound. However, in the year 2002 at Duke University, researchers observed an extremely cold fluid made up of lithium atoms with a very small $\eta/s$ ($<0.5$) close to its lower bound. Remarkably, in 2005, experiments at the Relativistic Heavy Ion Collider (RHIC) situated at BNL produced an immensely hot fluid (QGP) with an exceptionally small value of $\eta/s$, nearly reaching its lower bound of 0.08. This remarkably fluidic behavior observed in many-body systems at extremely low and high temperatures has captivated the interest of numerous theoretical physicists, ranging across disciplines such as condensed matter physics, nuclear physics, and string theory [209]. RHIC data suggested the presence of a strongly coupled sQGP medium, challenging previous notions of a weakly coupled gas, as predicted by high-temperature QCD due to the asymptotic freedom of QCD [210]. Various microscopic estimations rested on effective QCD models [211–218], together with hadronic models [106,219–229], have been undertaken recently to comprehend the underlying dynamics responsible for the low $\eta/s$ of RHIC matter. Notably, references [213,216,220–225] have identified three potential sources—resonance-type interactions [213,216,220–223], finite-size effects [224,225], and the influence of magnetic fields—for achieving the small $\eta/s$ observed in RHIC/LHC matter. We will discuss these sources at the end of the review, where we will remark on the estimated values of transport coefficients.

The article will be organized as follows. In Section 2, we discuss the basic tenets of relativistic fluid dynamics: the macroscopic conservation laws involving particle flow and energy–momentum flow. Then, we use these conservation laws to show how one can arrive at the equation of motion (EOM) of relativistic fluid dynamics (RFD). We count the number of independent variables involved in EOM and the number of independent EOM to show that the EOM of ideal fluids can be closed with an additional equation of state, but the case of dissipative fluid needs further consideration. We conclude this section by closing the EOM of dissipative fluids with the phenomenological consideration of linear relationships between dissipative/out-of-equilibrium flows and thermodynamic forces. Section 3 is further divided into three smaller sections: Sections 3.1–3.3. In Section 3.1,

we first introduce the distribution function and the definition of conserved flows as a suitable averaging over the distribution function. Then, we write down the relativistic BTE with a short discussion on the variants of $2 \leftrightarrow 2$ collision kernel widely used in the literature. We wind up this section with the derivation of the balance or transfer equation of a macroscopic flow pertaining to the system. In Section 3.2, we define entropy flow and show that the entropy production is always positive for a distribution function obeying BTE. Then, we proceed to discuss the condition of local equilibrium and global equilibrium distribution for classical and quantum particles. Section 3.3 is dedicated to establishing thermodynamic formulas involving local equilibrium thermodynamic variables. In this section, we established Euler's identity and the first law of thermodynamics. We end this section with the derivation of useful Gibbs–Duhem equations. In Section 4, we discuss the method of solving the BTE with a general collision kernel by a perturbative technique known as the Chapman–Enskog approximation (CEA). We begin this section by specifying the conditions needed for the collision kernel to ensure the validity of conservation laws and positive entropy production. We show that the CEA can be considered an order-by-order approximation in the Knudsen number. Keeping the approximation up to the first order in the Knudsen number, the distribution function and the fluid dynamic variables are divided into the ideal or local equilibrium and the first-order out-of-equilibrium parts. We also write down the first-order BTE that forms the basis of calculating transport coefficients. We close the section by rewriting the LHS of the first-order BTE by incorporating the constraints that the ideal fluid demands. This equation serves as a master equation for deriving transport coefficients in the rest of that article. The auxiliary Section 4.1 serves as a complete application to Section 4, where we used the first-order BTE with the $2 \leftrightarrow 2$ collision kernel to derive the formal expressions of the transport coefficients in terms of the collision integrals. We see here that to specify the form of the out-of-equilibrium distribution function from the first-order BTE completely, one needs to provide five matching conditions or the condition of fit for the out-of-equilibrium distribution function. These conditions of fit essentially define the local equilibrium fluid dynamic variables. For the sake of illustration, we used the matching conditions corresponding to the Landau–Lifshitz frame to obtain the out-of-equilibrium distribution, stress–energy tensor, out-of-equilibrium flows, and transport coefficients. Section 5 is dedicated to calculating the transport coefficients in simplified relaxation time-based collisional kernels. For all the RTA discussed in Section 5, one has to ensure the conservation laws by directly putting constraints on the integral of the out-of-equilibrium distribution function. We discuss the Anderson–Witting, Marle, and BGK models in Section 5.1, Section 5.2, and Section 5.3, respectively. In Sections 5.1 and 5.2, we give the expression of the out-of-equilibrium flows and the transport coefficients in the Landau–Lifshitz frame and Eckart frame, respectively. In Section 5.3, we develop the calculation of transport coefficients and out-of-equilibrium flows in a similar fashion for the BGK collision kernel. We choose the simplest possible matching by setting the out-of-equilibrium number density and energy density to zero. In this simplest matching, the result obtained in the BGK and Anderson–Witting models is the same. In Section 6, we put stress on the fact that all RTA calculation of transport coefficients have two parts: relaxation time, which depends on the system's microscopic dynamics, and a thermodynamic phase space part, which contains the macroscopic information about the system. Then, we continue to discuss the calculation of shear viscosity to entropy density $\eta/s$ carried out in the literature for the fluid formed in the HIC with the help of perturbative QCD and different field theoretical models. The value of $\eta/s$ estimated from different models like the linear sigma model, Nambu–Jona–Lasinio, and hadronic field theory are closer to the experimental result than the corresponding results obtained from the perturbative QCD calculation. We point out that the possible source of low $\eta/s$ can be the interaction of resonance types, finite medium-size effects, and the influence of the magnetic fields. We finally compared the $\eta/s$ obtained from the Anderson–Witting model with the existing theoretical calculation. We give a range of relaxation time so that our expression can cover the existing theoretical calculation in the QGP temperature range and a range of radius for

which a hard sphere scattering model of Anderson–Witting type could cover the existing theoretical calculation in the hadronic temperature domain. In Section 7, we compared the numerical results of bulk viscosity to entropy density, electrical conductivity, and thermal conductivity obtained from various model calculations. In the end, we give a brief summary of all the sections from Sections 2–7 in Section 8.

**Notations**: For quick reference, we have added below all the notations and conventions we have used throughout the article.

- Natural units: $\hbar = k_B = c = 1$
- Minkowski Metric: $\eta^{\mu\nu} = \text{dia}(1, -1, -1, -1)$
- Partial derivative: $\frac{\partial}{\partial x^\mu} \equiv \partial_\mu$
- The symmetric spatial rank-2 projector: $\Delta^{\mu\nu} \equiv \eta^{\mu\nu} - u^\mu u^\nu$ , $(u_\mu u^\mu = 1)$
- The symmetric spatial and traceless rank-4 projector: $\Delta^{\mu\nu\alpha\beta} \equiv \frac{1}{2}(\Delta^{\mu\alpha}\Delta^{\nu\beta} + \Delta^{\mu\beta}\Delta^{\nu\alpha}) - \frac{1}{3}\Delta^{\mu\nu}\Delta^{\alpha\beta}$
- The symmetric spatial and traceless projection of an arbitrary tensor $A^{\mu\nu}$: $A^{\langle\alpha\beta\rangle} \equiv \Delta^{\alpha\beta}_{\mu\nu}A^{\mu\nu}$

## 2. Relatisvistic Fluid Dynamics

One generally assumes a continuum distribution of matter-energy when one speaks about the fluid properties of a medium. The following two quantities are of fundamental importance for a fluid. One is the stress–energy tensor $T^{\mu\nu}$, and another is particle 4-flow $N^\mu$ [1]. The stress–energy tensor ($T^{\mu\nu}$) gives information about the energy density and energy–momentum flow inside the fluid medium, and it is a 2-rank tensor. The particle 4-flow ($N^\mu$) gives information about the particle density and particle flow inside the fluid medium; it is a 4-vector. We will write $T^{\mu\nu}$ in matrix form as:

$$T^{\mu\nu} = \begin{pmatrix} T^{00} & T^{01} & T^{02} & T^{03} \\ T^{10} & T^{11} & T^{12} & T^{13} \\ T^{20} & T^{21} & T^{22} & T^{23} \\ T^{30} & T^{31} & T^{32} & T^{33} \end{pmatrix}, \tag{1}$$

where,

1. $T^{00}$ is defined as the energy density,
2. $T^{0i} = T^{i0}$ is defined as $i$th component momentum density or the energy current density in $i$th direction,
3. $T^{ij} = T^{ji}$ is the $i$th component momentum current density in $j$th the direction or $j$th component momentum current density in $i$th the direction.

In the absence of any inflow of energy to the system (absence of external forces), the system's energy–momentum remains conserved. One can write 4 equations to ensure this energy–momentum conservation:

$$\frac{\partial T^{00}}{\partial x^0} = -\left(\frac{\partial T^{10}}{\partial x^1} + \frac{\partial T^{20}}{\partial x^2} + \frac{\partial T^{30}}{\partial x^3}\right), \implies \partial_0 T^{00} = -\partial_i T^{i0}, \tag{2}$$

$$\frac{\partial T^{01}}{\partial x^0} = -\left(\frac{\partial T^{11}}{\partial x^1} + \frac{\partial T^{21}}{\partial x^2} + \frac{\partial T^{31}}{\partial x^3}\right), \implies \partial_0 T^{01} = -\partial_i T^{i1}, \tag{3}$$

$$\frac{\partial T^{02}}{\partial x^0} = -\left(\frac{\partial T^{12}}{\partial x^1} + \frac{\partial T^{22}}{\partial x^2} + \frac{\partial T^{32}}{\partial x^3}\right), \implies \partial_0 T^{02} = -\partial_i T^{i2}, \tag{4}$$

$$\frac{\partial T^{03}}{\partial x^0} = -\left(\frac{\partial T^{13}}{\partial x^1} + \frac{\partial T^{23}}{\partial x^2} + \frac{\partial T^{33}}{\partial x^3}\right), \implies \partial_0 T^{03} = -\partial_i T^{i3}. \tag{5}$$

Each expression from Equations (2) to (5) is written in the form of usual conservation equations one encounters where the LHS is the time derivative of the density of a quantity,

and the RHS is the negative of the divergence of the corresponding current. One can collectively write Equations (2) to (5) in the following compact and covariant manner:

$$\frac{\partial T^{\mu\nu}(x)}{\partial x^{\mu}} = \partial_{\mu} T^{\mu\nu}(x) = 0 \,. \tag{6}$$

Equation (6) is the conservation equation for the stress–energy tensor of a fluid. We will write $N^{\mu}$ as the following column matrix:

$$N^{\mu} = \begin{pmatrix} N^0 \\ N^1 \\ N^2 \\ N^3 \end{pmatrix} \equiv (N^0, N^i) \,, \tag{7}$$

where,

1. $N^0$ is defined as the particle density,
2. $N^i$ is defined as the particle current.

If there is no creation or annihilation of particles, the total particle number of the system remains conserved. Therefore, to guarantee particle number conservation, we have:

$$\frac{\partial N^0}{\partial x^0} = -\left( \frac{\partial N^1}{\partial x^1} + \frac{\partial N^2}{\partial x^2} + \frac{\partial N^3}{\partial x^3} \right), \implies \partial_0 N^0 = -\partial_i N^i \,. \tag{8}$$

Equation (8) can be represented covariantly as follows:

$$\frac{\partial N^{\mu}(x)}{\partial x^{\mu}} = \partial_{\mu} N^{\mu}(x) = 0 \,, \tag{9}$$

Equation (9) represents the conservation of particle 4-flow of the fluid medium.

At the heart of the fluid dynamic description of a system lies the concept of fluid 4-velocity $u^{\mu}$, which is a time-like vector with the normalization condition $u^{\mu} u_{\mu} = 1$. To give this abstract vector a physical meaning, one divides the total fluid into many little fluid elements, which are large compared to the microscopic dimension of the system (particle size) but still very small compared to the macroscopic dimension of the system (say, the dimension of the fluid container). In that case, the average flow-velocity of a little fluid element is identified with $u^{\mu}$. For an observer moving with a fluid element (co-moving observer) $u^{\mu} = (1, \vec{0})$, the corresponding Lorentz frame associated with the observer is called the fluid rest frame or local rest frame (LRF). We should emphasize that there is arbitrariness in the definition of $u^{\mu}$ since one can take the average flow as the flow direction of any physical quantity. In the textbook by Landau and Lifshitz [10], the average flow has been defined to follow energy, whereas Eckart, in his seminal paper [18], has defined it to follow the particles. We will see that, in global thermodynamic equilibrium, one can unambiguously define the $u^{\mu}$ since, in this situation, the direction of energy flow and particle flow coincide. We will discuss more about it later when we write the general decomposition of $T^{\mu\nu}$ and $N^{\mu}$. Now, we shall describe some tensors that can be formed with the help of $u^{\mu}$, $\eta^{\mu\nu}$, and $\partial_{\mu}$; it will make our life easier when we write the general tensor decomposition of $T^{\mu\nu}$, $N^{\mu}$, and the dynamical equations of fluid.

1. The symmetric spatial rank-2 projector ($\Delta^{\mu\nu}$):
   Definition: $\Delta^{\mu\nu} \equiv \eta^{\mu\nu} - u^{\mu} u^{\nu}$ .
   Properties: $u_{\mu} \Delta^{\mu\nu} = 0$, $\Delta^{\mu}{}_{\nu} = \Delta_{\nu}{}^{\mu} \equiv \Delta^{\mu}_{\nu}$, $\eta^{\mu\nu} \Delta_{\mu\nu} = \Delta^{\mu\nu} \Delta_{\mu\nu} = 3$, and $\Delta^{\mu\nu} \Delta_{\nu\sigma} = \Delta^{\mu}_{\sigma}$.
   In the LRF, the matrix corresponds to $\Delta^{\mu\nu}$ becomes completely spatial:

$$\Delta^{\mu\nu} = \begin{pmatrix} 0 & 0 & 0 & 0 \\ 0 & -1 & 0 & 0 \\ 0 & 0 & -1 & 0 \\ 0 & 0 & 0 & -1 \end{pmatrix} \,. \tag{10}$$

Any 4-vector $A^\mu$ can be uniquely decomposed into a part parallel to $u^\mu$ and a part perpendicular to $u^\mu$ as $A^\mu = (u_\nu A^\nu)u^\mu + \Delta^{\mu\nu}A_\nu$, where the first part will have only temporal components and the second part have only spatial components in LRF.

2.  The spatial gradient ($\nabla^\mu$) and temporal gradient ($D$):
    Definition: $\nabla^\mu \equiv \Delta^{\mu\nu}\partial_\nu$, $D \equiv u^\mu \partial_\mu$ .
    Properties: $u_\mu \nabla^\mu = 0$. In LRF $\nabla^\mu \to -\partial_i$ , and $D \to \partial_0$. One can decompose $\partial_\mu$ in a general frame as: $\partial_\mu = \nabla_\mu + u_\mu D$ .

3.  The symmetric spatial and traceless rank-4 projector ($\Delta^{\mu\nu\alpha\beta}$):
    Definition: $\Delta^{\mu\nu\alpha\beta} = \frac{1}{2}(\Delta^{\mu\alpha}\Delta^{\nu\beta} + \Delta^{\mu\beta}\Delta^{\nu\alpha}) - \frac{1}{3}\Delta^{\mu\nu}\Delta^{\alpha\beta}$.
    Properties: $u_\mu \Delta^{\mu\nu\alpha\beta} = u_\alpha \Delta^{\mu\nu\alpha\beta} = 0$, $\Delta^{\mu\nu\alpha\beta} = \Delta^{\nu\mu\alpha\beta} = \Delta^{\mu\nu\beta\alpha}$, $\Delta^{\mu\nu}{}_{\alpha\beta} = \Delta_{\alpha\beta}{}^{\mu\nu} \equiv \Delta^{\mu\nu}_{\alpha\beta}$, $\Delta^{\mu\nu}_{\alpha\beta}\Delta^{\alpha\beta}_{\lambda\sigma} = \Delta^{\mu\nu}_{\lambda\sigma}$, $\Delta^{\mu\nu}_{\alpha\beta}\Delta^{\alpha\beta} = 0$ .
    The projector $\Delta^{\mu\nu}_{\alpha\beta}$ projects the symmetric traceless part of a 2-rank tensor onto the direction orthogonal to $u^\mu$, i.e., $A^{\langle\alpha\beta\rangle} = \Delta^{\alpha\beta}_{\mu\nu}A^{\mu\nu}$, where we define trace of any tensor $B^{\mu\nu}$ by the contraction $\eta^{\mu\nu}B_{\mu\nu}$ .

Now, if one looks into the literature [7,19,231,232], one can find the theory of ideal fluids where one takes $T^{\mu\nu}$ and $N^\mu$ as follows:

$$T^{\mu\nu} = \mathcal{E}_0 u^\mu u^\nu - P_0 \Delta^{\mu\nu} , \tag{11}$$

$$N^\mu = n_0 u^\mu , \tag{12}$$

where we suppressed the space–time (x) dependent of the variables $\mathcal{E}_0$, $u^\mu$, $P_0$, and $n_0$. For the sake of simplicity, we will first discuss the parameters contained in Equations (11) and (12) and the ideal fluid dynamical equation of motion (EOM) implied by them. In LRF, i.e., $u^\mu \to (1,\vec{0})$ , the $T^{\mu\nu}$ and $N^\mu$ given in Equations (11) and (12) become:

$$T^{\mu\nu} = \begin{pmatrix} \mathcal{E}_0 & 0 & 0 & 0 \\ 0 & P_0 & 0 & 0 \\ 0 & 0 & P_0 & 0 \\ 0 & 0 & 0 & P_0 \end{pmatrix}, \ N^\mu = \begin{pmatrix} n_0 \\ 0 \\ 0 \\ 0 \end{pmatrix} = (n_0, 0) . \tag{13}$$

Comparing Equation (13) with the definitions of tensor components given in Equations (1) and (7), we obtain $n_0$ to be the particle density in the LRF and $\mathcal{E}_0$ to be the energy density in the LRF (internal energy). Usually, the spatial diagonal part of the $T^{\mu\nu}$ in LRF, i.e., $P_0$, is identified with pressure. The EOM for the ideal fluid is obtained by substituting $T^{\mu\nu}$ and $N^\mu$ from Equations (11) and (12) in Equations (6) and (9) :

$$(\mathcal{E}_0 + P_0)Du^\mu = \nabla^\mu P_0 , \tag{14}$$

$$D\mathcal{E}_0 = -(\mathcal{E}_0 + P_0)\nabla_\mu u^\mu , \tag{15}$$

$$Dn_0 = -n_0 \nabla_\mu u^\mu , \tag{16}$$

where in writing Equations (14) and (15), we have to project Equation (6) in the direction perpendicular and parallel to $u^\mu$. Since the contraction of $u_\mu$ with Equation (14) vanishes, it has the content of three independent equations, so along with Equations (15) and (16), we have a total of 5 independent equations in terms of 6 variables $u^\mu$ ($u^\mu u_\mu = 1$), $\mathcal{E}_0$, $P_0$, and $n_0$ . This system of equations can only be closed if one defines a relation like $\mathcal{E}_0 = \mathcal{E}_0(P_0, n_0)$, known as the equation of state. Here, we have already identified LRF energy density $\mathcal{E}_0$ with the internal energy and $P_0$ with pressure but have not discussed thermodynamics so far. Usually, in an undergraduate textbook [233], one applies thermodynamics to a system without macroscopic flows, and in such situations, $P_0$, $n_0$, and $\mathcal{E}_0$ remain constant (not space–time dependent) throughout the system. For a system like fluid, one may speak about local thermal equilibrium, where one assumes that thermalization has occurred for each little fluid element. Therefore, one may associate thermodynamic quantities with each

fluid cell and apply laws of thermodynamics locally. We will discuss this version of local thermodynamics in detail in Section 3.3.

One can find the theory of dissipative fluids in literature [7,19,231,232], where one defines $T^{\mu\nu}$ and $N^\mu$ as:

$$T^{\mu\nu} = \mathcal{E}u^\mu u^\nu - (P^{(0)} + \Pi)\Delta^{\mu\nu} + h^\mu u^\nu + h^\nu u^\mu + \pi^{\mu\nu}\,, \tag{17}$$

$$N^\mu = nu^\mu + \nu^\mu\,, \tag{18}$$

where $u_\mu h^\mu = u_\mu \pi^{\mu\nu} = u_\mu \nu^\mu = 0$. For brevity, we suppressed the space–time (x) dependent of the variables $\mathcal{E}$, $u^\mu$, $P^{(0)}$, $\Pi$, $n$, $h^\mu$, $\pi^{\mu\nu}$, and $\nu^\mu$ in Equations (17) and (18). For the case of dissipative fluid, the most general tensor decomposition needs to be performed by taking $T^{\mu\nu}$ to be a symmetric 2-rank tensor and $N^\mu$ to be a 1-rank tensor. This decomposition has been employed in writing Equations (17) and (18). It is easy to see by using $u_\mu h^\mu = u_\mu \pi^{\mu\nu} = \Delta_{\mu\nu}\pi^{\mu\nu} = u_\mu \nu^\mu = 0$ that

$$\mathcal{E} = u_\mu u_\nu T^{\mu\nu}, \ P^{(0)} + \Pi = -\tfrac{1}{3}\Delta^{\mu\nu}T_{\mu\nu}, \ h^\mu = \Delta^\mu_\sigma T^{\sigma\nu}u_\nu, \ \pi^{\mu\nu} = \Delta^{\mu\nu}_{\alpha\beta}T^{\alpha\beta},$$
$$n = N^\mu u_\mu, \text{ and } \nu^\mu = \Delta^\mu_\sigma N^\sigma\,. \tag{19}$$

In this decomposition, $\mathcal{E}$, $n$, and $P^{(0)}$ are identified with internal energy, number density, and equilibrium pressure in the LRF of the fluid. The quantities $\Pi$, $h^\mu$, $\pi^{\mu\nu}$, and $\nu^\mu$ characterize the presence of dissipative effects, and they vanish in equilibrium. The scalar $\Pi$ is the out-of-equilibrium pressure of the fluid; it is called the bulk scalar. $P \equiv P^{(0)} + \Pi$ may be called the total pressure of the fluid. In a similar spirit one can also break $n$ and $\mathcal{E}$ as $n = n^{(0)} + \delta n$ and $\mathcal{E} = \mathcal{E}^{(0)} + \delta\mathcal{E}$, where $\delta n$ and $\delta\mathcal{E}$ correspond to out-of-equilibrium correction to local equilibrium number density and energy density. The 4-flow $h^\mu$ and $\nu^\mu$ are out-of-equilibrium energy flow (energy diffusion) and particle flow (particle diffusion), respectively. And the tensor $\pi^{\mu\nu}$ is known as the shear stress tensor of the fluid. One can define heat-flow 4-vector $q^\mu$ by the help of $h^\mu$ and $\nu^\mu$ as follows: $q^\mu \equiv h^\mu - \frac{\mathcal{E}+P}{n}\nu^\mu$. The EOM for the dissipative fluid is obtained by substituting $T^{\mu\nu}$ and $N^\mu$ from Equations (17) and (18) in Equations (6) and (9) :

$$(\mathcal{E} + P^{(0)} + \Pi)Du^\mu - \nabla^\mu(P^{(0)} + \Pi) + h^\mu(\nabla_\nu u^\nu) + h^\alpha \Delta^{\mu\nu}\nabla_\alpha u_\nu$$
$$+ \Delta^{\mu\nu}Dh_\nu + \Delta^{\mu\nu}\partial_\alpha \pi^\alpha_\nu = 0\,, \tag{20}$$

$$D\mathcal{E} + (\mathcal{E} + P^{(0)} + \Pi)\nabla_\mu u^\mu + \partial_\mu h^\mu + u_\nu Dh^\nu - \pi^{\alpha\beta}\sigma_{\alpha\beta} = 0\,, \tag{21}$$

$$Dn + n\nabla_\mu u^\mu + \partial_\mu \nu^\mu = 0\,, \tag{22}$$

where $\sigma_{\alpha\beta} \equiv \Delta^{\mu\nu}_{\alpha\beta}\partial_\mu u_\nu$. In obtaining Equations (20) and (21), we have to project Equation (6) in the direction perpendicular and parallel to $u^\mu$. We also use the decomposition $\partial_\mu = \nabla_\mu + u_\mu D$.

If we count the number of independent variables contained in $T^{\mu\nu}$ and $N^\mu$, they are a total of fourteen: ten come from $T^{\mu\nu}$ owing to its symmetric nature, and four come from $N^\mu$. Now, when we decompose the tensors $T^{\mu\nu}$ and $N^\mu$ with the help of a time like unit vector $u^\mu$ with $u_\mu u^\mu = 1$ in Equations (17) and (18) with the conditions $u_\mu h^\mu = u_\mu \pi^{\mu\nu} = u_\mu \nu^\mu = 0$, we have a total of 17 variables: 3 come from $h^\mu$; 3 come from $\nu^\mu$; 5 come from $\pi^{\mu\nu}$; and 3 come from $n(= n^{(0)} + \delta n)$, $\mathcal{E}(= \mathcal{E}^{(0)} + \delta\mathcal{E})$ and $P(= P^{(0)} + \Pi)$. The extra 3 variables come from the arbitrariness of $u^\mu$. The arbitrariness of $u^\mu$ can be fixed by the choice of hydrodynamic frames [32]. Two choices are very popular in the literature, i.e.,

- The Landau–Lifshitz frame (LF): $T^{\mu\nu}u_\nu = (T^{\alpha\beta}u_\alpha u_\beta)u^\mu$, $\implies T^{\mu\nu}u_\nu = \mathcal{E}u^\mu$ .
- The Eckart frame (EF): $N^\mu = (\sqrt{N^\nu N_\nu})u^\mu$, $\implies N^\mu = nu^\mu$ .

The LF condition defines $u^\mu$ to be an eigen vector of $T^{\mu\nu}$; this hydrodynamic frame choice makes $h^\mu = \Delta^\mu_\lambda T^{\lambda\nu}u_\nu = 0$, and the heat flow becomes $q^\mu = -\frac{\mathcal{E}+P}{n}\nu^\mu$. On the other hand, the EF condition makes $\nu^\mu = 0$; as a result, the heat flow and energy diffusion

become equal, i.e., $h^\mu = q^\mu$. We saw that the number of independent variables after choosing the LF or EF frame, which is 14, is still more than the number of independent equations, which is 5. Therefore, to solve the EOM for the fluid, it is necessary to decrease the number of variables from 14 to 5 . Let us see how this reduction in variables from 14 to 5 can be carried out by taking a phenomenological viewpoint. We have written the thermodynamic variables: $\mathcal{E}$, $P$, and $n$, as local equilibrium part + out-of-equilibrium correction, one can postulate that the equilibrium part of these variables obeys the laws of thermodynamics, i.e., Eulers' identity: $s^{(0)}T = \mathcal{E}^{(0)} + P^{(0)} - \mu\, n^{(0)}$, and the first law of thermodynamics: $ds^{(0)} = \frac{1}{T}(d\mathcal{E}^{(0)} - \mu\, dn^{(0)})$. These two relations guarantee that only two of the five local equilibrium thermodynamic variables—$\mathcal{E}^{(0)}$, $n^{(0)}$, $P^{(0)}$, $\mu$, and $T$— are independent. One usually considers these two independent variables as $\mu$ and $T$. The out-of-equilibrium corrections—$\delta\mathcal{E}$, $\delta n$, and $\delta P$ for the variables $\mathcal{E}$, $n$, and $P$—can, in general, be written to be a linear combination of the gradients of local-equilibrium variables $\partial_\mu u^\mu$, $u^\mu \partial_\mu T$, and $u^\mu \partial_\mu \left(\frac{\mu}{T}\right)$ [17]. As a result of this procedure, we find that there are only 5 independent local equilibrium variables, namely, $\mu$, $T$, and $u^\mu$. Now, let us see how to reduce the 9 independent variables present in $\pi^{\mu\nu}$, $\Pi$, and $\nu^\mu$ in LF or $\pi^{\mu\nu}$, $\Pi$, and $h^\mu$ in EF. This can be carried out by adopting a phenomenological viewpoint as carried out in Ref. ([234]) and defining the entropy 4-current $S^\mu$ for the fluid. Demanding that the entropy production $\partial_\mu S^\mu$ is positive at any space–time location, one obtains linear relationships between the thermodynamic forces $\sigma^{\mu\nu}$, $\nabla^\mu u_\mu$ and, $\nabla^\mu \left(\frac{\mu}{T}\right)$ and the dissipative flows $\pi^{\mu\nu}$, $\Pi$, and $\nu^\mu$ respectively. The proportionality constant between them is called the transport coefficient. In the LF, we have the following linear relations:

$$
\begin{aligned}
\pi^{\mu\nu} &= 2\eta\,\sigma^{\mu\nu}\,, \\
\Pi &= -\zeta\,\nabla_\mu u^\mu\,, \\
\nu^\mu &= \kappa\,\nabla^\mu\left(\frac{\mu}{T}\right),
\end{aligned}
\tag{23}
$$

where $\eta$, $\zeta$, and, $\kappa$ refer to shear viscosity, bulk viscosity, and particle diffusion. In the LF $q^\mu = -\frac{\mathcal{E}+P}{n}\nu^\mu = -\frac{\mathcal{E}+P}{n}\kappa\,\nabla^\mu\left(\frac{\mu}{T}\right) = -\lambda\nabla^\mu\left(\frac{\mu}{T}\right)$, where the heat diffusion coefficient $\lambda \equiv \frac{\mathcal{E}+P}{n}\kappa$ . Similarly, in EF, one obtains the following linear relationships between thermodynamic forces and dissipative flows:

$$
\begin{aligned}
\pi^{\mu\nu} &= 2\eta\,\sigma^{\mu\nu}\,, \\
\Pi &= -\zeta\,\nabla_\mu u^\mu\,, \\
h^\mu = q^\mu &= -\lambda\,\nabla^\mu\left(\frac{\mu}{T}\right),
\end{aligned}
\tag{24}
$$

where $\eta$, $\zeta$, and, $\lambda$ refer to shear viscosity, bulk viscosity, and energy diffusion. Now, one can see with the help of the expressions given in Equation (23) or Equation (24) that we have successfully reduced the number of independent variables from 14 to 5. This reduction has been carried out by expressing all the dissipative flows and the out-of-equilibrium correction to the local thermodynamic variables in terms of gradients of five independent local thermodynamic variables: $u^\mu$, $\mu$, and $T$. In the process of this reduction, we took the help of Euler's thermodynamic identity, the first law of thermodynamics, and the linear connections between the dissipative flows and thermodynamic forces. We will prove the first two facts in Section 3.3 and the last fact in Section 4.1.

## 3. Kinetic Theory

### 3.1. Boltzmann Equation and Balance Equation

In the description of a system by Boltzmann equation-based kinetic theory, one takes the help of a single-particle distribution function $f(x^\alpha, p^\alpha) \equiv f(x, p)$ to obtain the average macroscopic densities and flows of the system. The distribution function encodes all the microscopic information of the system, and a suitably defined average of $f$ gives macroscopic information about the system. The distribution function $f$ is defined in such a

way that the $f$ multiplied by an elementary phase volume gives the number of particles in that volume, i.e., $f(x,p)\, d^3x d^3p =$ the number of particles in the volume element $d^3x$ about $x^\mu$ within a momentum range $d^3p$ about $p^\mu$. The number density of particles and the particle current density can be defined with the help of $f$ as follows:

$$\text{number density} = N^0 = \int \frac{d^3p}{(2\pi)^3}\, f(x,p)\,, \tag{25}$$

$$\text{particle current density} = N^i = \int \frac{d^3p}{(2\pi)^3}\, v^i f(x,p)\,, \tag{26}$$

where $v^i$ in Equation (26) stands for particle velocity. We can rewrite Equations (25) and (26) collectively in a covariant form as:

$$N^\mu = \int \frac{d^3p}{(2\pi)^3\, p_0}\, p^\mu f(x,p)\,, \tag{27}$$

where $v^i = \frac{p^i}{E}$ and $E = p_0 =$ particle energy. One also defines the energy density, energy current density, and momentum current density with the help of $f$ as follows:

$$\text{energy density} = T^{00} = \int \frac{d^3p}{(2\pi)^3}\, E\, f(x,p)\,, \tag{28}$$

$$\text{energy current density} = T^{0i} = T^{i0} = \int \frac{d^3p}{(2\pi)^3}\, E v^i\, f(x,p)\,, \tag{29}$$

$$\text{momentum current density} = T^{ij} = \int \frac{d^3p}{(2\pi)^3}\, p^i v^j\, f(x,p)\,. \tag{30}$$

We can rewrite Equations (28)–(30) collectively in a covariant form as:

$$T^{\mu\nu} = \int \frac{d^3p}{(2\pi)^3\, p_0}\, p^\mu p^\nu f(x,p)\,. \tag{31}$$

From Equations (27) and (31), it can be seen that $N^\mu$ and $T^{\mu\nu}$ transform as a 4-vector and 2-rank tensor, respectively, since $\frac{d^3p}{(2\pi)^3\, p_0} = \frac{d^3p'}{(2\pi)^3\, p'_0}$ is a Lorentz-invariant measure and $f(x,p)$ is a Lorentz scalar. The fluid dynamical quantities we defined in Section 2 by decomposing $T^{\mu\nu}$ and $N^\mu$ can be written as appropriate integrals over the distribution function:

$$\mathcal{E} = u_\mu u_\nu T^{\mu\nu} = u_\mu u_\nu \int \frac{d^3p}{(2\pi)^3\, p_0}\, p^\mu p^\nu\, f\,, \tag{32}$$

$$P = -\frac{1}{3}\Delta_{\mu\nu} T^{\mu\nu} = -\frac{1}{3}\Delta_{\mu\nu} \int \frac{d^3p}{(2\pi)^3\, p_0}\, p^\mu p^\nu\, f\,, \tag{33}$$

$$h^\mu = \Delta^\mu_\sigma\, u_\nu T^{\sigma\nu} = \Delta^\mu_\sigma\, u_\nu \int \frac{d^3p}{(2\pi)^3\, p_0}\, p^\sigma p^\nu\, f\,, \tag{34}$$

$$\pi^{\mu\nu} = \Delta^{\mu\nu}_{\alpha\beta} T^{\alpha\beta} = \Delta^{\mu\nu}_{\alpha\beta} \int \frac{d^3p}{(2\pi)^3\, p_0}\, p^\alpha p^\beta\, f\,, \tag{35}$$

$$n = u_\mu N^\mu = u_\mu \int \frac{d^3p}{(2\pi)^3\, p_0}\, p^\mu\, f\,, \tag{36}$$

$$\nu^\mu = \Delta^\mu_\nu N^\nu = \Delta^\mu_\nu \int \frac{d^3p}{(2\pi)^3\, p_0}\, p^\nu\, f\,. \tag{37}$$

To completely understand the system's dynamics, one needs to determine $f$ at all space–time points. This can be accomplished by solving the underlying equation obeyed by $f$. The relativistic Boltzmann Transport Equation (BTE) gives the time evolution of the single-particle phase space distribution function $f(x^\alpha, p^\alpha)$. Many good textbooks have

given its derivation in non-relativistic [235] and relativistic setups [236,237]. Here, we will write down its form and briefly discuss the properties of it. The relativistic BTE is given by,

$$p^\alpha \frac{\partial f}{\partial x^\alpha} + m \frac{\partial (f F^\alpha)}{\partial p^\alpha} = C[f] \,, \tag{38}$$

where $m$ is the rest mass of particles, and $F^\alpha$ is the 4-force on the particles given by $F^\alpha = \frac{dp^\alpha}{d\tau}$. The LHS of Equation (38) gives the streaming part of the particle distribution in the presence of the 4-force $F^\alpha$, and the RHS $C[f]$ gives the abrupt change in the distribution function $f$ due to random incessant collisions of particles with each other. In usual dilute gas approximation, one takes $2 \leftrightarrow 2$ collisions between the gas particles, neglecting other multi-particle collisions. In this simple case, $C[f]$ becomes:

$$C[f] = \int \frac{d^3 p_*}{(2\pi)^3 \, p_*^0} \frac{d^3 p'_*}{(2\pi)^3 \, p'^0_*} \frac{d^3 p'}{(2\pi)^3 \, p'^0} \, W \, (f' f'_* - f f_*) \,, \tag{39}$$

where $W$ is the transition rate. The 1st term in Equation (39) gives information about the particle of momentum $p^\alpha$ gained because of the collisions of type $(p', p'_*) \rightarrow (p, p_*)$, and the second term in Equation (39) gives information about particle of momentum $p^\alpha$ lost because of the collisions of type $(p, p_*) \rightarrow (p', p'_*)$. The collision kernel $C[f]$ can be modified to encompass the quantum nature of the particles by incorporating the Bose enhancement factor for bosons and the Pauli blocking factor for fermions in the $C[f]$ as:

$$C[f] = \int \frac{d^3 p_*}{(2\pi)^3 \, p_*^0} \frac{d^3 p'_*}{(2\pi)^3 \, p'^0_*} \frac{d^3 p'}{(2\pi)^3 \, p'^0} \, W \, [f' f'_* (1 + af)(1 + af_*) - f f_* (1 + af')(1 + af'_*)] \,, \tag{40}$$

where we defined $f(x, p_*) \equiv f_*$, $f(x, p') \equiv f'$, $f(x, p'_*) \equiv f'_*$. $a = 1$ for bosons, and $a = -1$ for fermions. The BTE with the $C[f]$ given in Equation (40) is often called the Relativistic Uheling-Uhlenbeck Equation(UUE) [36,205]. The BTE with $C[f]$ given in Equation (39) or Equation (40) is notoriously complicated to solve due to the non-linear nature of its collision term, which makes it a non-linear partial integro-differential equation. Therefore, the determination of fluid dynamical variables from the expressions given in Equations (32)–(37) is a difficult task. In this section, we will derive the conservation equations discussed in Section 2 with the help of BTE. The discussion about the determination of fluid dynamic variables will be presented in Section 4 after discussing entropy production and local equilibrium thermodynamics in Sections 3.2 and 3.3. If we have any function $\psi(x^\alpha, p^\alpha) \equiv \psi(x, p)$, we may define the macroscopic flow corresponding to it as:

$$\int \frac{d^3 p}{(2\pi)^3 \, p_0} \, p^\mu \, \psi(x, p) f(x, p) \,. \tag{41}$$

It can be seen that with $\psi = 1$, and $p^\nu$, we obtain the particle flow $N^\mu$ and energy–momentum flow or stress–energy tensor $T^{\mu\nu}$, respectively. We can derive the balance/transfer equation (space–time evolution) of any general flow defined in Equation (41) by multiplying the function $\psi(x, p)$ in Equation (38) and integrating with respect to the measure $\frac{d^3 p}{(2\pi)^3 \, p_0}$ as follows:

$$\int \frac{d^3 p}{(2\pi)^3 \, p_0} \, \psi(x, p) \left[ p^\mu \frac{\partial f}{\partial x^\mu} + m \frac{\partial F^\mu f}{\partial p^\mu} \right] = I[\psi] \,,$$

$$\text{where} \quad I[\psi] = \int \frac{d^3 p}{(2\pi)^3 \, p_0} \frac{d^3 p_*}{(2\pi)^3 \, p_*^0} \frac{d^3 p'_*}{(2\pi)^3 \, p'^0_*} \frac{d^3 p'}{(2\pi)^3 \, p'^0} \, \psi(x, p)$$

$$W \, [f' f'_* (1 + af)(1 + af_*) - f f_* (1 + af')(1 + af'_*)] \,, \tag{42}$$

$$\implies \frac{\partial}{\partial x^\mu} \int \frac{d^3 p}{(2\pi)^3 \, p_0} p^\mu f \psi - \int \frac{d^3 p}{(2\pi)^3 \, p_0} f \left[ p^\mu \frac{\partial \psi}{\partial x^\mu} + m F^\mu \frac{\partial \psi}{\partial p^\mu} \right] = I[\psi] \,,$$

where we have used the result $\int \frac{d^3p}{(2\pi)^3} \frac{\partial(\psi F^\mu f)}{\partial p^\mu} = 0$. The RHS of Equation (42) can be further simplified by using the symmetry properties of transition rate $W$, i.e., $W(p, p_*; p', p'_*) = W(p_*, p; p', p'_*) = W(p, p_*; p'_*, p') = W(p', p'_*; p, p_*)$ as [236] :

$$
\begin{aligned}
I[\psi] &= \frac{1}{4} \int \frac{d^3p}{(2\pi)^3 \, p_0} \frac{d^3p_*}{(2\pi)^3 \, p^0_*} \frac{d^3p'_*}{(2\pi)^3 \, p'^0_*} \frac{d^3p'}{(2\pi)^3 \, p'^0} \, W \\
&\quad [f'f'_*(1+af)(1+af_*) - ff_*(1+af')(1+af'_*)](\psi + \psi_* - \psi' - \psi'_*) \,,
\end{aligned}
\tag{43}
$$

where we defined $\psi(x, p_*) \equiv \psi_*$, $\psi(x, p') \equiv \psi'$, $\psi(x, p'_*) \equiv \psi'_*$ . Equation (42) with $I[\psi]$ given in Equation (43) is the general balance/transfer equation for a flow corresponding to $\psi$ . Since in collision, the four momenta of particles are conserved, the transition rate $W \propto \delta^4(p'^\alpha + p'^\alpha_* - p^\alpha - p^\alpha_*)$. The delta function contained in $W$ ensures energy–momentum conservation and puts the constraint $p'^\alpha + p'^\alpha_* - p^\alpha - p^\alpha_* = 0$ on the integrand. Now, let us define summational invariants, from which we will be able to prove the energy–momentum and particle conservation for the system.

- We will define a function $\psi(x, p^\alpha)$, usually known as summational invariant, with the following condition:

$$
\psi(x, p^\alpha) + \psi(x, p^\alpha_*) = \psi(x, p'^\alpha) + \psi(x, p'^\alpha_*), \text{ where, } p'^\alpha + p'^\alpha_* = p^\alpha + p^\alpha_* \,.
\tag{44}
$$

It is obvious that $p^\nu$, and 1 are two summational invariants, and the substitution of them in Equation (42) gives the balance equation for the stress–energy tensor and particle 4-current, respectively:

$$
\frac{\partial}{\partial x^\mu} \int \frac{d^3p}{(2\pi)^3 p^0} p^\mu p^\nu f - m \int \frac{d^3p}{(2\pi)^3 p^0} \, f \, F^\nu = 0 \,,
$$

$$
\implies \partial_\mu T^{\mu\nu} = m \int \frac{d^3p}{(2\pi)^3 p^0} \, f \, F^\nu \,,
\tag{45}
$$

$$
\text{and,} \quad \frac{\partial}{\partial x^\mu} \int \frac{d^3p}{(2\pi)^3 p^0} p^\mu f = 0 \,,
$$

$$
\implies \partial_\mu N^\mu = 0 \,.
\tag{46}
$$

We see that Equations (6) and (9) are exactly reproduced from Equations (45) and (46) in the absence of external forces.

### 3.2. Entropy Production and Equilibrium

Now, let us discuss two further kinetic theory concepts that have paramount importance in the theory of thermodynamics, namely, entropy flow and entropy production. Entropy flow $S^\mu$ and entropy density $s$ within the realm of kinetic theory may be defined by Equation (41) with $\psi = -\ln f + \left(\frac{1+af}{af}\right) \ln(1+af)$, i.e.,

$$
S^\mu = \int \frac{d^3p}{(2\pi)^3 p^0} \, p^\mu \left( -\ln f + \left(\frac{1+af}{af}\right) \ln(1+af) \right) f \,,
\tag{47}
$$

$$
s = u_\mu S^\mu \,.
\tag{48}
$$

If we substitute this $\psi$ in Equation (42), one finds, after some tedious algebraic, the following equation [236,237]:

$$
\begin{aligned}
&\partial_\mu \int \frac{d^3p}{(2\pi)^3 p^0} \, p^\mu \left( -\ln f + \left(\frac{1+af}{af}\right) \ln(1+af) \right) f = \frac{1}{4} \int ff_*(1+af') \\
&(1+af'_*) \left[ \ln \frac{f'f'_*(1+af)(1+af_*)}{ff_*(1+af')(1+af'_*)} \right] \left[ \frac{f'f'_*(1+af)(1+af_*)}{ff_*(1+af')(1+af'_*)} - 1 \right] d\chi
\end{aligned}
$$

$$\implies \partial_\mu S^\mu = \frac{1}{4} \int f f_* (1 + af')(1 + af'_*) \quad \left[ \ln \frac{f' f'_* (1 + af)(1 + af_*)}{f f_* (1 + af')(1 + af'_*)} \right]$$
$$\left[ \frac{f' f'_* (1 + af)(1 + af_*)}{f f_* (1 + af')(1 + af'_*)} - 1 \right] d\chi,$$

(49)

where $d\chi \equiv \frac{d^3 p}{(2\pi)^3 \, p_0} \frac{d^3 p_*}{(2\pi)^3 \, p_*^0} \frac{d^3 p'_*}{(2\pi)^3 \, p_*'^0} \frac{d^3 p'}{(2\pi)^3 \, p'^0}$. The RHS of Equation (49) gives the entropy production at each space–time location. Unlike the particle flow and energy–momentum flow, the entropy 4-flow ($S^\mu$) is not conserved, and in general, one can show from RHS of Equation (49) that entropy production is always positive [236,237]. Having discussed the entropy flow and entropy production, we will now discuss two more essential concepts in the kinetic theory, i.e., the concepts of local equilibrium distribution $f^0$ and global equilibrium distribution $f^{eq}$.

- The local equilibrium distribution ($f^0$) obeys the condition of detailed balance, which is expressed as follows :

$$f'^0 f'^0_* - f^0 f^0_* = 0, \text{ for classical particles},$$

(50)

$$f'^0 f'^0_* (1 + af^0)(1 + af^0_*) - f^0 f^0_* (1 + af'^0)(1 + af'^0_*) = 0, \text{ for quantum particles}.$$

(51)

Therefore, $C[f^0] = 0$ for local equilibrium distributions. Equation (50) can be written as:

$$f'^0 f'^0_* = f^0 f^0_*,$$
$$\implies \frac{f'^0 f'^0_*}{f^0 f^0_*} = 1,$$
$$\implies \ln \frac{f'^0 f'^0_*}{f^0 f^0_*} = 0,$$
$$\implies \ln f^0 + \ln f^0_* = \ln f'^0 + \ln f'^0_*.$$

(52)

Comparing Equation (52) with the definition of summational invariant given in Equation (44), we realize that $\ln f^0$ is a summational invariant. Since any scalar summational invariant can be expressed as [236]: $A(x) + B_\alpha(x) p^\alpha$, we have

$$\ln f^0 = A(x) + B_\alpha(x) p^\alpha,$$
$$\implies \quad f^0 = e^{A(x) + B_\alpha(x) p^\alpha}.$$

(53)

It is useful to make the reparameterization of variables with $A = \frac{\mu}{T}$ and $B_\alpha = \frac{-u_\alpha}{T}$, where $u_\alpha u^\alpha = 1$. One usually identifies $u_\alpha$, $\mu$, and $T$ as the fluid velocity, chemical potential, and temperature of the system. In these new variables, $f^0$ becomes:

$$f^0 = e^{-(u_\alpha p^\alpha - \mu)/T}.$$

(54)

Similarly for quantum particles, Equation (51) can be written as:

$$f'^0 f'^0_* (1 + af^0)(1 + af^0_*) = f^0 f^0_* (1 + af'^0)(1 + af'^0_*),$$
$$\implies \quad \frac{f^0 f^0_*}{(1 + af^0)(1 + af^0_*)} = \frac{f'^0 f'^0_*}{(1 + af'^0)(1 + af'^0_*)},$$
$$\implies \quad \ln \frac{f^0}{1 + af^0} + \ln \frac{f^0_*}{1 + af^0_*} = \ln \frac{f'^0}{1 + af'^0} + \ln \frac{f'^0_*}{1 + af'^0_*}.$$

(55)

Comparing Equation (56) with the definition of summational invariant given in Equation (44), we see that $\ln \frac{f^0}{1+af^0}$ is a summational invariant. Therefore, we can write,

$$
\ln \frac{f^0}{1+af^0} = A(x) + B_\alpha(x)p^\alpha \,,
$$

$$
\implies \quad \ln \frac{f^0}{1+af^0} = \frac{\mu - u_\alpha p^\alpha}{T} \,, \tag{56}
$$

$$
\implies \quad f^0 = \frac{1}{e^{(u_\alpha p^\alpha - \mu)/T} - a} \,.
$$

- The global equilibrium distribution $f^{eq}$ should satisfy the Boltzmann Equation, and entropy production defined in Equation (49) should vanish.

In this review, we will be concerned with local equilibrium distribution $f^0$ because, in the fluid dynamic limit, one usually expands the total distribution function $f$ around $f^0$ (see Section 4). Nevertheless, one can obtain more information on global equilibrium distribution $f^{eq}$ in Ref. [236].

### 3.3. Local Equilibrium Thermodynamics

We see in Section 3.2 that the local equilibrium distribution $f^0$ given in Equation (57) does not satisfy the BTE; rather, it causes the collision term on the RHS of the BTE to vanish. Therefore, this distribution is a good distribution for the starting point of an approximation scheme and, in fact, in Chapman–Enskog(CE) expansion, one writes down the actual distribution function that satisfies the BTE as a power series around $f^0$ (we will discuss on it elaborately in Section 4). Nevertheless, we will see that the $f^0$ given in Equation (57) gives the stress–energy tensor, and particle four flow that is similar to an ideal fluid as described in Equations (11) and (12) of Section 2. The particle flow and stress–energy tensor defined by $f^0$ are:

$$
N^{(0)\mu} \equiv \int \frac{d^3p}{(2\pi)^3\, p_0}\, p^\mu f^0 \,, \tag{57}
$$

$$
T^{(0)\mu\nu} \equiv \int \frac{d^3p}{(2\pi)^3\, p_0}\, p^\mu p^\nu f^0 \,. \tag{58}
$$

The integrations in Equations (57) and (58) can be evaluated by moving to the LRF of the fluid. Here, we will write the result,

$$
N^{(0)\mu} = I_{10}\left(\frac{\mu}{T}, \frac{1}{T}\right) u^\mu, \tag{59}
$$

$$
T^{(0)\mu\nu} = I_{20}\left(\frac{\mu}{T}, \frac{1}{T}\right) u^\mu u^\nu - I_{21}\left(\frac{\mu}{T}, \frac{1}{T}\right) \Delta^{\mu\nu} \,, \tag{60}
$$

where the integrals $I_{nq}$ are defined as:

$$
\begin{aligned}
I_{nq} &\equiv \frac{1}{(2q+1)!!} \int \frac{d^3p}{(2\pi)^3\, p_0}\, f^0\, (u_\alpha p^\alpha)^{n-2q} \left((u_\alpha p^\alpha)^2 - m^2\right)^q \\
&= \frac{1}{(2q+1)!!} \int \frac{d^3p}{(2\pi)^3\, p_0}\, f^0\, (u_\alpha p^\alpha)^{n-2q} \left(-\Delta_{\alpha\beta} p^\alpha p^\beta\right)^q \,.
\end{aligned} \tag{61}
$$

$n^{(0)} \equiv I_{10}$, $\mathcal{E}^{(0)} \equiv I_{20}$, and $P^{(0)} \equiv I_{21}$ are, respectively, the LRF number density, energy density, and pressure given by $f^0$. In the LRF $u^\mu \to (1, \vec{0})$, we have $f^0 = 1/(e^{(E-\mu)/T} - a)$, and $I_{nq}$ can be written as:

$$
\begin{aligned}
I_{nq} &= \frac{1}{(2q+1)!!} \int \frac{d^3p}{(2\pi)^3 \, p_0} \, f^0 \, E^{n-2q} \, (E^2 - m^2)^q \\
&= \frac{1}{(2q+1)!!} \int \frac{d^3p}{(2\pi)^3} \, f^0 \, E^{n-2q-1} \, |\vec{p}|^{2q}.
\end{aligned}
\tag{62}
$$

The integral in Equation (62) can be written with the change of variable: $p = m \, sinhx$, $\Longrightarrow$ $E = \sqrt{\vec{p}^2 + m^2} = m \, coshx$ as:

$$
I_{nq} = \frac{m^{2+n}}{2\pi^2 \, (2q+1)!!} \int_0^\infty \frac{dx \, (sinhx)^{2q+2} \, (coshx)^{n-2q}}{e^{m\beta coshx - \alpha} - a} \, ,
\tag{63}
$$

where we defined $\beta \equiv \frac{1}{T}$ and $\alpha \equiv \frac{\mu}{T}$. A comparison of the form of Equations (59) and (60) with Equations (12) and (11), respectively, suggests that the distribution $f^0$ gives rise to the same form of the stress–energy tensor and particle flow as that of an ideal fluid. In Section 2, we discussed the properties of ideal fluids without kinetic theory, i.e., without introducing a distribution function; therefore, one could choose any function $\mathcal{E}_0(P_0, n_0)$ to close the EOM. But with a background in kinetic theory, the function $\mathcal{E}^{(0)}(P^{(0)}, n^{(0)})$ has to be calculated from the expressions $n^{(0)} = I_{10}$, $\mathcal{E}^{(0)} = I_{20}$, and $P^{(0)} = I_{21}$. We may write the local equilibrium entropy flow and entropy density with the help of $f^0$ as:

$$
S^{(0)\mu} = \int \frac{d^3p}{(2\pi)^3 p^0} \, p^\mu \left( -\ln f^0 + \left( \frac{1 + af^0}{af^0} \right) \ln\left(1 + af^0\right) \right) f^0 \, ,
\tag{64}
$$

$$
s^{(0)} = u_\mu S^{(0)\mu} = u_\mu \int \frac{d^3p}{(2\pi)^3 p^0} \, p^\mu \left( -\ln f^0 + \left( \frac{1 + af^0}{af^0} \right) \ln\left(1 + af^0\right) \right) f^0 \, .
\tag{65}
$$

Now, we will simplify Equation (65) by moving to LRF and using the transformation $p = m \, sinhx$ as:

$$
\begin{aligned}
s^{(0)} &= \int \frac{p^2 dp}{2\pi^2} \left( -f^0 \, \ln f^0 + \frac{1 + af^0}{a} \, \ln\left(1 + af^0\right) \right) \\
\Longrightarrow s^{(0)} &= \frac{m^3}{2\pi^2} \left[ \int -f^0 \, \ln f^0 \, (sinhx)^2 \, coshx \, dx \right. \\
&\qquad \left. + \frac{1}{a} \int (1 + af^0) \, \ln\left(1 + af^0\right)(sinhx)^2 \, coshx \, dx \right] ,
\end{aligned}
\tag{66}
$$

the two integrals in Equation (67) can be simplified further by using the technique integration by parts with $-f^0 \, \ln f^0$ and $(1 + af^0) \, \ln(1 + af^0)$ as the first functions, respectively; by doing this and noticing that the integrated part vanishes for both the integrals, one has:

$$
\begin{aligned}
s^{(0)} &= \frac{m^3}{6\pi^2} \int \left( \ln \frac{f^0}{1 + af^0} \right) (sinhx)^3 \, \frac{df^0}{dx} \, dx \\
\Longrightarrow \quad s^{(0)} &= \frac{m^3}{6\pi^2} \int (-m \, \beta \, coshx + \alpha) \, (sinhx)^3 \, \frac{df^0}{dx} \, dx \, .
\end{aligned}
\tag{67}
$$

We can further simplify it using the technique integration by parts with $(-m \, \beta \, coshx + \alpha) \, (sinhx)^3$ as the first function and noticing that the integrated part vanishes:

$$
\begin{aligned}
s^{(0)} &= -\frac{m^3}{6\pi^2} \int f^0 \left( -3m\,\beta\,(sinhx)^2\,(coshx)^2 - m\,\beta\,(sinhx)^4 + 3\alpha\,(sinhx)^2\,coshx \right) dx \\
\implies s^{(0)} &= \frac{m^4\beta}{2\pi^2} \int f^0\,(sinhx)^2\,(coshx)^2\,dx + \frac{m^4\beta}{6\pi^2} \int f^0\,(sinhx)^4\,dx \\
&\quad - \frac{m^3\alpha}{2\pi^2} \int f^0\,(sinhx)^2\,coshx\,dx \\
\implies s^{(0)} &= \beta\,I_{20} + \beta\,I_{21} - \alpha\,I_{10} \\
\implies s^{(0)} &= \frac{\mathcal{E}^{(0)}}{T} + \frac{P^{(0)}}{T} - \frac{\mu n^{(0)}}{T} \,,
\end{aligned}
\tag{68}
$$

where to obtain the last two lines, we used Equation (63). Equation (69) is Euler's equation, and we proved that it remains valid in local equilibrium. Now, we will prove the validity of the first law of thermodynamics for local equilibrium. To fulfill this purpose, we will define the following integrals and identities,

$$
J_{nq} \equiv \frac{1}{(2q+1)!!} \int \frac{d^3p}{(2\pi)^3\,p_0} f^0 (1 + af^0)(u_\alpha p^\alpha)^{n-2q}((u_\alpha p^\alpha)^2 - m^2)^q \,,
\tag{69}
$$

$$
= \frac{1}{(2q+1)!!} \int \frac{d^3p}{(2\pi)^3\,p_0} f^0 (1 + af^0)(u_\alpha p^\alpha)^{n-2q}(-\Delta_{\alpha\beta} p^\alpha p^\beta)^q \,,
\tag{70}
$$

$$
dI_{nq} = -J_{n+1\,q}\,d\beta + J_{nq}\,d\alpha, \text{ where } \frac{\partial I_{nq}}{\partial \beta} = -J_{n+1\,q}, \text{ and } \frac{\partial I_{nq}}{\partial \alpha} = J_{nq} \,,
\tag{71}
$$

$$
\beta J_{nq} = I_{n-1\,q-1} + (n - 2q)I_{n-1\,q},
\tag{72}
$$

$$
I_{nq} = m^2 I_{n-2\,q} + (2q+3)I_{n\,q+1} \,,
\tag{73}
$$

$$
J_{nq} = m^2 J_{n-2\,q} + (2q+3)J_{n\,q+1} \,,
\tag{74}
$$

where we have used the notation $\alpha \equiv \frac{\mu}{T}$ and $\beta \equiv \frac{1}{T}$. To show the validity of first law of thermodynamics, first, we will rewrite Equation (69) in terms of $\alpha$ and $\beta$ and take differentials in both sides as follows:

$$
\begin{aligned}
s^{(0)} &= -\alpha n^{(0)} + \beta \mathcal{E}^{(0)} + \beta I_{21} \,, \\
\implies ds^{(0)} &= -\alpha dn^{(0)} + \beta d\mathcal{E}^{(0)} - n^{(0)} d\alpha + \mathcal{E}^{(0)} d\beta + I_{21} d\beta + \beta dI_{21} \,, \\
\implies ds^{(0)} &= -\alpha dn^{(0)} + \beta d\mathcal{E}^{(0)} - n^{(0)} d\alpha + \mathcal{E}^{(0)} d\beta + I_{21} d\beta + \beta\left( \frac{\partial I_{21}}{\partial \alpha} d\alpha + \frac{\partial I_{21}}{\partial \beta} d\beta \right) \,, \\
\implies ds^{(0)} &= -\alpha dn^{(0)} + \beta d\mathcal{E}^{(0)} - n^{(0)} d\alpha + \mathcal{E}^{(0)} d\beta + I_{21} d\beta + \beta(J_{21} d\alpha - J_{31} d\beta) \,, \\
\implies ds^{(0)} &= -\alpha dn^{(0)} + \beta d\mathcal{E}^{(0)} + (\beta J_{21} - n^{(0)})d\alpha + (\mathcal{E}^{(0)} + I_{21} - \beta J_{31})d\beta \,,
\end{aligned}
\tag{75}
$$

(using Equation (71)). Now, from Equation (72) we have $\beta J_{21} = I_{10} = n^{(0)}$ and $\beta J_{31} = I_{20} + I_{21} = \mathcal{E}^{(0)} + I_{21}$. Using this in Equation (75), we have

$$
ds^{(0)} = -\alpha dn^{(0)} + \beta d\mathcal{E}^{(0)} \,,
$$

$$
\implies ds^{(0)} = \frac{1}{T}(d\mathcal{E}^{(0)} - \mu dn^{(0)}) \,.
\tag{76}
$$

Equation (76) is the first law of thermodynamics in local equilibrium. We can establish an important equation between the differentials of the intensive thermodynamic variables $\mu, T,$ and $P^{(0)}$ with the help of Euler's Equations (69) and (76) as:

$$s^{(0)} = -\frac{\mu n^{(0)}}{T} + \frac{\mathcal{E}^{(0)}}{T} + \frac{P^{(0)}}{T} \, ,$$

$$\implies \quad ds^{(0)} = \frac{1}{T}[-d(\mu n^{(0)}) + d\mathcal{E}^{(0)} + dP^{(0)}] - [-\mu n^{(0)} + \mathcal{E}^{(0)} + P^{(0)}]\frac{dT}{T^2} \, ,$$

$$\implies \quad ds^{(0)} = \frac{1}{T}[-\mu dn^{(0)} - n^{(0)}d\mu + d\mathcal{E}^{(0)} + dP^{(0)} - s^{(0)}dT] \, , \tag{77}$$

$$\implies \quad dP^{(0)} = n^{(0)}d\mu + s^{(0)}dT.$$

(using Equation (76)). One can rewrite Equation (78) with $\frac{\mu}{T} = \alpha$ and $\frac{1}{T}$ as independent variables in place of $\mu$ and $T$ as:

$$n^{(0)}d\left(\frac{\mu}{T}\right) = \frac{1}{T}dP^{(0)} + (\mathcal{E}^{(0)} + P^{(0)})d\left(\frac{1}{T}\right) \, . \tag{78}$$

Equations (78) and (78) are called Gibbs–Duhem equations, which proved to be very useful. We will use them to eliminate the chemical potential gradient while solving the Boltzmann equation in the Chapman–Enskog approximation.

## 4. Transport Coefficients in Chapman–Enskog Approximation

Whenever we have a many-particle system in non-equilibrium, its approach to equilibrium proceeds in two different steps; in the earlier stage, the non-equilibrium system approaches a state of local equilibrium where one can define local thermodynamic fields in terms of hydrodynamic/fluid velocity, temperature, and chemical potential. In the later stage, the non-uniformities or the gradients of the thermodynamic fields gradually disappear, and the system approaches a state of global equilibrium. In the Chapman–Enskog approximation (CEA), one looks for the solution of the BTE in this later stage [237]. To carry out CEA, let us write down the BTE in the absence of external forces :

$$p^\alpha \frac{\partial f}{\partial x^\alpha} = C[f] \, , \tag{79}$$

where $C[f]$ is the collision kernel. For now, we will discuss the layout of CEA with a general collision kernel $C[f]$. Nevertheless, for practical purposes, either one can use the $2 \leftrightarrow 2$ classical collision term given by Equation (39) or $2 \leftrightarrow 2$ quantum collision term given by Equation (40). One can also use different relaxation-type models for $C[f]$; we will discuss them in Section 5. Now, we will write down Equation (42) with respect to a generalized collision term as follows:

$$\frac{\partial}{\partial x^\mu} \int \frac{d^3p}{(2\pi)^3} \frac{1}{p^0} p^\mu f\psi - \int \frac{d^3p}{(2\pi)^3} \frac{1}{p^0} f\left[p^\mu \frac{\partial \psi}{\partial x^\mu}\right] = I[\psi] \, , \tag{80}$$

where $I[\psi] = \int C[f] \, \psi \, \frac{d^3p}{(2\pi)^3 p^0}$, and we put $F^\mu = 0$ since we are assuming that the system is free from external forces. Equation (80) gives the space–time evolution of the flow defined by $\psi$. If we take $\psi = 1$, $p^\nu$, and $-\ln f + \left(\frac{1+af}{af}\right)\ln(1 + af)$, we respectively obtain:

$$\partial_\mu N^\mu = \int C[f] \frac{d^3p}{(2\pi)^3 p^0} \, , \tag{81}$$

$$\partial_\mu T^{\mu\nu} = \int C[f] \, p^\nu \frac{d^3p}{(2\pi)^3 p^0} \, , \tag{82}$$

$$\text{and,} \quad \partial_\mu S^\mu = \int \ln\left(\frac{1+af}{f}\right) C[f] \frac{d^3p}{(2\pi)^3 p^0} \, , \tag{83}$$

From Equation (9), Equation (6), and the principle of increase in entropy, i.e., $\partial_\mu S^\mu \geq 0$, we obtain the following conditions on the collision kernel $C[f]$:

$$\int C[f] \frac{d^3p}{(2\pi)^3 \, p^0} = 0 \,, \tag{84}$$

$$\int C[f] \, p^\nu \frac{d^3p}{(2\pi)^3 \, p^0} = 0 \,, \tag{85}$$

$$\text{and,} \quad \int \ln\left(\frac{1+af}{f}\right) C[f] \frac{d^3p}{(2\pi)^3 \, p^0} \geq 0 \,, \tag{86}$$

We have already proved in Sections (3.1) and (3.2) that Equations (84)–(86) are satisfied for $2 \leftrightarrow 2$ collision kernel. For any model of collision kernel $C[f]$, one should check the validity of Equations (84)–(86).

To proceed with CEA, we will rewrite Equation (79) by using the decomposition of $\partial_\alpha$ described in Section 2 as:

$$p^\alpha(u_\alpha D + \nabla_\alpha)f = C[f] \,. \tag{87}$$

There are two length scales involved in Equation (87). One is the mean free path (mfp) $\lambda$, and the other is the length $L$ over which the distribution function $f$ significantly varies. The mfp $\lambda$ can be determined from the formula $\lambda = \frac{1}{n\sigma}$, where $n$ and $\sigma$ are the particle density and scattering cross-section, respectively. Meanwhile, the length $L$ is comparable with the linear dimension of the fluid container. Therefore, one can set an approximation scheme [7,20,235–237] to solve Equation (87) perturbatively by introducing dimensionless gradient operators $\nabla_\mu \equiv \frac{1}{L}\hat{\nabla}_\mu$ and $D \equiv \frac{1}{L}\hat{D}$ [7]. Multiplying Equation (87) by $\lambda$ and introducing the dimensionless gradients, we have:

$$Kn \, p^\alpha(u_\alpha \hat{D} + \hat{\nabla}_\alpha)f = \lambda C[f] \,, \tag{88}$$

where $Kn = \frac{\lambda}{L}$ is known as the Knusden number for the system. And then one perturbatively solves for $f$ by assuming $f = f^0 + (Kn)^1 f^1 + (Kn)^2 f^2 + \ldots\ldots\ldots = \sum_{n=0}^\infty (Kn)^n f^n$. Upon solving Equation (88), one finds that $f^1$ contains first-order derivatives of thermodynamic variables $u^\mu, \mu$, and $T$ and $f^2$ contains second-order derivatives of the variables $u^\mu, \mu$, and $T$, and so on. Alternatively, one can also introduce a formal parameter $\epsilon$ in the LHS of Equation (87) to write [236,237]:

$$\epsilon \, p^\alpha(u_\alpha D + \nabla_\alpha)f = C[f] \,, \tag{89}$$

with $f = f^0 + \epsilon^1 f^1 + \epsilon^2 f^2 + \ldots\ldots\ldots = \sum_{n=0}^\infty \epsilon^n f^n$, where $\epsilon$ is a book-keeping parameter introduced here to keep track of the order of approximation. Both the approaches are equivalent, and one can see the detailed derivation of fluid dynamics from these approaches in Refs. ([7,236,237]).

In this article, we will try to solve the BTE in a simplified manner by following Ref. ([236]). We will write the total solution of Equation (87) as $f = f^0 + f^0 \tilde{f}^0 \phi = f^0(1 + \tilde{f}^0 \phi)$, where $\tilde{f}^0 = 1 + af^0$ and $\phi$ is still an unknown function. Upon substitution of $f = f^0 + f^0 \tilde{f}^0 \phi$ in Equation (87) and keeping the terms, which are first-order in $Kn$, we have:

$$p^\alpha(u_\alpha D + \nabla_\alpha)f^0 = \mathcal{L}[\phi] \,. \tag{90}$$

$\mathcal{L}[\phi]$ is defined from the expression $C[f^0 + f^0 \tilde{f}^0 \phi] = \mathcal{L}[\phi] + \mathcal{O}(Kn)$, where the terms $\mathcal{O}(Kn)$ contain terms involving a two or higher power of $Kn$. Equation (90) may be called the first-order BTE. $\mathcal{L}[\phi]$ is different for different collision models and satisfies two fundamental properties,

$$\int \mathcal{L}[\phi] \, \frac{d^3 p}{(2\pi)^3 \, p^0} = 0 \,, \tag{91}$$

$$\int \mathcal{L}[\phi] \, p^\nu \, \frac{d^3 p}{(2\pi)^3 \, p^0} = 0 \,, \tag{92}$$

which come from Equations (84) and (85), respectively. Our starting point of calculation of transport coefficients and equations of fluid dynamics will be Equation (90), along with Equations (91) and (92). Once solution $\phi$ is known by solving Equation (90), we observe that one can write the total stress–energy tensor and the total particle flow by breaking them into two parts:

$$T^{\mu\nu} = \int \frac{d^3 p}{(2\pi)^3 \, p_0} \, p^\mu p^\nu \, (f^0 + f^0 \tilde{f}^0 \phi) \equiv T^{(0)\mu\nu} + T^{(D)\mu\nu},$$

where $T^{(0)\mu\nu}$ is the ideal part defined by the local equilibrium distribution $f^0$ as ,

$$T^{(0)\mu\nu} \equiv \int \frac{d^3 p}{(2\pi)^3 \, p_0} \, p^\mu p^\nu f^0 \,,$$

and $T^{(D)\mu\nu}$ is the out-of-equilibrium part defined by the correction: $f^0 \tilde{f}^0 \phi$ , to the local equilibrium distribution as ,

$$T^{(D)\mu\nu} \equiv \int \frac{d^3 p}{(2\pi)^3 \, p_0} \, p^\mu p^\nu f^0 \tilde{f}^0 \phi \,,$$

similarly,

$$N^{\mu} = \int \frac{d^3 p}{(2\pi)^3 \, p_0} \, p^\nu \, (f^0 + f^0 \tilde{f}^0 \phi) \equiv N^{(0)} + N^{(D)\mu},$$

where $N^{(0)\mu}$ is the ideal part defined by the local equilibrium distribution $f^0$ as ,

$$N^{(0)\mu} \equiv \int \frac{d^3 p}{(2\pi)^3 \, p_0} \, p^\mu f^0 \,,$$

and $N^{(D)\mu}$ is the out-of-equilibrium part defined by the correction: $f^0 \tilde{f}^0 \phi$ , to the local equilibrium distribution as ,

$$N^{(D)\mu} \equiv \int \frac{d^3 p}{(2\pi)^3 \, p_0} \, p^\mu f^0 \tilde{f}^0 \phi \,.$$

The fluid dynamical variables can be obtained by rewriting Equations (32) to (37) by the substitution $f = f^0 + f^0 \tilde{f}^0 \phi$ as follows :

$$\mathcal{E} = u_\mu u_\nu T^{(0)\mu\nu} + u_\mu u_\nu T^{(D)\mu\nu} \equiv \mathcal{E}^{(0)} + \delta\mathcal{E} \,, \tag{93}$$

$$P = -\frac{1}{3} \Delta_{\mu\nu} T^{(0)\mu\nu} - \frac{1}{3} \Delta_{\mu\nu} T^{(D)\mu\nu} \equiv P^{(0)} + \Pi, \implies \Pi = -\frac{1}{3} \Delta_{\mu\nu} T^{(D)\mu\nu} \,, \tag{94}$$

$$h^\mu = \Delta^\mu_\sigma \, u_\nu T^{(0)\mu\nu} + \Delta^\mu_\sigma \, u_\nu T^{(D)\mu\nu} = \Delta^\mu_\sigma \, u_\nu T^{(D)\mu\nu} \,, \tag{95}$$

$$\pi^{\mu\nu} = \Delta^{\mu\nu}_{\alpha\beta} T^{(0)\alpha\beta} + \Delta^{\mu\nu}_{\alpha\beta} T^{(D)\alpha\beta} = \Delta^{\mu\nu}_{\alpha\beta} T^{(D)\alpha\beta}, \tag{96}$$

$$n = u_\mu N^{(0)\mu} + u_\mu N^{(D)\mu} \equiv n^{(0)} + \delta n \,, \tag{97}$$

$$\nu^\mu = \Delta^\mu_\nu N^{(0)\nu} + \Delta^\mu_\nu N^{(D)\nu} = \Delta^\mu_\nu N^{(D)\nu} \,, \tag{98}$$

where we used the results $\Delta^\mu_\sigma \, u_\nu T^{(0)\mu\nu} = \Delta^{\mu\nu}_{\alpha\beta} T^{(0)\alpha\beta} = \Delta^\mu_\nu N^{(0)\nu} = 0$, which one can check explicitly by using the form of $T^{(0)\mu\nu}$ and $N^{(0)\mu}$ given in Section 3.3. Let us simplify the LHS

of Equation (90) by substituting the local equilibrium distribution $f^0 = 1/(e^{(u_\alpha p^\alpha - \mu)/T} - a)$ as:

$$p^\alpha u_\alpha \quad \left[ \frac{\partial f^0}{\partial(\frac{\mu}{T})} D\left(\frac{\mu}{T}\right) + \frac{\partial f^0}{\partial(\frac{1}{T})} D\left(\frac{1}{T}\right) + \frac{\partial f^0}{\partial u^\beta} D u^\beta \right] + p^\alpha \left[ \frac{\partial f^0}{\partial(\frac{\mu}{T})} \nabla_\alpha \left(\frac{\mu}{T}\right) + \right.$$

$$\left. \frac{\partial f^0}{\partial(\frac{1}{T})} \nabla_\alpha \left(\frac{1}{T}\right) + \frac{\partial f^0}{\partial u^\beta} \nabla_\alpha u_\beta \right] = \mathcal{L}[\phi] \,, \tag{99}$$

Again, simplifying Equation (99) by substituting the results $\frac{\partial f^0}{\partial(\frac{\mu}{T})} = f^0(1 + af^0) = f^0 \tilde{f}^0$, $\frac{\partial f^0}{\partial(\frac{1}{T})} = -f^0(1 + af^0)u^\beta p_\beta = -f^0 \tilde{f}^0 u^\beta p_\beta$, and $\frac{\partial f^0}{\partial u^\beta} = -f^0(1 + af^0)\frac{p_\beta}{T} = -f^0 \tilde{f}^0 \frac{p_\beta}{T}$ we have

$$f^0 \tilde{f}^0 \left[ p^\alpha u_\alpha \left( D\left(\frac{\mu}{T}\right) - u^\beta p_\beta D\left(\frac{1}{T}\right) - \frac{p_\beta}{T} D u^\beta \right) + p^\alpha \left( \nabla_\alpha \left(\frac{\mu}{T}\right) - u^\beta p_\beta \nabla_\alpha \left(\frac{1}{T}\right) - \frac{p_\beta}{T} \nabla_\alpha u^\beta \right) \right]$$

$$= \mathcal{L}[\phi] \tag{100}$$

Now, we will integrate Equation (100) with respect to the measure $\frac{d^3 p}{(2\pi)^3 p^0}$ to obtain:

$$\left[ J_{10} D\left(\frac{\mu}{T}\right) - J_{20} D\left(\frac{1}{T}\right) - \frac{1}{T} J_{20} u^\beta D u_\beta \right]$$

$$+ \quad \left[ J_{10} u^\alpha \nabla_\alpha \left(\frac{\mu}{T}\right) - J_{20} u^\alpha \nabla_\alpha \left(\frac{1}{T}\right) - \frac{1}{T}(J_{20} u^\alpha u^\beta - J_{21} \Delta^{\alpha\beta}) \nabla_\alpha u_\beta \right] = 0 \,, \tag{101}$$

where we have used the results:

$$\int f^0 \tilde{f}^0 (p^\alpha u_\alpha) \frac{d^3 p}{(2\pi)^3 p^0} = J_{10} \,,$$

$$\int f^0 \tilde{f}^0 (p^\alpha u_\alpha)^2 \frac{d^3 p}{(2\pi)^3 p^0} = J_{20} \,,$$

$$\int f^0 \tilde{f}^0 (p^\alpha u_\alpha) p^\beta \frac{d^3 p}{(2\pi)^3 p^0} = J_{20} u^\beta \,,$$

$$\int f^0 \tilde{f}^0 p^\alpha \frac{d^3 p}{(2\pi)^3 p^0} = J_{10} u^\alpha \,,$$

$$\text{and} \quad \int f^0 \tilde{f}^0 p^\alpha p^\beta \frac{d^3 p}{(2\pi)^3 p^0} = J_{20} u^\alpha u^\beta - J_{21} \Delta^{\alpha\beta} \,.$$

We can rewrite Equation (101) by using the properties of spatial and temporal gradients discussed in Section 2 as:

$$J_{10} D\left(\frac{\mu}{T}\right) - J_{20} D\left(\frac{1}{T}\right) = -\frac{J_{21}}{T} \nabla_\beta u^\beta \,. \tag{102}$$

On the other hand, if we multiply Equation (100) by $p^\lambda$ and integrate it with respect to the measure $\frac{d^3 p}{(2\pi)^3 p^0}$, we obtain:

$$J_{20} u^\lambda D\left(\frac{\mu}{T}\right) - J_{30} u^\lambda D\left(\frac{1}{T}\right) - \frac{1}{T} D u_\beta (J_{30} u^\lambda u^\beta - J_{31} \Delta^{\lambda\beta}) + (J_{20} u^\lambda u^\alpha - J_{21} \Delta^{\lambda\alpha}) \nabla_\alpha \left(\frac{\mu}{T}\right)$$

$$- (J_{30} u^\lambda u^\alpha - J_{31} \Delta^{\lambda\alpha}) \nabla_\alpha \left(\frac{1}{T}\right) - \frac{1}{T} \nabla_\alpha u_\beta (J_{30} u^\alpha u^\beta u^\lambda - J_{31}(\Delta^{\alpha\beta} u^\lambda + \Delta^{\alpha\lambda} u^\beta + \Delta^{\lambda\beta} u^\alpha)) \tag{103}$$

$$= 0 \,,$$

where we have used the identities:

$$\int f^0 \tilde{f}^0 p^\lambda (p^\alpha u_\alpha) \frac{d^3 p}{(2\pi)^3 \, p^0} = J_{20} u^\lambda \,,$$

$$\int f^0 \tilde{f}^0 p^\lambda (p^\alpha u_\alpha)^2 \frac{d^3 p}{(2\pi)^3 \, p^0} = J_{30} u^\lambda \,,$$

$$\int f^0 \tilde{f}^0 p^\lambda p^\alpha (u_\beta p^\beta) \frac{d^3 p}{(2\pi)^3 \, p^0} = J_{30} u^\lambda u^\alpha - J_{31} \Delta^{\lambda \alpha},$$

and $$\int f^0 \tilde{f}^0 p^\alpha p^\beta p^\lambda \frac{d^3 p}{(2\pi)^3 \, p^0} = J_{30} u^\alpha u^\beta u^\lambda - J_{31} \left( \Delta^{\alpha \beta} u^\lambda + \Delta^{\alpha \lambda} u^\beta + \Delta^{\lambda \beta} u^\alpha \right) .$$

Taking the projection of Equation (104) along fluid velocity, we have

$$J_{20} D\left(\frac{\mu}{T}\right) - J_{30} D\left(\frac{1}{T}\right) = -\frac{J_{31}}{T} \left(\nabla_\beta u^\beta\right) . \tag{104}$$

Taking the projection of Equation (104) perpendicular to fluid velocity, we have

$$\frac{1}{T} \Delta^{\lambda \beta} \Delta_{\lambda \sigma} Du_\beta J_{31} - J_{21} \Delta_{\lambda \sigma} \nabla^\alpha \left(\frac{\mu}{T}\right) + J_{31} \Delta_{\lambda \sigma} \nabla^\lambda \left(\frac{1}{T}\right) = 0 \,,$$

$$\implies \quad \frac{1}{T} J_{31} Du_\sigma - J_{21} \nabla_\sigma \left(\frac{\mu}{T}\right) + J_{31} \nabla_\sigma \left(\frac{1}{T}\right) = 0 \,, \tag{105}$$

$$\implies \quad \frac{1}{T} J_{31} Du_\sigma = J_{21} \nabla_\sigma \left(\frac{\mu}{T}\right) - J_{31} \nabla_\sigma \left(\frac{1}{T}\right) .$$

Now, we will show that Equations (102), (104), and (106) represent the conservation laws of particle 4-flow and the stress–energy tensor of an ideal fluid. To serve this purpose, we will rewrite Equation (102) as follows:

$$Dn^{(0)} = DI_{10} = \frac{\partial I_{10}}{\partial \alpha} D\alpha + \frac{\partial I_{10}}{\partial \beta} D\beta \,,$$

$$\implies \quad Dn^{(0)} = J_{10} D\left(\frac{\mu}{T}\right) - J_{20} D\left(\frac{1}{T}\right) \,, \tag{106}$$

where we have used Equation (71). From Equations (106) and (102), we obtain :

$$Dn^{(0)} = -\frac{J_{21}}{T} \nabla_\beta u^\beta = -n^{(0)} \nabla_\beta u^\beta \,. \tag{107}$$

Comparing Equation (107) with Equation (16), one sees that Equation (107) is similar to the particle flow conservation equation for ideal fluids with local equilibrium number density $n^{(0)}$ as the total number density. Similarly, by taking the time derivative of $\mathcal{E}^{(0)}$ and the space derivative of $P^{(0)}$ and putting them back in Equations (104) and (106), respectively, we have

$$D\mathcal{E}^{(0)} = DI_{20} = \frac{\partial I_{20}}{\partial \beta} D\beta + \frac{\partial I_{20}}{\partial \alpha} D\alpha \,,$$

$$\implies \quad D\mathcal{E}^{(0)} = -J_{30} D\left(\frac{1}{T}\right) + J_{20} D\left(\frac{\mu}{T}\right) \,, \tag{108}$$

$$\implies \quad D\mathcal{E}^{(0)} = -\frac{J_{31}}{T} \left(\nabla_\beta u^\beta\right) \,,$$

$$\implies \quad D\mathcal{E}^{(0)} = -(\mathcal{E}^{(0)} + P^{(0)}) \nabla_\beta u^\beta \,,$$

where we have used Equations (71) and (104), where to obtain the last equality, we used the identity $\beta J_{31} = I_{20} + I_{21} = \mathcal{E}^{(0)} + P^{(0)}$ .

$$\nabla_\sigma P^{(0)} = \nabla_\sigma I_{21} = \frac{\partial I_{21}}{\partial \beta} \nabla_\sigma \beta + \frac{\partial I_{21}}{\partial \alpha} \nabla_\sigma \alpha \,,$$

$$\implies \nabla_\sigma P^{(0)} = -J_{31}\Delta_\sigma\left(\frac{1}{T}\right) + J_{21}\Delta_\sigma\left(\frac{\mu}{T}\right) \,, \tag{109}$$

$$\implies \frac{1}{T}J_{31}Du_\sigma = \nabla_\sigma P^{(0)} \,,$$

$$\implies (\mathcal{E}^{(0)} + P^{(0)})Du_\sigma = \nabla_\sigma P^{(0)} \,.$$

where we have used Equations (71) and (106). Comparing Equations (109) and (110) with Equations (14) and (15), one sees that Equations (109) and (110) are similar to the stress–energy tensor conservation equation for ideal fluids with local equilibrium pressure $P^{(0)}$ and energy $\mathcal{E}^{(0)}$ as the total pressure and energy density. The conservation Equations (107)–(110) or equivalently Equations (102), (104), and (106) can be used to remove the time derivatives of $\frac{\mu}{T}$, $\frac{1}{T}$, and $u^\mu$ from Equation (100) . Using Equations (102), (104), and (106) in Equation (100), we have:

$$f^0 \tilde{f}^0 \left[ Q_2(p^\alpha u_\alpha)\nabla_\beta u^\beta - Q_1(u^\alpha p_\alpha)^2\nabla_\beta u^\beta - (u_\alpha p^\alpha)p_\beta\left(\frac{n^{(0)}}{\mathcal{E}^{(0)} + P^{(0)}}\nabla^\beta\left(\frac{\mu}{T}\right) - \nabla^\beta\left(\frac{1}{T}\right)\right) \right]$$

$$+ f^0 \tilde{f}^0 p^\alpha \left[ \nabla_\alpha\left(\frac{\mu}{T}\right) - u^\beta p_\beta \nabla_\alpha\left(\frac{1}{T}\right) - \frac{p_\beta}{T}\nabla_\alpha u^\beta \right] = \mathcal{L}[\phi] \,, \tag{110}$$

where we used $D\left(\frac{1}{T}\right) = \frac{(\mathcal{E}^{(0)}+P^{(0)})J_{10} - n^{(0)}J_{20}}{D_{20}}\nabla^\beta u_\beta \equiv Q_1 \nabla^\beta u_\beta \,,\ D\left(\frac{\mu}{T}\right) = \frac{(\mathcal{E}^{(0)}+P^{(0)})J_{20} - n^{(0)}J_{30}}{D_{20}}\nabla^\beta u_\beta \equiv Q_2 \nabla^\beta u_\beta$, and defined $D_{20} \equiv J_{30}J_{10} - J_{20}^2$ . The terms of Equation (110) can be rearranged to obtain:

$$\left( Q_2(p^\alpha u_\alpha) - Q_1(p^\alpha u_\alpha)^2 \right)\nabla_\beta u^\beta - \frac{p^\alpha p^\beta}{T}\nabla_\alpha u_\beta - p_\beta\left[\frac{n^{(0)}}{\mathcal{E}^{(0)} + P^{(0)}}u_\alpha p^\alpha - 1\right]\nabla^\beta\left(\frac{\mu}{T}\right) = \frac{\mathcal{L}[\phi]}{f^0 \tilde{f}^0} \,, \tag{111}$$

by using the properties of the $\Delta$ projectors we discussed in Section 2, we can show that $\nabla_\alpha u_\beta = \nabla_{\langle\alpha} u_{\beta\rangle} + \frac{1}{3}\Delta_{\alpha\beta}(\nabla^\lambda u_\lambda)$; substituting this in Equation (111) we obtain,

$$\left( Q_2(p^\alpha u_\alpha) - Q_1(p^\alpha u_\alpha)^2 - \frac{1}{3T}\left(m^2 - \left(u_\alpha p^\alpha\right)^2\right) \right)\nabla_\beta u^\beta - \frac{p^\alpha p^\beta}{T}\nabla_{\langle\alpha} u_{\beta\rangle} -$$

$$p_\beta\left[\frac{n^{(0)}}{\mathcal{E}^{(0)} + P^{(0)}}u_\alpha p^\alpha - 1\right]\nabla^\beta\left(\frac{\mu}{T}\right) = \frac{\mathcal{L}[\phi]}{f^0 \tilde{f}^0} \,. \tag{112}$$

One can use the Gibbs–Duhem Equation (78) to convert the spatial derivative of $\frac{\mu}{T}$ into the spatial derivative of $T$ and $P$. From Equation (78), we have

$$n^{(0)}\nabla_\alpha\left(\frac{\mu}{T}\right) = \frac{1}{T}\nabla_\alpha P^{(0)} + (\mathcal{E}^{(0)} + P^{(0)})\nabla_\alpha\left(\frac{1}{T}\right) \,,$$

$$\implies \nabla_\alpha\left(\frac{\mu}{T}\right) = \frac{1}{n^{(0)}T}\nabla_\alpha P^{(0)} + \frac{(\mathcal{E}^{(0)} + P^{(0)})}{n^{(0)}}\nabla_\alpha\left(\frac{1}{T}\right) \,. \tag{113}$$

Using Equation (113) in Equation (112), we have

$$\left[ Q_2(p^\alpha u_\alpha) - Q_1(p^\alpha u_\alpha)^2 - \frac{1}{3T}(m^2 - \left(u_\alpha p^\alpha\right)^2) \right]\nabla_\beta u^\beta - \frac{p^\alpha p^\beta}{T}\nabla_{\langle\alpha} u_{\beta\rangle}$$

$$+ \frac{1}{T}\left[ \left((u_\alpha p^\alpha)p_\beta - \frac{\mathcal{E}^{(0)} + P^{(0)}}{n^{(0)}}p_\beta\right)\left(\frac{1}{T}\nabla^\beta T - \frac{1}{\mathcal{E}^{(0)} + P^{(0)}}\nabla^\beta P^{(0)}\right) \right] = \frac{\mathcal{L}[\phi]}{f^0 \tilde{f}^0} \tag{114}$$

Since Equation (114) is an inhomogeneous equation in $\phi$, one can assume a solution of the form $\phi = \phi_h + \phi_p$, where $\phi_h$ is the solution of the homogenous equation $\mathcal{L}[\phi_h] = 0$ and $\phi_p$ is the particular solution.

### 4.1. Transport Coefficients in Chapman–Enskog Approximation for $2 \leftrightarrow 2$ Collision term

In this section, we will derive all the transport coefficients, i.e., bulk viscosity, shear viscosity, particle diffusion, and energy diffusion coefficient of a system, by assuming the collision term $C[f]$ of the form Equation (40). Equation (90) for the $2 \leftrightarrow 2$ collision term becomes:

$$
\begin{aligned}
p^\alpha (u_\alpha D + \nabla_\alpha) f^0 \;=\; & -\int \frac{d^3 p_*}{(2\pi)^3 \, p_*^0} \frac{d^3 p_*'}{(2\pi)^3 \, p_*'^0} \frac{d^3 p'}{(2\pi)^3 \, p'^0} \; W f^0 f_*^0 (1 + a f'^0)(1 + a f_*'^0) \\
& (\phi + \phi^* - \phi' - \phi_*') ,
\end{aligned}
\tag{115}
$$

where $\mathcal{L}[\phi] = -\int \frac{d^3 p_*}{(2\pi)^3 \, p_*^0} \frac{d^3 p_*'}{(2\pi)^3 \, p_*'^0} \frac{d^3 p'}{(2\pi)^3 \, p'^0} \; W f^0 f_*^0 (1 + a f'^0)(1 + a f_*'^0)(\phi + \phi^* - \phi' - \phi_*')$, which can be derived by straight forward calculations from Equation (40). The operator $\mathcal{L}[\phi]$ enjoys the following important property:

$$
\int \frac{d^3 p}{(2\pi)^3 \, p^0} \; \psi \, \mathcal{L}[\phi] = \int \frac{d^3 p}{(2\pi)^3 \, p^0} \; \phi \, \mathcal{L}[\psi] ,
\tag{116}
$$

where $\psi$ is an arbitrary function of $x^\mu$ and $p^\mu$. One sees by using Equation (116) that Equations (91) and (92) are valid for the current $\mathcal{L}[\phi]$. The simplified version of LHS of Equation (115) is already obtained in Equation (114); therefore, one has the following equation to solve:

$$
\begin{aligned}
& \left[ Q_2 (p^\alpha u_\alpha) - Q_1 (p^\alpha u_\alpha)^2 - \frac{1}{3T}\left(m^2 - \left(u_\alpha p^\alpha\right)^2\right) \right] \nabla_\beta u^\beta - \frac{p^\alpha p^\beta}{T} \nabla_{\langle \alpha} u_{\beta \rangle} \\
& + \frac{1}{T} \left[ \left( (u_\alpha p^\alpha) p_\beta - \frac{\mathcal{E}^{(0)} + P^{(0)}}{n^{(0)}} p_\beta \right) \left( \frac{1}{T} \nabla^\beta T - \frac{1}{\mathcal{E}^{(0)} + P^{(0)}} \nabla^\beta P^{(0)} \right) \right] \\
& = -\frac{1}{f^0 \tilde{f}^0} \int \frac{d^3 p_*}{(2\pi)^3 \, p_*^0} \frac{d^3 p_*'}{(2\pi)^3 \, p_*'^0} \frac{d^3 p'}{(2\pi)^3 \, p'^0} \; W f^0 f_*^0 (1 + a f'^0)(1 + a f_*'^0) \\
& (\phi + \phi^* - \phi' - \phi_*') ,
\end{aligned}
\tag{117}
$$

since the linear integral operator on the RHS of Equation (118) only acts on momentum variables, one can guess an approximate solution to Equation (118) in the following form [236]:

$$
\begin{aligned}
\phi = \phi_h + & \left[ a_0 + a_1 \frac{u^\alpha p_\alpha}{T} + a_2 \left( \frac{u_\alpha p^\alpha}{T} \right)^2 \right] \nabla_\beta u^\beta + a_5 p_\alpha p_\beta \nabla^{\langle \alpha} u^{\beta \rangle} \\
& + \left( a_3 + a_4 \frac{u_\alpha p^\alpha}{T} \right) p_\beta \left( \frac{1}{T} \nabla^\beta T - \frac{1}{\mathcal{E}^{(0)} + P^{(0)}} \nabla^\beta P^{(0)} \right) ,
\end{aligned}
\tag{118}
$$

where in Equation (118) the coefficients $a_0, a_1, a_2, a_3,$ and $a_4$ are assumed to be independent of $p^\alpha$. The homogenous solution $\phi_h$ can be written in the following general form: $\phi_h = \tilde{a}_0 + b_\alpha p^\alpha$, where $\tilde{a}_0$ and $b_\alpha$ are independent of $p^\alpha$. By using the decomposition described in Section 2, we can write $b_\alpha = (b_\beta u^\beta) u_\alpha + \Delta_{\alpha\beta} b^\beta$. Using this decomposition of $b_\alpha$, one can write $\phi_h = \tilde{a}_0 + \tilde{b}_0 u_\alpha p^\alpha + \tilde{b}_\alpha p^\alpha$, where we defined $\tilde{b}_0 \equiv b_\beta u^\beta$ and $\tilde{b}_\alpha \equiv \Delta^{\alpha\beta} b_\beta$. The approximate solution $\phi$ with this decomposition of $\phi_h$ can be written as:

$$
\begin{aligned}
\phi = \tilde{a}_0 + \tilde{b}_0 u_\alpha p^\alpha + \tilde{b}_\alpha p^\alpha + & \left[ a_0 + a_1 \frac{u^\alpha p_\alpha}{T} + a_2 \left( \frac{u_\alpha p^\alpha}{T} \right)^2 \right] \nabla_\beta u^\beta + a_5 p_\alpha p_\beta \nabla^{\langle \alpha} u^{\beta \rangle} \\
+ & \left( a_3 + a_4 \frac{u_\alpha p^\alpha}{T} \right) p_\beta \left( \frac{1}{T} \nabla^\beta T - \frac{1}{\mathcal{E}^{(0)} + P^{(0)}} \nabla^\beta P^{(0)} \right) ,
\end{aligned}
\tag{119}
$$

For the expression of $\phi$ guessed in Equation (119), the $T^{(D)\mu\nu}$ is given by:

$$
\begin{aligned}
T^{(D)\mu\nu} &= \int f^0 \tilde{f}^0 \phi \, p^\mu p^\nu \frac{d^3 p}{(2\pi)^3 \, p^0} = (\tilde{a}_0 J_{20} + \tilde{b}_0 J_{30}) u^\mu u^\nu - (\tilde{a}_0 J_{21} + \tilde{b}_0 J_{31}) \Delta^{\mu\nu} \\
&\quad - J_{31} (\tilde{b}^\mu u^\nu + \tilde{b}^\nu u^\mu) + \left[ \left( a_0 J_{20} + \frac{a_1}{T} J_{30} + \frac{a_2}{T^2} J_{40} \right) u^\mu u^\nu \right. \\
&\quad \left. - \left( a_0 J_{21} + \frac{a_1}{T} J_{31} + \frac{a_2}{T^2} J_{41} \right) \Delta^{\mu\nu} \right] \nabla_\beta u^\beta \\
&\quad - \left( a_3 J_{31} + \frac{a_4}{T} J_{41} \right) \left[ \left( \frac{1}{T} \nabla^\mu T - \frac{1}{\mathcal{E}^{(0)} + P^{(0)}} \nabla^\mu P^{(0)} \right) u^\nu \right. \\
&\quad \left. + \left( \frac{1}{T} \nabla^\nu T - \frac{1}{\mathcal{E}^{(0)} + P^{(0)}} \nabla^\nu P^{(0)} \right) u^\mu \right] \\
&\quad + 2 a_5 J_{42} \nabla^{\langle \mu} u^{\nu \rangle} \,.
\end{aligned}
\tag{120}
$$

To uniquely fix the form of $\phi$, one must provide five matching conditions to be satisfied by $\phi$. Traditionally, the matching conditions are provided by choosing a particular hydrodynamic frame and setting the out-of-equilibrium thermodynamic variables to zero. The first three matching conditions with the LF choice arise by defining the fluid velocity field, and the other two arise by setting $\delta\mathcal{E} = 0$ and $\delta n = 0$. The $u^\mu$ in LF frame is defined as:

$$
\begin{aligned}
T^{\mu\nu} u_\nu &= \mathcal{E} u^\mu \,, \\
u_\nu T^{(0)\mu\nu} + u_\nu T^{(D)\mu\nu} &= \mathcal{E} u^\mu \,.
\end{aligned}
\tag{121}
$$

Using Equations (60) and (93) in Equation (121), we have

$$
\begin{aligned}
\mathcal{E}^{(0)} u^\mu + u_\nu T^{(D)\mu\nu} &= (\mathcal{E}^{(0)} + \delta\mathcal{E}) u^\mu \,, \\
\int \frac{d^3 p}{(2\pi)^3 \, p_0} \, p^\mu (u_\nu p^\nu) f^0 \tilde{f}^0 \phi &= \delta\mathcal{E} \, u^\mu \,.
\end{aligned}
\tag{122}
$$

Taking the projection of Equation (122) perpendicular to fluid velocity $u^\mu$, we have

$$
\Delta_{\mu\lambda} \int \frac{d^3 p}{(2\pi)^3 \, p_0} \, p^\mu (u_\nu p^\nu) f^0 \tilde{f}^0 \phi = 0 \,,
\tag{123}
$$

Now, we will set the out-of-equilibrium number density and energy density to zero by writing,

$$
\mathcal{E} = u_\mu u_\nu T^{\mu\nu} = u_\mu u_\nu T^{(0)\mu\nu} + u_\mu u_\nu T^{(D)\mu\nu} = \mathcal{E}^{(0)} \,,
\tag{124}
$$

$$
n = u_\mu N^\mu = u_\mu N^{(0)\mu} + u_\mu N^{(D)\mu} = n^{(0)} \,.
\tag{125}
$$

By using Equations (93) and (97), we see that Equations (124) and (125) lead to the following constraints on $\phi$:

$$
\delta\mathcal{E} = u_\mu u_\nu T^{(D)\mu\nu} = \int \frac{d^3 p}{(2\pi)^3 \, p_0} \, (u_\mu p^\mu)^2 f^0 \tilde{f}^0 \phi = 0 \,,
\tag{126}
$$

$$
\delta n = u_\mu N^{(D)\mu} = \int \frac{d^3 p}{(2\pi)^3 \, p_0} \, (u_\mu p^\mu) f^0 \tilde{f}^0 \phi = 0 \,.
\tag{127}
$$

Therefore, in LF, out of five matching conditions, three are given by Equation (123) and two by Equations (126) and (127), respectively.

The constraints set by Equations (123), (126), and (127) can be safely applied to $\phi_h$ and $\phi_p$ separately. Since $\phi_p$ is a linear combination of independent dissipative forces, one can also apply the constraints separately to the coefficients of each dissipative force. Applying the matching condition given by Equation (127), which matches the non-equilibrium number density to the local equilibrium number density, we have

$$\tilde{a}_0 J_{10} + \tilde{b}_0 J_{20} = 0 \,, \tag{128}$$

$$a_0 J_{10} + \frac{a_1}{T} J_{20} + \frac{a_2}{T^2} J_{30} = 0 \,. \tag{129}$$

Applying the matching condition given by Equation (126), which matches the non-equilibrium energy density to the local equilibrium energy density, we have

$$\tilde{a}_0 J_{20} + \tilde{b}_0 J_{30} = 0 \,, \tag{130}$$

$$a_0 J_{20} + \frac{a_1}{T} J_{30} + \frac{a_2}{T^2} J_{40} = 0 \,. \tag{131}$$

By solving Equations (128) and (130), one obtains $\tilde{a}_0 = \tilde{b}_0 = 0$. Again, applying the matching condition given by Equation (123), which sets the dissipative part of energy flow to zero, we have

$$-J_{31} \tilde{b}_\mu = 0, \implies \tilde{b}_\mu = 0 \,, \tag{132}$$

$$a_3 J_{31} + \frac{a_4}{T} J_{41} = 0 \,. \tag{133}$$

Using the result $\tilde{a}_0 = \tilde{b}_0 = 0, \tilde{b}_\mu = 0$ and Equations (131) and (133), we can rewrite Equation (121) as:

$$T^{(D)\mu\nu} = \left[ \left( a_0 J_{21} + \frac{a_1}{T} J_{31} + \frac{a_2}{T^2} J_{41} \right) \Delta^{\mu\nu} \right] \nabla_\beta u^\beta + 2 a_5 J_{42} \nabla^{\langle \mu} u^{\nu \rangle} \,. \tag{134}$$

Solving Equations (129) and (131), one obtains:

$$a_1 = \frac{a_2}{T} \frac{-J_{30}(J_{40} J_{10} - J_{20} J_{30})}{J_{10}(J_{30}^2 - J_{20} J_{40}) + J_{40} J_{10} - J_{20} J_{30}} \,, a_0 = \frac{a_2}{T^2} \frac{J_{30}(J_{40} J_{20} - J_{30}^2)}{J_{10}(J_{30}^2 - J_{20} J_{40}) + J_{40} J_{10} - J_{20} J_{30}} \,. \tag{135}$$

Using Equation (135), we obtain:

$$a_0 J_{21} + \frac{a_1}{T} J_{31} + \frac{a_2}{T^2} J_{41} = \frac{a_2}{T^2} \frac{(J_{30}^2 - J_{20} J_{40})(J_{41} J_{10} - J_{21} J_{30}) + (J_{40} J_{10} - J_{20} J_{30})(J_{41} - J_{30} J_{31})}{J_{10}(J_{30}^2 - J_{20} J_{40}) + (J_{40} J_{10} - J_{20} J_{30})}$$

$$= \frac{a_2}{T^2} R_1 \,, \tag{136}$$

where $R_1 \equiv \dfrac{(J_{30}^2 - J_{20} J_{40})(J_{41} J_{10} - J_{21} J_{30}) + (J_{40} J_{10} - J_{20} J_{30})(J_{41} - J_{30} J_{31})}{J_{10}(J_{30}^2 - J_{20} J_{40}) + (J_{40} J_{10} - J_{20} J_{30})} \,.$

By substituting the result $\tilde{a}_0 = \tilde{b}_0 = 0, b_\mu = 0$ in Equation (119), we have

$$\phi = \left[ a_0 + a_1 \frac{u^\alpha p_\alpha}{T} + a_2 \left( \frac{u_\alpha p^\alpha}{T} \right)^2 \right] \nabla_\beta u^\beta + a_5 p_\alpha p_\beta \nabla^{\langle \alpha} u^{\beta \rangle}$$

$$+ \left( a_3 + a_4 \frac{u_\alpha p^\alpha}{T} \right) p_\beta \left( \frac{1}{T} \nabla^\beta T - \frac{1}{\mathcal{E}^{(0)} + P^{(0)}} \nabla^\beta P^{(0)} \right) \,, \tag{137}$$

Using Equation (137) in Equation (118), we have

$$
\begin{aligned}
&\left[ Q_2(p^\alpha u_\alpha) - Q_1(p^\alpha u_\alpha)^2 - \frac{1}{3T}(m^2 - (u_\alpha p^\alpha)^2) \right] \nabla_\beta u^\beta - \frac{p^\alpha p^\beta}{T} \nabla_{\langle \alpha} u_{\beta \rangle} \\
&+ \frac{1}{T}\left[ \left( (u_\alpha p^\alpha) p_\beta - \frac{\mathcal{E}^{(0)} + P^{(0)}}{n^{(0)}} p_\beta \right) \left( \frac{1}{T}\nabla^\beta T - \frac{1}{\mathcal{E}^{(0)} + P^{(0)}} \nabla^\beta P^{(0)} \right) \right] f^0 \tilde{f}^0 \\
&= \mathcal{L}\left[ \left( a_0 + a_1 \frac{u^\alpha p_\alpha}{T} + a_2 \left( \frac{u_\alpha p^\alpha}{T} \right)^2 \right) \nabla_\beta u^\beta + a_5 p_\alpha p_\beta \nabla^{\langle \alpha} u^{\beta \rangle} \right. \\
&\left. + \left( a_3 + a_4 \frac{u_\alpha p^\alpha}{T} \right) p_\beta \left( \frac{1}{T}\nabla^\beta T - \frac{1}{\mathcal{E}^{(0)} + P^{(0)}} \nabla^\beta P^{(0)} \right) \right],
\end{aligned} \tag{138}
$$

where out of $a_0, a_1$, and, $a_2$ only one is independent (see Equation (135)), and similarly out of $a_3$ and $a_4$ only one coefficient is independent (see Equation (133)). Using the result that $\mathcal{L}[1] = \mathcal{L}[p^\alpha] = 0$, we have

$$
\begin{aligned}
&\left[ Q_2(p^\alpha u_\alpha) - Q_1(p^\alpha u_\alpha)^2 - \frac{1}{3T}(m^2 - (u_\alpha p^\alpha)^2) \right] \nabla_\beta u^\beta - \frac{p^\alpha p^\beta}{T} \nabla_{\langle \alpha} u_{\beta \rangle} \\
&+ \frac{1}{T}\left[ \left( (u_\alpha p^\alpha) p_\beta - \frac{\mathcal{E}^{(0)} + P^{(0)}}{n^{(0)}} p_\beta \right) \left( \frac{1}{T}\nabla^\beta T - \frac{1}{\mathcal{E}^{(0)} + P^{(0)}} \nabla^\beta P^{(0)} \right) \right] f^0 \tilde{f}^0 \\
&= \frac{a_2}{T^2}(\nabla_\lambda u^\lambda) u_\alpha u_\beta \mathcal{L}[p^\alpha p^\beta] + a_5 \nabla_{\langle \alpha} u_{\beta \rangle} \mathcal{L}[p^\alpha p^\beta] \\
&+ \frac{a_4}{T}\left( \frac{1}{T}\nabla_\beta T - \frac{1}{\mathcal{E}^{(0)} + P^{(0)}} \nabla_\beta P^{(0)} \right) u_\alpha \mathcal{L}[p^\alpha p^\beta],
\end{aligned} \tag{139}
$$

Equating the coefficients of $\nabla_\beta u^\beta$, $\frac{1}{T}\nabla^\beta T - \frac{1}{\mathcal{E}^{(0)} + P^{(0)}} \nabla^\beta P^{(0)}$, and $\nabla_{\langle \alpha} u_{\beta \rangle}$ from both the sides of Equation (140), we have

$$
f^0 \tilde{f}^0 \left[ Q_2(p^\alpha u_\alpha) - Q_1(p^\alpha u_\alpha)^2 - \frac{1}{3T}(m^2 - (u_\alpha p^\alpha)^2) \right] = \frac{a_2}{T^2} u_\alpha u_\beta \mathcal{L}[p^\alpha p^\beta], \tag{140}
$$

$$
\frac{f^0 \tilde{f}^0}{T}\left( (u_\alpha p^\alpha) p_\beta - \frac{\mathcal{E}^{(0)} + P^{(0)}}{n^{(0)}} p_\beta \right) = \frac{a_4}{T} u_\alpha \mathcal{L}[p^\alpha p_\beta], \tag{141}
$$

$$
-f^0 \tilde{f}^0 \frac{p^\alpha p^\beta}{T} = a_5 \mathcal{L}[p^\alpha p^\beta]. \tag{142}
$$

Now, we will evaluate $a_2$, $a_4$, and $a_5$ from Equations (140), (141), and (142), respectively. Multiplying Equation (140) by $p^\gamma u_\gamma \, p^\delta u_\delta$ and integrating with respect to the measure $\frac{d^3 p}{(2\pi)^3 \, p^0}$, we have:

$$
\begin{aligned}
&\int Q_2(p^\alpha u_\alpha)^3 f^0 \tilde{f}^0 \frac{d^3 p}{(2\pi)^3 \, p^0} - \int Q_1(p^\alpha u_\alpha)^4 f^0 \tilde{f}^0 \frac{d^3 p}{(2\pi)^3 \, p^0} \\
&+ \frac{1}{3T}\int (u^\delta p_\delta)^2 (-\Delta^{\alpha\beta} p_\alpha p_\beta) f^0 \tilde{f}^0 \frac{d^3 p}{(2\pi)^3 \, p^0} \\
&= \frac{a_2}{T^2}\int p^\gamma u_\gamma \, p^\delta u_\delta \, u_\alpha u_\beta \mathcal{L}[p^\alpha p^\beta] \frac{d^3 p}{(2\pi)^3 \, p^0},
\end{aligned} \tag{143}
$$

$$
\implies \quad Q_2 J_{30} - Q_1 J_{40} + \frac{1}{T}J_{41} = \frac{a_2}{T^2} u_\alpha u_\beta u_\gamma u_\delta \int p^\gamma p^\delta \mathcal{L}[p^\alpha p^\beta] \frac{d^3 p}{(2\pi)^3 \, p^0},
$$

$$
\implies \quad a_2 = \frac{T^2}{I_1}\left( Q_2 J_{30} - Q_1 J_{40} + \frac{1}{T}J_{41} \right),
$$

where we defined $I_1 \equiv u_\alpha u_\beta u_\gamma u_\delta \int p^\gamma p^\delta \mathcal{L}[p^\alpha p^\beta] \frac{d^3 p}{(2\pi)^3 \, p^0}$ . Multiplying Equation (141) by $\Delta^{\alpha\beta} p_\alpha (u^\delta p_\delta)$ and integrating with respect to the measure $\frac{d^3 p}{(2\pi)^3 \, p^0}$, we have:

$$\int f^0 \tilde{f}^0 \, (u_\gamma p^\gamma) \, p_\beta \, \Delta^{\alpha\beta} p_\alpha \, (u^\delta p_\delta) \, \frac{d^3 p}{(2\pi)^3 \, p^0}$$

$$-\frac{\mathcal{E}^{(0)} + P^{(0)}}{n^{(0)}} \int f^0 \tilde{f}^0 \, p_\beta \, \Delta^{\alpha\beta} p_\alpha \, (u^\delta p_\delta) \, \frac{d^3 p}{(2\pi)^3 \, p^0}$$

$$= a_4 \int u_\gamma u^\delta \, (\Delta^{\alpha\beta} p_\alpha p_\delta) \, \mathcal{L}[p^\gamma p_\beta] \, \frac{d^3 p}{(2\pi)^3 \, p^0} \, ,$$

$$\implies \quad -\int f^0 \tilde{f}^0 \, (u^\delta p_\delta)^2 (-\Delta_{\alpha\beta} p^\alpha p^\beta) \, \frac{d^3 p}{(2\pi)^3 \, p^0}$$

$$+\frac{\mathcal{E}^{(0)} + P^{(0)}}{n^{(0)}} \int f^0 \tilde{f}^0 \, (u^\delta p_\delta) \, (-\Delta_{\alpha\beta} p^\alpha p^\beta) \, \frac{d^3 p}{(2\pi)^3 \, p^0}$$

$$= a_4 \, u_\gamma u_\delta \int (p^\beta p^\delta - u^\alpha u^\beta \, p_\alpha p^\delta) \mathcal{L}[p^\gamma p_\beta] \, , \tag{144}$$

$$\implies \quad -3 J_{41} + 3 \frac{\mathcal{E}^{(0)} + P^{(0)}}{n^{(0)}} J_{31} = a_4 \left[ u_\gamma u_\delta \int p^\beta p^\delta \mathcal{L}[p^\gamma p_\beta] \, \frac{d^3 p}{(2\pi)^3 \, p^0} \right.$$

$$\left. -u_\alpha u_\beta u_\gamma u_\delta \int p^\alpha p^\delta \mathcal{L}[p^\gamma p^\beta] \, \frac{d^3 p}{(2\pi)^3 \, p^0} \right] \, ,$$

$$\implies \quad -3 J_{41} + 3 \frac{\mathcal{E}^{(0)} + P^{(0)}}{n^{(0)}} J_{31} = a_4 [I_2 - I_1] \, ,$$

$$\implies \quad a_4 = \frac{1}{I_2 - I_1} \left[ -3 J_{41} + 3 \frac{\mathcal{E}^{(0)} + P^{(0)}}{n^{(0)}} J_{31} \right] \, ,$$

where we defined $I_2 \equiv u_\gamma u_\delta \int p^\beta p^\delta \mathcal{L}[p^\gamma p_\beta] \frac{d^3 p}{(2\pi)^3 \, p^0}$ . Multiplying Equation (142) by $\Delta^{\gamma\delta\alpha\beta} p_\gamma p_\delta$ and integrating with respect to the measure $\frac{d^3 p}{(2\pi)^3 \, p^0}$ we have:

$$-\frac{1}{T} \int f^0 \tilde{f}^0 (\Delta^{\gamma\delta\alpha\beta} p_\gamma p_\delta p_\alpha p_\beta) \, \frac{d^3 p}{(2\pi)^3 \, p^0} = a_5 \int \Delta^{\gamma\delta\alpha\beta} p_\gamma p_\delta \mathcal{L}[p_\alpha p_\beta] \, \frac{d^3 p}{(2\pi)^3 \, p^0} \, , \tag{145}$$

by using the contractions $\Delta^{\gamma\delta\alpha\beta} p_\gamma p_\delta = p^\alpha p^\beta - (u_\delta p^\delta) p^\alpha u^\beta - (u_\delta p^\delta) p^\beta u^\alpha + (u^\gamma p_\gamma)^2 u^\alpha u^\beta - \frac{1}{3} \Delta^{\alpha\beta} (m^2 - (u_\gamma p^\gamma)^2)$ and $\Delta^{\gamma\delta\alpha\beta} p_\gamma p_\delta p_\alpha p_\beta = m^4 - 2m^2 (u_\delta p^\delta)^2 + (u_\gamma p^\gamma)^4 - \frac{1}{3} (m^2 - (u^\gamma p_\gamma)^2)^2$ , we have:

$$-\frac{1}{T} (m^4 J_{00} - 2m^2 J_{20} + J_{40} - 5 J_{42}) = a_5 \left[ \int p^\alpha p^\beta \mathcal{L}[p_\alpha p_\beta] \, \frac{d^3 p}{(2\pi)^3 \, p^0} \right.$$

$$-u_\delta u^\beta \int p^\delta p^\alpha \mathcal{L}[p_\alpha p_\beta] \, \frac{d^3 p}{(2\pi)^3 \, p^0} - u_\delta u^\alpha \int p^\delta p^\beta \mathcal{L}[p_\alpha p_\beta] \, \frac{d^3 p}{(2\pi)^3 \, p^0}$$

$$+u^\gamma u^\delta u^\alpha u^\beta \int p_\gamma p_\delta \, \mathcal{L}[p_\alpha p_\beta] \, \frac{d^3 p}{(2\pi)^3 \, p^0} - \frac{1}{3} \int \eta^{\alpha\beta} (m^2 - (u^\gamma p_\gamma)^2)^2 \mathcal{L}[p_\alpha p_\beta] \, \frac{d^3 p}{(2\pi)^3 \, p^0}$$

$$+\frac{1}{3} u_\alpha u_\beta \int (m^2 - (u^\gamma p_\gamma)^2)^2 \mathcal{L}[p_\alpha p_\beta] \, \frac{d^3 p}{(2\pi)^3 \, p^0} \right] \, , \tag{146}$$

$$\implies \quad -\frac{1}{T} (m^4 J_{00} - 2m^2 J_{20} + J_{40} - 5 J_{42}) = a_5 \left[ I_3 - 2 I_2 + I_1 - \frac{1}{3} I_1 \right] \, ,$$

$$\implies \quad a_5 = -\frac{m^4 J_{00} - 2m^2 J_{20} + J_{40} - 5 J_{42}}{T \left( I_3 - 2 I_2 + \frac{2}{3} I_1 \right)} \, ,$$

where we have used $\eta^{\alpha\beta}\mathcal{L}[p_\alpha p_\beta] = 0$ and $\int \mathcal{L}[p_\alpha p_\beta] \frac{d^3p}{(2\pi)^3 \, p^0} = 0$ and defined $I_3 \equiv \int p^\alpha p^\beta \mathcal{L}[p_\alpha p_\beta] \frac{d^3p}{(2\pi)^3 \, p^0}$ . Now, since we know all the unknown coefficients from $a_0$ to $a_5$ are known, we can write $T^{(D)\mu\nu}$ from Equation (134) as follows:

$$T^{(D)\mu\nu} = \left[ \frac{R_1}{I_1}\left( Q_2 J_{30} - Q_1 J_{40} + \frac{1}{T}J_{41} \right)(\nabla_\beta u^\beta) \right]\Delta^{\mu\nu} - 2\frac{m^4 J_{00} - 2m^2 J_{20} + J_{40} - 5J_{42}}{T(I_3 - 2I_2 + \frac{2}{3}I_1)}J_{42}\nabla^{\langle\mu}u^{\nu\rangle} . \tag{147}$$

Using the definitions of bulk stress and shear stress tensor given in Equations (94) and (96), we have :

$$\Pi = -\frac{R_1}{I_1}\left( Q_2 J_{30} - Q_1 J_{40} + \frac{1}{T}J_{41} \right)\nabla_\beta u^\beta , \tag{148}$$

$$\text{and} \qquad \pi^{\mu\nu} = -2\frac{m^4 J_{00} - 2m^2 J_{20} + J_{40} - 5J_{42}}{T(I_3 - 2I_2 + \frac{2}{3}I_1)}J_{42}\,\nabla^{\langle\mu}u^{\nu\rangle} . \tag{149}$$

Comparing this with the usual definition of bulk viscosity $\Pi = -\zeta\,\nabla_\beta u^\beta$, and shear viscosity $\pi^{\mu\nu} = 2\eta\,\nabla^{\langle\mu}u^{\nu\rangle}$, we obtain,

$$\zeta = \frac{R_1}{I_1}\left( Q_2 J_{30} - Q_1 J_{40} + \frac{1}{T}J_{41} \right) , \tag{150}$$

$$\text{and} \qquad \eta = -2\frac{m^4 J_{00} - 2m^2 J_{20} + J_{40} - 5J_{42}}{T(I_3 - 2I_2 + \frac{2}{3}I_1)}J_{42} . \tag{151}$$

Similarly, we can obtain the diffusion flow as:

$$v^\mu = \Delta^\mu_\lambda \int \phi f^0 \tilde{f}^0 p^\lambda \frac{d^3p}{(2\pi)^3 \, p^0} ,$$

$$\implies \quad v^\mu = \frac{a_4}{T}\Delta^\mu_\lambda \left( \frac{1}{T}\nabla_\beta T - \frac{1}{\mathcal{E}^{(0)} + P^{(0)}}\nabla_\beta P^{(0)} \right)\int \left[ (u_\alpha p^\alpha)p^\lambda p^\beta - \frac{J_{41}}{J_{31}}p^\lambda p^\beta \right]\frac{d^3p}{(2\pi)^3 \, p^0} , \tag{152}$$

$$\implies \quad v^\mu = \frac{a_4}{T}\frac{J_{41}J_{21} - J_{31}^2}{J_{31}}\left( \frac{1}{T}\nabla^\mu T - \frac{1}{\mathcal{E}^{(0)} + P^{(0)}}\nabla^\mu P^{(0)} \right)$$

$$\implies \quad v^\mu = \frac{a_4(J_{21}J_{31}^2 - J_{41}J_{21}^2)}{J_{31}^2}\nabla^\mu\left( \frac{\mu}{T} \right) ,$$

where to obtain the last step, we used the results: $\frac{1}{T}\nabla^\mu T - \frac{1}{\mathcal{E}^{(0)}+P^{(0)}}\nabla^\mu P^{(0)} = -\frac{TJ_{21}}{J_{31}}\nabla^\mu\left( \frac{\mu}{T} \right)$ . The heat current $q^\mu$ in LF is given by :

$$q^\mu = -\frac{\mathcal{E}^{(0)} + P^{(0)}}{n^{(0)}}v^\mu \quad = -\frac{a_4\,(J_{31}^2 - J_{41}J_{21})}{J_{31}}\nabla^\mu\left( \frac{\mu}{T} \right)$$

$$= \frac{a_4(J_{31}^2 - J_{41}J_{21})}{J_{21}T}\left( \frac{1}{T}\nabla^\mu T - \frac{1}{\mathcal{E}^{(0)} + P^{(0)}}\nabla^\mu P^{(0)} \right) .$$

So, the particle diffusion coefficient $\kappa$ and the heat diffusion coefficient $\lambda$, which are defined by the relations $v^\mu = \kappa\nabla^\mu\left( \frac{\mu}{T} \right)$ and $q^\mu = -\lambda\nabla^\mu\left( \frac{\mu}{T} \right)$, are given by:

$$\kappa = \frac{a_4\,(J_{21}J_{31}^2 - J_{41}J_{21}^2)}{J_{31}^2} , \tag{153}$$

$$\lambda = \frac{\mathcal{E}^{(0)} + P^{(0)}}{n^{(0)}}\kappa = \frac{a_4\,(J_{31}^2 - J_{41}J_{21})}{J_{31}} \tag{154}$$

### 5. Transport Coefficients in different Models of Relaxation Time Approximations

There are several models of collision kernel that can make the calculation of transport coefficients simpler. One of the sectors of these models is known by relaxation time approximation, where one replaces the usual $2 \leftrightarrow 2$ collision kernel of the Boltzmann Equation by a term proportional to $-\frac{f - f^0}{\tau_c}$, where $f^0$ and $\tau_c$ are the local equilibrium distribution and average time of collision between particles of the system. The most widely used models of these types are 1. The Anderson–Witting Model, 2. Marle's Model, and 3. The Bhatnagar-Gross-Krook (BGK) Model. We will discuss the validity of the conservation equations and the form of transport coefficients for each model one by one.

#### 5.1. Anderson–Witting Model

The BTE in this approximation is given by,

$$p^\mu \partial_\mu f = -\frac{(u_\alpha p^\alpha)}{\tau_c} (f - f^0) , \tag{155}$$

therefore, we have $C[f] = -\frac{(u_\alpha p^\alpha)}{\tau_c} (f - f^0)$. If we demand the conservation laws to remain valid, we will have the following constraints:

$$\int -\frac{(u_\alpha p^\alpha)}{\tau_c} (f - f^0) \frac{d^3 p}{(2\pi)^3 \, p^0} = 0 , \tag{156}$$

$$\int -\frac{(u_\alpha p^\alpha)}{\tau_c} p^\mu (f - f^0) \frac{d^3 p}{(2\pi)^3 \, p^0} = 0 . \tag{157}$$

We should notice that the conservation laws are not satisfied by themselves, as we have seen for the $2 \leftrightarrow 2$ collision kernel. Since in CE expansion, one solves $f$ perturbatively, the above constraints can only be satisfied perturbatively. The operator $\mathcal{L}$ in the Anderson–Witting collison model is: $\mathcal{L}[\phi] = -\frac{(u_\alpha p^\alpha)}{\tau_c} f^0 \tilde{f}^0 \phi$ . In order to satisfy the conservation laws, one puts the following constraints on $\phi$:

$$\int (p^\alpha u_\alpha) f^0 \tilde{f}^0 \phi \frac{d^3 p}{(2\pi)^3 \, p^0} = 0 , \tag{158}$$

$$\int p^\mu (p^\alpha u_\alpha) f^0 \tilde{f}^0 \phi \frac{d^3 p}{(2\pi)^3 \, p^0} = 0 , \tag{159}$$

where we have assumed momentum-independent $\tau_c$ . Equation (159) can be written as two equations of the following form:

$$\int (p^\mu u_\mu)^2 f^0 \tilde{f}^0 \phi \frac{d^3 p}{(2\pi)^3 \, p^0} = 0 , \tag{160}$$

$$\Delta_{\mu\nu} \int p^\mu (p^\alpha u_\alpha) f^0 \tilde{f}^0 \phi \frac{d^3 p}{(2\pi)^3 \, p^0} = 0 . \tag{161}$$

Equation (158) matches the non-equilibrium number density $n$ with the local equilibrium number density $n^{(0)}$, and Equation (160) matches the non-equilibrium energy density $\mathcal{E}$ with the local equilibrium number density $\mathcal{E}^{(0)}$. Equation (161) is the condition one imposes on LF choice; therefore, we realized that in Anderson–Witting's model, one is forced to choose LF in order to satisfy the energy–momentum conservation law. From Equation (114), we have:

$$\left[Q_2(p^\alpha u_\alpha) - Q_1(p^\alpha u_\alpha)^2 - \frac{1}{3T}\left(m^2 - \left(u_\alpha p^\alpha\right)^2\right)\right]\nabla_\beta u^\beta - \frac{p^\alpha p^\beta}{T}\nabla_{\langle\alpha}u_{\beta\rangle}$$

$$+\frac{1}{T}\left[\left((u_\alpha p^\alpha)p_\beta - \frac{\mathcal{E}^{(0)} + P^{(0)}}{n^{(0)}}p_\beta\right)\left(\frac{1}{T}\nabla^\beta T - \frac{1}{\mathcal{E}^{(0)} + P^{(0)}}\nabla^\beta P^{(0)}\right)\right] = \frac{\mathcal{L}[\phi]}{f^0\tilde{f}^0}$$

$$= -\frac{(u_\alpha p^\alpha)}{\tau_c}\phi\,,$$

$$\implies \quad \phi = -\frac{\tau_c}{(u^\alpha p_\alpha)}\left[\left(Q_2(p^\alpha u_\alpha) - Q_1(p^\alpha u_\alpha)^2 - \frac{1}{3T}\left(m^2 - \left(u_\alpha p^\alpha\right)^2\right)\right)\nabla_\beta u^\beta - \frac{p^\alpha p^\beta}{T}\nabla_{\langle\alpha}u_{\beta\rangle}\right.$$

$$\left.+\frac{1}{T}\left((u_\alpha p^\alpha)p_\beta - \frac{\mathcal{E}^{(0)} + P^{(0)}}{n^{(0)}}p_\beta\right)\left(\frac{1}{T}\nabla^\beta T - \frac{1}{\mathcal{E}^{(0)} + P^{(0)}}\nabla^\beta P^{(0)}\right)\right] \tag{162}$$

$$\implies \quad \phi = -\tau_c\left[\left(Q_2 - Q_1(p^\alpha u_\alpha) - \frac{1}{3T}(p^\alpha u_\alpha)^{-1}(m^2 - \left(u_\alpha p^\alpha\right)^2)\right)\nabla_\beta u^\beta\right.$$

$$-\frac{p^\alpha p^\beta}{T}(p^\alpha u_\alpha)^{-1}\nabla_{\langle\alpha}u_{\beta\rangle}$$

$$\left.+\frac{1}{T}\left(p_\beta - \frac{\mathcal{E}^{(0)} + P^{(0)}}{n^{(0)}}(p^\alpha u_\alpha)^{-1}p_\beta\right)\left(\frac{1}{T}\nabla^\beta T - \frac{1}{\mathcal{E}^{(0)} + P^{(0)}}\nabla^\beta P^{(0)}\right)\right]$$

One can obtain the particle flow and the dissipative part of the stress–energy tensor from Equation (163) as follows:

$$N^\mu = \int [f^0 + f^0\tilde{f}^0\phi]\,p^\mu\,\frac{d^3p}{(2\pi)^3\,p^0} = n^{(0)}u^\mu + \int f^0\tilde{f}^0\phi\,p^\mu\,\frac{d^3p}{(2\pi)^3\,p^0}\,,$$

$$T^{(D)\mu\nu} = \int f^0\tilde{f}^0\phi\,p^\mu p^\nu\,\frac{d^3p}{(2\pi)^3\,p^0}\,, \tag{163}$$

substituting the $\phi$ from Equation (163), we can write $N^\mu$ and $T^{(D)\mu\nu}$ as:

$$N^\mu = n^{(0)}u^\mu + \frac{\tau_c}{T}\left(\frac{J_{21}^2 - J_{11}J_{31}}{J_{21}}\right)\left(\frac{1}{T}\nabla^\mu T - \frac{1}{\mathcal{E}^{(0)} + P^{(0)}}\nabla^\mu P^{(0)}\right), \tag{164}$$

$$T^{(D)\mu\nu} = -\tau_c\left(Q_1 J_{31} - Q_2 J_{21} + \frac{m^2}{3T}J_{11} - \frac{J_{31}}{3T}\right)\Delta^{\mu\nu}(\nabla_\beta u^\beta) + \frac{2\tau_c}{T}J_{32}\nabla^{\langle\mu}u^{\nu\rangle}\,. \tag{165}$$

Using the decomposition provided in Equation (19) for Equations (165) and (164), the out-of-equilibrium flows for the system can be obtained as:

$$\Pi = \tau_c\left(Q_1 J_{31} - Q_2 J_{21} + \frac{m^2}{3T}J_{11} - \frac{J_{31}}{3T}\right)(\nabla_\beta u^\beta)\,, \tag{166}$$

$$\pi^{\mu\nu} = \frac{2\tau_c}{T}J_{32}\nabla^{\langle\mu}u^{\nu\rangle}\,, \tag{167}$$

$$h^\mu = 0\,, \tag{168}$$

$$\nu^\mu = \frac{\tau_c}{T}\left(\frac{J_{21}^2 - J_{11}J_{31}}{J_{21}}\right)\left(\frac{1}{T}\nabla^\mu T - \frac{1}{\mathcal{E}^{(0)} + P^{(0)}}\nabla^\mu P^{(0)}\right)$$

$$= \tau_c\frac{J_{11}J_{31} - J_{21}^2}{J_{31}}\nabla^\mu\left(\frac{\mu}{T}\right)\,, \tag{169}$$

$$q^\mu = \frac{\tau_c}{T} \frac{J_{11}J_{31}^2 - J_{31}J_{21}^2}{J_{21}^2} \left( \frac{1}{T} \nabla^\mu T - \frac{1}{\mathcal{E}^{(0)} + P^{(0)}} \nabla^\mu P^{(0)} \right)$$

$$= -\tau_c \frac{J_{11}J_{31} - J_{21}^2}{J_{21}} \nabla^\mu \left( \frac{\mu}{T} \right). \tag{170}$$

The transport coefficients in the Anderson–Witting model can be written as:

$$\zeta_{AW} = \tau_c \left( Q_2 J_{21} - Q_1 J_{31} - \frac{m^2}{3T} J_{11} + \frac{J_{31}}{3T} \right), \tag{171}$$

$$\eta_{AW} = \frac{\tau_c}{T} J_{32}, \tag{172}$$

$$\kappa_{AW} = \tau_c \frac{J_{11}J_{31} - J_{21}^2}{J_{31}}, \tag{173}$$

$$\lambda_{AW} = \tau_c \frac{J_{11}J_{31} - J_{21}^2}{J_{21}}. \tag{174}$$

*5.2. Marle's Model*

The BTE in this approximation is given by,

$$p^\mu \partial_\mu f = -\frac{m}{\tau_c}(f - f^0), \tag{175}$$

the collision kernel in this model is $C[f] = -\frac{m}{\tau_c}(f - f^0)$. If we demand that the conservation laws remain valid, we will have the following constraints:

$$\int -\frac{m}{\tau_c}(f - f^0) \frac{d^3p}{(2\pi)^3 \, p^0} = 0, \implies \int (f - f^0) \frac{d^3p}{(2\pi)^3 \, p^0} = 0, \tag{176}$$

$$\int -p^\mu \frac{m}{\tau_c}(f - f^0) \frac{d^3p}{(2\pi)^3 \, p^0} = 0, \implies \int p^\mu (f - f^0) \frac{d^3p}{(2\pi)^3 \, p^0} = 0, \tag{177}$$

where we assumed $\tau_c$ to be momentum-independent. We should notice that the conservation laws are not satisfied by themselves, as we have noticed for the $2 \leftrightarrow 2$ collision kernel in Section 4.1. Since in CE approximation one solves $f$ perturbatively, the above constraints can only be satisfied perturbatively. The operator $\mathcal{L}$ in Marle's collison model is: $\mathcal{L}[\phi] = -\frac{m}{\tau_c} f^0 \tilde{f}^0 \phi$. In order to preserve the conservation laws, one puts the following constraints on $\phi$:

$$\int f^0 \tilde{f}^0 \phi \frac{d^3p}{(2\pi)^3 \, p^0} = 0, \tag{178}$$

$$\int p^\mu f^0 \tilde{f}^0 \phi \frac{d^3p}{(2\pi)^3 \, p^0} = 0. \tag{179}$$

The Equation (179) can be written as two equations of the following form:

$$\int (p^\mu u_\mu) f^0 \tilde{f}^0 \phi \frac{d^3p}{(2\pi)^3 \, p^0} = 0, \tag{180}$$

$$\Delta_{\mu\nu} \int p^\mu f^0 \tilde{f}^0 \phi \frac{d^3p}{(2\pi)^3 \, p^0} = 0, \tag{181}$$

Equation (180) matches the non-equilibrium number density $n$ to local equilibrium number density $n^{(0)}$, and Equation (181) makes the diffusion flow $\nu^\mu = 0$. Equation (181) is the condition one imposes on EF choice; therefore, we realized that in Marle's model, one must choose EF to satisfy the energy–momentum conservation law. From Equation (114), we have:

$$\left[ Q_2(p^\alpha u_\alpha) - Q_1(p^\alpha u_\alpha)^2 - \frac{1}{3T}(m^2 - (u_\alpha p^\alpha)^2) \right] \nabla_\beta u^\beta - \frac{p^\alpha p^\beta}{T} \nabla_{\langle \alpha} u_{\beta \rangle}$$

$$+ \frac{1}{T}\left[ \left( (u_\alpha p^\alpha)p_\beta - \frac{\mathcal{E}^{(0)} + P^{(0)}}{n^{(0)}} p_\beta \right) \left( \frac{1}{T}\nabla^\beta T - \frac{1}{\mathcal{E}^{(0)} + P^{(0)}} \nabla^\beta P^{(0)} \right) \right] = \frac{\mathcal{L}[\phi]}{f^0 \tilde{f}^0} = \frac{-m}{\tau_c}\phi,$$

$$\implies \quad \phi = -\frac{\tau_c}{m}\left[ \left( Q_2(p^\alpha u_\alpha) - Q_1(p^\alpha u_\alpha)^2 - \frac{1}{3T}(m^2 - (u_\alpha p^\alpha)^2) \right) \nabla_\beta u^\beta - \frac{p^\alpha p^\beta}{T} \nabla_{\langle \alpha} u_{\beta \rangle} \right. \tag{182}$$

$$\left. + \frac{1}{T}\left( (u_\alpha p^\alpha)p_\beta - \frac{\mathcal{E}^{(0)} + P^{(0)}}{n^{(0)}} p_\beta \right) \left( \frac{1}{T}\nabla^\beta T - \frac{1}{\mathcal{E}^{(0)} + P^{(0)}} \nabla^\beta P^{(0)} \right) \right].$$

One can obtain the particle flow and the dissipative part of the stress–energy tensor from Equation (183) as follows:

$$N^\mu = \int [f^0 + f^0 \tilde{f}^0 \phi]\, p^\mu \, \frac{d^3 p}{(2\pi)^3\, p^0} = n^{(0)} u^\mu, \tag{183}$$

$$T^{(D)\mu\nu} = \int f^0 \tilde{f}^0 \phi\, p^\mu p^\nu \, \frac{d^3 p}{(2\pi)^3\, p^0},$$

substituting the $\phi$ from Equation (183), we can write $T^{(D)\mu\nu}$ as:

$$\begin{aligned}
T^{(D)\mu\nu} = \frac{\tau_c}{m}\Bigg[ & \left( Q_1 J_{40} + \frac{m^2}{3T}J_{20} - Q_2 J_{30} - \frac{J_{40}}{3T} \right)(\nabla_\beta u^\beta)u^\mu u^\nu \\
& + \left( Q_2 J_{31} - Q_1 J_{41} - \frac{m^2}{3T}J_{21} + \frac{J_{41}}{3T} \right)(\nabla_\beta u^\beta)\Delta^{\mu\nu} + \frac{2}{T}J_{42}\nabla_{\langle \mu} u_{\nu \rangle} \\
& + \frac{1}{T^2}\frac{J_{21}J_{41} - J_{31}^2}{J_{21}}\left( \left( \frac{1}{T}\nabla^\mu T - \frac{1}{\mathcal{E}^{(0)} + P^{(0)}} \nabla^\mu P^{(0)} \right)u^\nu \right. \\
& \left. + \left( \frac{1}{T}\nabla^\nu T - \frac{1}{\mathcal{E}^{(0)} + P^{(0)}} \nabla^\nu P^{(0)} \right)u^\mu \right) \Bigg]
\end{aligned} \tag{184}$$

Using the decomposition provided in Equation (19) for Equations (185) and (183), the dissipative flows for the system can be obtained as:

$$\Pi = -\frac{\tau_c}{m}\left( Q_2 J_{31} - Q_1 J_{41} - \frac{m^2}{3T}J_{21} + \frac{J_{41}}{3T} \right)(\nabla_\beta u^\beta), \tag{185}$$

$$\pi^{\mu\nu} = \frac{2\tau_c}{m\,T}J_{42}\nabla_{\langle \mu} u_{\nu \rangle}, \tag{186}$$

$$h^\mu = q^\mu = \frac{\tau_c}{m\,T^2}\frac{J_{21}J_{41} - J_{31}^2}{J_{21}}\left( \frac{1}{T}\nabla^\mu T - \frac{1}{\mathcal{E}^{(0)} + P^{(0)}} \nabla^\mu P^{(0)} \right)$$

$$= -\frac{\tau_c}{m\,T}\frac{J_{21}J_{41} - J_{31}^2}{J_{31}}\nabla^\mu\left( \frac{\mu}{T} \right), \tag{187}$$

$$\nu^\mu = 0, \tag{188}$$

The transport coefficients in the Marle's model can be written as:

$$\zeta_M = \frac{\tau_c}{m}\left( Q_2 J_{31} - Q_1 J_{41} - \frac{m^2}{3T}J_{21} + \frac{J_{41}}{3T} \right), \tag{189}$$

$$\eta_M = \frac{\tau_c}{m\,T}J_{41}, \tag{190}$$

$$\lambda_M = \frac{\tau_c}{m\,T}\frac{J_{21}J_{41} - J_{31}^2}{J_{31}}. \tag{191}$$

We can observe in Marle's model all the transport coefficients diverge in the massless limit; this makes the model inappropriate for describing particles with low rest masses.

*5.3. BGK Model*

The BTE with the BGK collision kernel can be written as,

$$p^\mu \partial_\mu f = -\frac{(u_\alpha p^\alpha)}{\tau_c} (f - \frac{n}{n^{(0)}} f^0) \,, \tag{192}$$

therefore, we have $C[f] = -\frac{(u_\alpha p^\alpha)}{\tau_c} (f - \frac{n}{n^{(0)}} f^0)$. Here, $n$ is the number density defined by the distribution $f$, i.e., $n = \int (u^\mu p_\mu) f \frac{d^3 p}{(2\pi)^3 p^0}$, and $n^{(0)}$ is the local equilibrium number density defined by $n^{(0)} = \int (u^\mu p_\mu) f^0 \frac{d^3 p}{(2\pi)^3 p^0}$.

If we demand the conservation laws to remain valid, we will have the following constraints:

$$\int -\frac{(u_\alpha p^\alpha)}{\tau_c} (f - \frac{n}{n^{(0)}} f^0) \frac{d^3 p}{(2\pi)^3 p^0} = 0 \,, \tag{193}$$

$$\int -\frac{(u_\alpha p^\alpha)}{\tau_c} p^\mu (f - \frac{n}{n^{(0)}} f^0) \frac{d^3 p}{(2\pi)^3 p^0} = 0 \,. \tag{194}$$

We should notice that the conservation laws are not satisfied by themselves, and one can only satisfy the above constraints perturbatively. The operator $\mathcal{L}$ in BGK's collision model is:

$$\mathcal{L}[\phi] = -\frac{(u^\mu p_\mu)}{\tau_c} f^0 \tilde{f}^0 \phi + \frac{(u^\mu p_\mu)}{\tau_c} \frac{f^0}{n^{(0)}} \int (p^\nu u_\nu) f^0 \tilde{f}^0 \phi \frac{d^3 p}{(2\pi)^3 p^0} \,. \tag{195}$$

To satisfy the conservation laws, one puts the following constraints on $\mathcal{L}[\phi]$:

$$\begin{aligned}
\int \mathcal{L}[\phi] \frac{d^3 p}{(2\pi)^3 p^0} &= -\int \frac{(u^\mu p_\mu)}{\tau_c} f^0 \tilde{f}^0 \phi \frac{d^3 p}{(2\pi)^3 p^0} \\
&\quad + \left( \int (p^\nu u_\nu) f^0 \tilde{f}^0 \phi \frac{d^3 p}{(2\pi)^3 p^0} \right) \int \frac{(u^\mu p_\mu)}{\tau_c} \frac{f^0}{n^{(0)}} \frac{d^3 p}{(2\pi)^3 p^0} \\
&= 0 \,,
\end{aligned} \tag{196}$$

$$\begin{aligned}
\int p^\mu \mathcal{L}[\phi] \frac{d^3 p}{(2\pi)^3 p^0} &= -\int p^\mu \frac{(u^\nu p_\nu)}{\tau_c} f^0 \tilde{f}^0 \phi \frac{d^3 p}{(2\pi)^3 p^0} \\
&\quad + \left( \int (p^\nu u_\nu) f^0 \tilde{f}^0 \phi \frac{d^3 p}{(2\pi)^3 p^0} \right) \int p^\mu \frac{(u^\lambda p_\lambda)}{\tau_c} \frac{f^0}{n^{(0)}} \frac{d^3 p}{(2\pi)^3 p^0} \\
&= 0 \,.
\end{aligned} \tag{197}$$

We can observe that Equation (197) is automatically satisfied, and Equation (198) can be decomposed into two equations of the following form:

$$u_\mu u_\nu \left[ -T^{(D)\mu\nu} + \frac{\delta n}{n^{(0)}} T^{(0)\mu\nu} \right] = 0$$

$$\implies \quad \delta n \, \mathcal{E}^{(0)} = n^{(0)} \, \delta\mathcal{E} \,, \tag{198}$$

$$\text{and,} \quad -\Delta_\nu^\lambda u_\mu T^{(D)\mu\nu} + \frac{\delta n}{n^{(0)}} \Delta_\nu^\lambda u_\mu T^{(0)\mu\nu} = 0$$

$$\implies \quad h^\lambda = 0 \,, \tag{199}$$

where in obtaining Equation (198), we used the definitions:

$$\delta\mathcal{E} = \int f^0 \tilde{f}^0 \, \phi \, (u_\mu p^\mu)^2 \frac{d^3 p}{(2\pi)^3 p^0}, \quad \delta n = \int f^0 \tilde{f}^0 \, \phi \, (u_\mu p^\mu) \frac{d^3 p}{(2\pi)^3 p^0} \,,$$

and in obtaining Equation (199), we used the definition: $h^\lambda = \Delta_\nu^\lambda u_\mu T^{(D)\mu\nu}$. Therefore, in the BGK model, the matching conditions (Equations (198) and (199)) for $\phi$ come from the energy–momentum and particle conservation too. Once again, we see from Equation (199) that, similar to the Anderson–Witting model, we are forced to work on LF choice. One can easily see that Equation (198) can be satisfied if we set both the non-equilibrium energy density and number density to zero, i.e., $\delta\mathcal{E} = \delta n = 0$. Therefore, we will put the following constraints on $\phi$ to solve Equation (114):

$$\int \frac{d^3p}{(2\pi)^3 \, p_0} \, (u_\mu p^\mu)^2 \, f^0 \tilde{f}^0 \phi = \int \frac{d^3p}{(2\pi)^3 \, p_0} \, (u_\mu p^\mu) \, f^0 \tilde{f}^0 \phi = 0 \,, \tag{200}$$

$$\Delta_{\mu\nu} \int p^\mu \, (p^\alpha u_\alpha) \, f^0 \tilde{f}^0 \phi \frac{d^3p}{(2\pi)^3 \, p^0} = 0 \,. \tag{201}$$

Now, we will write Equation (114) with the BGK collision model as:

$$\left[ Q_2(p^\alpha u_\alpha) - Q_1(p^\alpha u_\alpha)^2 - \frac{1}{3T}(m^2 - (u_\alpha p^\alpha)^2) \right] \nabla_\beta u^\beta - \frac{p^\alpha p^\beta}{T} \nabla_{\langle\alpha} u_{\beta\rangle}$$

$$+ \frac{1}{T} \left[ \left( (u_\alpha p^\alpha) p_\beta - \frac{\mathcal{E}^{(0)} + P^{(0)}}{n^{(0)}} p_\beta \right) \left( \frac{1}{T} \nabla^\beta T - \frac{1}{\mathcal{E}^{(0)} + P^{(0)}} \nabla^\beta P^{(0)} \right) \right] = \frac{\mathcal{L}[\phi]}{f^0 \tilde{f}^0}$$

$$= -\frac{(u^\mu p_\mu)}{\tau_c} \phi + (u^\mu p_\mu) \tau_c \frac{1}{n^{(0)} \tilde{f}^0} \int (p^\nu u_\nu) f^0 \tilde{f}^0 \phi \frac{d^3p}{(2\pi)^3 \, p^0} \,, \tag{202}$$

$$\left[ Q_2(p^\alpha u_\alpha) - Q_1(p^\alpha u_\alpha)^2 - \frac{1}{3T}(m^2 - (u_\alpha p^\alpha)^2) \right] \nabla_\beta u^\beta - \frac{p^\alpha p^\beta}{T} \nabla_{\langle\alpha} u_{\beta\rangle}$$

$$+ \frac{1}{T} \left[ \left( (u_\alpha p^\alpha) p_\beta - \frac{\mathcal{E}^{(0)} + P^{(0)}}{n^{(0)}} p_\beta \right) \left( \frac{1}{T} \nabla^\beta T - \frac{1}{\mathcal{E}^{(0)} + P^{(0)}} \nabla^\beta P^{(0)} \right) \right]$$

$$= -\frac{(u^\mu p_\mu)}{\tau_c} f^0 \tilde{f}^0 \phi \,,$$

where the collision operator $\mathcal{L}[\phi]$ reduces to $\mathcal{L}[\phi] = -\frac{(u^\mu p_\mu)}{\tau_c} f^0 \tilde{f}^0 \phi$ because of the matching $\int \frac{d^3p}{(2\pi)^3 \, p_0} \, (u_\mu p^\mu) f^0 \tilde{f}^0 \phi = 0$. The $\phi$ obtained here is exactly equal to the $\phi$ obtained in the Anderson–Witting model; therefore, the transport coefficients will be exactly the same as that of Section 5.1. This is a consequence of the matching $\delta n = \delta\mathcal{E} = 0$, which is the simplest scenario in which the constraint given by Equation (198) is satisfied.

## 6. Numerical Values of Shear Viscosity to Entropy Density Ratio

Looking at the frameworks discussed in the Sections 5.1 and 5.2, we find a common mathematical pattern in the expressions for transport coefficients in Equations (171)–(174) and Equations (189)–(191). Essentially, a transport coefficient is determined by multiplying the thermodynamic phase space with the relaxation time. This relationship can be expressed as:

$$\text{Transport coefficient} = (\text{thermodynamical phase space}) \times (\text{Relaxation Time}), \tag{203}$$

The relaxation time considered here can either be independent of momentum or averaged over momentum.

If we consider a massless case, the integration expressions for transport coefficients take on a straightforward analytic form. The thermodynamic phase-space contribution of $\eta$ for both bosons and fermions can be expressed as $\frac{4\pi^2}{450} T^4$ and $\frac{7\pi^2}{900} T^4$, respectively. The entropy density, a thermodynamic quantity, is given by $s = \frac{4\pi^2}{90} T^3$ for bosons and $\frac{7\pi^2}{180} T^3$ for fermions. Therefore, the dimensionless ratio $\eta/s$ can be determined as $\frac{\tau_c T}{5}$, exhibiting a monotonically increasing trend with temperature $T$ when assuming a temperature-

independent relaxation time. In the context of either bosonic or fermionic systems, distinct microscopic calculations may yield varying $\tau_c(T)$ values, consequently shaping the temperature dependence of $\eta/s$.

Arnold and colleagues [210] extensively summarized calculations for $\eta/s$ employing the perturbative approach in finite temperature QCD, utilizing a re-summed version referred to as the hard thermal loop (HTL). By using the leading-order results from the HTL calculation [210] for quark matter and chiral perturbation theory (ChPT) for hadronic matter [238], Refs. [239,240] interestingly suggested a valley-like profile for $\eta/s(T)$ similar to nitrogen, helium, and water. Nonetheless, the magnitudes they propose ($\frac{10}{4\pi} - \frac{20}{4\pi}$) differ significantly from what was expected in experiments [241], as interpreted through large-scale hydrodynamical simulations [242].

The result of $\eta/s$ obtained by various hydrodynamic groups is visually represented in Figure 4 of Ref. [234]. From this illustration, a preliminary estimate of the order of magnitude, $\eta/s = \frac{1}{4\pi} - \frac{5}{4\pi}$, is anticipated for the matter at LHC or RHIC. So, the ranges $\eta/s = \frac{10}{4\pi} - \frac{20}{4\pi}$ from the standard theories HTL in quark temperature range and ChPT in hadronic temperature are considerably larger than the ranges $\eta/s = \frac{1}{4\pi} - \frac{5}{4\pi}$, measured in RHIC and LHC experiments. This gap suggests estimating the $\eta/s$ from different existing or new model calculations for quark and hadronic matters. Examples of these models in the quark sector are the linear sigma model (LSM) [215], Nambu–Jona–Lasinio (NJL) [211–214], and Polyakov-loop quark meson (PQM) [216], where they can also describe the quasi-particle state of quarks within a hadron at finite temperature. On the other hand, the hadronic models are SMASH [243] codes, URQMD [244], the unitarization methodology [219], hadronic field theory (HFT) [106,220–223], and the hadron resonance gas (HRG) model [224,225], etc., which have provided estimation of $\eta/s$ within the hadronic temperature domain. The calculated values of $\eta/s$ for both hadronic and quark phases, situated both below and above the transition temperature $T_c$, are presented in Table 1, including selective studies whose results closely approach the KSS bound. Among those references, let us briefly address the ideas from Refs. [213,216,220–225], which suggest three reasons why the $\eta/s$ of RHIC or LHC matter is exceptionally low, close to the KSS bound. These reasons are discussed below.

**Table 1.** The value of $\eta/s$ from diverse model estimations, presented in the first column along with corresponding references, is examined across temperature ranges below (in the second column) and above (in the third column) the transition temperature $T_c$.

| Framework [Reference] | | $T \leq T_c$ | $T \geq T_c$ |
|---|---|---|---|
| HTL [210] | Arnold et al. | - | 1.8 |
| LQCD [245] | Meyer | - | 0.1 |
| NJL [211] | Marty et al. | 1–0.3 | 0.3–0.08 |
| NJL [212] | Sasaki et al. | 1–0.5 | 0.5–0.55 |
| NJL [213] | Ghosh et al. | - | 0.5–0.12 |
| NJL [214] | Deb et al. | 2–0.25 | 0.25–0.5 |
| LSM [215] | Chakraborty and Kapusta | 0.87–0.55 | 0.55–0.62 |
| PQM [216] | Singha et al. | 5–0.5 | 0.3–0.08 |
| URQMD [244] | Demir and Bass | 1 | - |
| SMASH [243] | Rose et al. | 1 | - |
| Unitarization [219] | Fernandez-Fraile and Nicola | 0.8–0.3 | - |
| HFT [220–223] | Ghosh et al. | 0.4–0.1 | - |
| HFT [106] | Kalikotay et al. | 0.8–0.25 | - |
| HRG [224,225] | Ghosh et al. | 0.13–0.28 | - |

(1). Resonance type interaction:

In the references listed in Table 1, Refs. [213,216,220–223] explored an effective interaction involving quark–resonance [213,216] and hadron–resonance [220–223] types, which could be a reason for the low value of ratio between viscosity and entropy density ($\eta/s$) in both quark and hadronic matter. In 1994, Quack and Klevansky [246] introduced the idea

of quark propagation with quark–meson loop correction in the NJL model. This idea was subsequently adopted for viscosity calculations by references such as Refs. [213,217,247]. Through calculations involving quark–sigma and quark–pion loops, the relaxation time of quark ($\tau_c$) is determined from the imaginary part of the quark self-energy ($\Pi$), applying the relationship $\tau_c \sim 1/\text{Im}\Pi$. In addition to the NJL model explored in studies like [213,217,247], the PQM model [216], employing similar quark–meson loop calculations, also identified remarkably small values for both quark relaxation time ($\tau_c$) and viscosity-to-entropy ratio ($\eta/s$), approaching the KSS bound. However, these findings are applicable within a limited temperature range near the transition temperature. An alternative approach to compute quark relaxation time, utilizing the same quark–meson Lagrangian density as introduced by Quack and Klevansky [246], has been adopted by studies such as [211,212,214].

Just like the effective quark–resonance interaction, where $\pi$ and $\sigma$ emerge as resonances in quark matter, Refs. [106,220–223] consider effective hadron–resonance interactions. In these studies, resonances such as $\sigma$, $\rho$, $\phi$, $K^*$ mesons, and $N^*$, $\Delta$, and $\Delta^*$ baryons, appear as resonances in the medium of $\pi$, $K$, and $N$. Specifically, Refs. [220–223] calculate the pion relaxation time from $\pi\sigma$ and $\pi\rho$ loops; the kaon relaxation time from $KK^*$ and $K\phi$ loops; and the nucleon relaxation time from $\pi N^*$, $\pi\Delta^*$, and $\pi\Delta$ loops. On the contrary, Ref. [106] estimated these relaxation times using a resonance-scattering type diagram. In contrast to standard ChPT calculations, both hadronic field theory (HFT) calculations [106,220–223] identified significantly small values for $\eta/s$. Therefore, the resonance-type interaction in both hadronic and quark matter may significantly contribute to the observed low $\eta/s$ in LHC or RHIC matter.

(2). Finite-size effect:

Another potential factor contributing to this phenomenon is the finite size effect within the medium [224,225]. The thermodynamic phase space of transport coefficients is reduced due to the quantum effect of the finite size system because momentum integration will start from zero-point momentum instead of zero momentum.

Alternatively, the relaxation time of hadrons may encounter finite-size effects when focusing solely on relaxation scales lower than the system size. Refs. [224,225] have comprehensively demonstrated how the finite size of hadronic matter within the hadron resonance gas (HRG) model can significantly diminish the values of $\eta/s$. The impact of finite size on $\eta/s$ in effective QCD models is also explored in Ref. [248].

(3). Effect of magnetic field:

Another potential explanation for a low $\eta/s$ arises from the influence of a strong magnetic field, potentially generated in non-central heavy-ion collisions. In Refs. [158,160,166,203,249], the shear viscosity of quark matter has been computed in the presence of a magnetic field, showcasing a significant reduction in $\eta/s$. This reduction is attributed to a lower effective relaxation time formed by the combination of synchrotron frequency and particle relaxation time. However, it is imperative to conduct further investigations before confidently asserting that a magnetic field can be considered one of the factors leading to a lower $\eta/s$ in RHIC/LHC matter.

After knowing the three sources for which $\eta/s$ can have a very low value, let us examine its $T$-dependent profiles with a rough numerical band. We will try to understand the earlier microscopic estimations of $\eta/s$ in terms of RTA-based expressions for QGP and HRG phases, where their relaxation times will be tuned to cover the earlier theoretical data. Let us first build our master formulae of $\eta/s$ of QGP and HRG phases from the earlier discussed framework, mainly Section 5.1. The expression for $\eta$ obtained in Section 5.1 can be recast in the following integral form in the LRF of the fluid:

$$
\begin{aligned}
\eta_{AW} &= \frac{\tau_c}{T} J_{32} \\
&= \frac{\tau_c}{T} \frac{1}{5!!} \int \frac{d^3p}{(2\pi)^3 E} \frac{1}{E} (m^2 - E^2)^2 f^0 (1 + a f^0) \\
&= \frac{\tau_c}{15T} \int \frac{d^3p}{(2\pi)^3} \left( \frac{m^2 - E^2}{E} \right)^2 f^0 (1 + a f^0), \\
&= \frac{\tau_c}{15T} \int \frac{d^3p}{(2\pi)^3} \left( \frac{p^2}{E} \right)^2 f^0 (1 + a f^0),
\end{aligned}
\tag{204}
$$

where $f^0 = 1 / \left( e^{\frac{(E-\mu)}{T}} - a \right)$ with $a = 1$ for bosons and $a = -1$ for fermions. Similarly, the thermodynamic variables $n$, $\mathcal{E}$, and $P$ can be expressed as:

$$
n = I_{10} = \int \frac{d^3p}{(2\pi)^3 \, p_0} f^0 \, (u_\alpha p^\alpha) = \int \frac{d^3p}{(2\pi)^3 E} f^0 \, E = \int \frac{d^3p}{(2\pi)^3} f^0,
\tag{205}
$$

$$
\mathcal{E} = I_{20} = \int \frac{d^3p}{(2\pi)^3 \, p_0} f^0 \, (u_\alpha p^\alpha)^2 = \int \frac{d^3p}{(2\pi)^3 E} f^0 \, E^2 = \int \frac{d^3p}{(2\pi)^3} f^0 \, E,
\tag{206}
$$

$$
P = I_{21} = \frac{1}{3} \int \frac{d^3p}{(2\pi)^3 \, p_0} f^0 \, ((u_\alpha p^\alpha)^2 - m^2) = \frac{1}{3} \int \frac{d^3p}{(2\pi)^3} f^0 \, \frac{p^2}{E},
\tag{207}
$$

$$
s = \frac{\mathcal{E} + P - \mu \, n}{T},
\tag{208}
$$

where, for brevity, we neglected the overhead zeros from the local equilibrium thermodynamic variables. In the QGP, the constituents quarks u,d, and s, and the gluons, can be assumed to be ultra-relativistic and obey the dispersion $E = p$. In order to estimate the shear viscosity of the QGP from Equation (205), we have to use the linear dispersion relation along with all the degeneracies associated with the system: spin, flavor, and color. We have the following degeneracies in the quark sector: two spin states, three flavors, and three colors. In the gluon sector, we see the following degeneracies: two spin states and eight independent color states. And since we have a relativistic system, we also have to consider the antiparticles (anti-quarks) along with particles (quarks); this corresponds to one extra multiplication factor of two in the degeneracy factor of the quark sector in the $\mu = 0$ limit. The final expressions of $\eta$ for the QGP phase can be written in general form by substituting $m = 0$, $p = E$, and $\mu = 0$ and introducing spin, flavor, color, and particle–antiparticle degeneracies as follows:

$$
\eta_{QGP} = \tau_c \left[ g_q \int_0^\infty \frac{d^3p}{(2\pi)^3} \left\{ \frac{p^2}{15} \right\} \beta f_q^0 (1 - f_q^0) + g_g \int_0^\infty \frac{d^3p}{(2\pi)^3} \left\{ \frac{p^2}{15} \right\} \beta f_g^0 (1 + f_g^0) \right]
\tag{209}
$$

where $g_q = 2 \times 3 \times 2 \times 3 = 36$ and $g_g = 2 \times 8 = 16$ are quark and gluon degeneracy factors, respectively. Being Fermion, quark will follow the Fermi–Dirac distribution function $f_q = 1 / \{ e^{\beta E} + 1 \}$ (where $E = p$), and being Boson, gluon will follow the Bose–Einstein distribution function $f_g = 1 / \{ e^{\beta E} - 1 \}$ (where $E = p$). The final expressions of the thermodynamic variables for QGP can be written by substituting $m = 0$, $p = E$, and $\mu = 0$ and introducing spin, flavor, color, and particle–antiparticle degeneracies in Equations (205) to (208) as follows:

$$
\begin{aligned}
n_{QGP} &= 12 \left[ \int \frac{d^3p}{(2\pi)^3} f_q^0 - \int \frac{d^3p}{(2\pi)^3} f_{\bar{q}}^0 \right] + g_g \int \frac{d^3p}{(2\pi)^3} f_g^0 \\
&= g_g \int \frac{d^3p}{(2\pi)^3} f_g^0,
\end{aligned}
\tag{210}
$$

$$\mathcal{E}_{QGP} = g_q \int \frac{d^3p}{(2\pi)^3} f_q^0 \, E + g_g \int \frac{d^3p}{(2\pi)^3} f_g^0 \, E \,, \tag{211}$$

$$P_{QGP} = \frac{g_q}{3} \int \frac{d^3p}{(2\pi)^3} f_q^0 \frac{p^2}{E} + \frac{g_g}{3} \int \frac{d^3p}{(2\pi)^3} f_g^0 \frac{p^2}{E} \,, \tag{212}$$

$$s_{QGP} = \frac{\mathcal{E}_{QGP} + P_{QGP} - \mu \, n_{QGP}}{T} = \frac{\mathcal{E}_{QGP} + P_{QGP}}{T} \; (\text{since } \mu = 0) \tag{213}$$

In Figure 1, within the QGP temperature range (roughly $T$= 0.200–0.400 GeV), we have plotted the $\eta/s$ vs. $T$ by using Equation (209) for $\eta$ and Equation (213) for $s$. We have selected a few Refs. [211,216,250], whose $\eta/s$ follow an increasing trend with temperature (within QGP temperature) with the order of magnitude $\frac{1}{4\pi} - \frac{10}{4\pi}$. By taking constant relaxation time and tuning it from $\tau_c = 0.41$ fm (blue dash line) to $\tau_c = 3.94$ fm (red solid line), we can cover the theoretical data of Marty et al. [211], Singha et al. [216], and Plumari et al. [250]. This gives the impression that the relaxation time of QGP lies somewhere between 0.4 to 4.0 fm in the momentum and temperature-independent relaxation time model of the Anderson–Witting type.

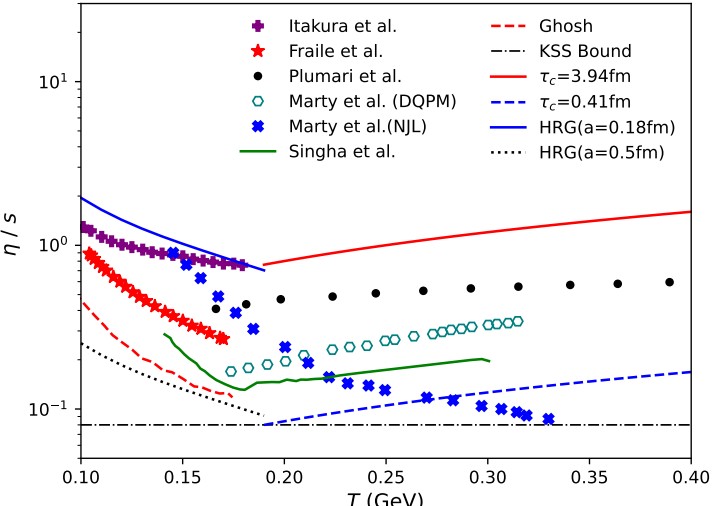

**Figure 1.** (Color online) $\eta/s$ vs. temperature from earlier references. KSS bound $\eta/s = \frac{1}{4\pi}$ (black dash-dotted horizontal line) and RTA curves for massless QGP and HRG within the range of $\eta/s = \frac{1}{4\pi} - \frac{10}{4\pi}$.

Next, we can also write down the final expression of $\eta$ in the hadronic phase by using Equation (205) for multicomponent baryonic and mesonic mixtures in the limit $\mu = 0$. Final expressions of $\eta$ for the hadronic (H) phase with all the baryons (B) and mesons (M) can be written as:

$$\begin{aligned}
\eta_H = & \left[ \sum_B g_B \, \tau_{cB} \int_0^\infty \frac{d^3p}{(2\pi)^3} \left\{ \frac{p^4}{15E^2} \right\} \beta f_B^0 (1 - f_B^0) \right. \\
& \left. + \sum_M g_M \, \tau_{cM} \int_0^\infty \frac{d^3p}{(2\pi)^3} \left\{ \frac{p^4}{15E^2} \right\} \beta f_M^0 (1 + f_M^0) \right], \tag{214}
\end{aligned}$$

where $g_B = 2S_B + 1$ and $g_M = 2S_M + 1$ are spin degeneracy factors for baryons and mesons, respectively. Being Fermion, baryons will follow the Fermi–Dirac distribution function $f_B = 1/\{e^{\beta E} + 1\}$ (where $E = \sqrt{p^2 + m_B^2}$), and being Boson, mesons will follow the Bose–Einstein distribution function $f_M = 1/\{e^{\beta E} - 1\}$ (where $E = \sqrt{p^2 + m_M^2}$). The

final expressions of the thermodynamic variables for the hadronic phase can be written by introducing spin degeneracies in Equations (205)–(208) as follows:

$$n_B = \sum_B g_B \left[ \int \frac{d^3 p}{(2\pi)^3} f_B^0 - \int \frac{d^3 p}{(2\pi)^3} f_{\bar{B}}^0 \right], \tag{215}$$

$$n_H = \sum_B g_B \int \frac{d^3 p}{(2\pi)^3} f_B^0 + \sum_M g_M \int \frac{d^3 p}{(2\pi)^3} f_M^0, \tag{216}$$

$$\mathcal{E}_H = \sum_B g_B \int \frac{d^3 p}{(2\pi)^3} f_B^0 E + \sum_M g_M \int \frac{d^3 p}{(2\pi)^3} f_M^0 E, \tag{217}$$

$$P_H = \sum_B \frac{g_B}{3} \int \frac{d^3 p}{(2\pi)^3} f_B^0 \frac{p^2}{E} + \sum_M \frac{g_M}{3} \int \frac{d^3 p}{(2\pi)^3} f_M^0 \frac{p^2}{E}, \tag{218}$$

$$s_H = \frac{\mathcal{E}_H + P_H - \mu_B n_B}{T} = \frac{\mathcal{E}_H + P_H}{T} \ (\text{since } \mu_B = 0), \tag{219}$$

where $n_B$, $n_H$, $\mathcal{E}_H$, $P_H$, and $s_H$ are, respectively, the net baryon density, total hadron density, total energy density of hadrons, pressure of hadrons, and entropy density of hadrons. The net baryon density vanishes at $\mu_B = 0$ since $f_{\bar{B}}^0 = f_B^0$.

Using Equations (216)–(219), one can obtain HRG thermodynamics, which is in good agreement with lattice QCD thermodynamics within the hadronic temperature range ($T \approx 0.100$–$0.160$ GeV). These ideal HRG thermodynamics (red solid lines) are shown in Figure 2 and compared with LQCD data (blue solid lines) of Ref. ([251]). For HRG, the $\tau_{cB}$ and $\tau_{cM}$ have been calculated by assuming a hard sphere scattering model with $\tau_{cB} = \frac{1}{n_H \sigma v_B}$ and $\tau_{cM} = \frac{1}{n_H \sigma v_M}$, where $\sigma = \pi a^2$ and $a$ is hard sphere scattering length. The average velocity of baryons $v_B$ and mesons $v_M$ are given by,

$$v_B = \frac{\int \frac{d^3 p}{(2\pi)^3} \frac{p}{E} f_B^0}{\int \frac{d^3 p}{(2\pi)^3} f_B^0}, \tag{220}$$

$$v_M = \frac{\int \frac{d^3 p}{(2\pi)^3} \frac{p}{E} f_M^0}{\int \frac{d^3 p}{(2\pi)^3} f_M^0}. \tag{221}$$

The expression of shear viscosity of HRG with the hard sphere scattering model is given by,

$$\eta_H = \frac{1}{n_H \pi a^2} \left[ \sum_B g_B \frac{1}{v_B} \int_0^\infty \frac{d^3 p}{(2\pi)^3} \left\{ \frac{p^4}{15E^2} \right\} \beta f_B^0 (1 - f_B^0) \right.$$

$$\left. + \sum_M g_M \frac{1}{v_M} \int_0^\infty \frac{d^3 p}{(2\pi)^3} \left\{ \frac{p^4}{15E^2} \right\} \beta f_M^0 (1 + f_M^0) \right]. \tag{222}$$

In Figure 1, within the hadronic temperature range (roughly $T = 0.100$–$0.200$ GeV), we have plotted the $\eta/s$ vs. $T$ by using Equation (222) for $\eta$ and Equation (219) for $s$. We have selected few Refs. [216,219–222], whose $\eta/s$ follow a decreasing trend with temperature (within hadronic temperature range) with the order of magnitude $\frac{1}{4\pi} - \frac{10}{4\pi}$. To obtain a decreasing trend of $\eta/s$ with $T$, we need a decreasing $T$-dependent $\tau_c$ instead of constant $\tau_c$. Considering relaxation time as inversely proportional to density, velocity, and hard-sphere scattering cross-section, we can obtain the decreasing $T$-dependent $\tau_c$. By taking constant hard sphere scattering cross section and tuning its scattering length from $a = 0.18$ fm (black dotted line) to $a = 0.5$ fm (blue solid line), we can cover the theoretical data of Fraile et al. [211], Ghosh et al. [220–222], and Singha et al. [216].

Now, based on the predicted range $\eta/s = \frac{1}{4\pi} - \frac{5}{4\pi}$ for RHIC or LHC matter, obtained by various hydrodynamic groups, reported in the review article [234], the microscopic estimations of Refs. [211,216,220–222] are more preferable from experimental data

side. Effective QCD models calculations like the quark–meson (QM) model [216,252] and Nambu–Jona–Lasinio (NJL) model [211,214] can indirectly map the QCD interaction in the non-perturbative domain via a temperature-dependent quark mass. They are also successful in reproducing the gross LQCD thermodynamics in the entire temperature domain. Interestingly, $\eta/s$ from the QM model [216,252] NJL model [214] provide the decreasing and increasing trends in hadronic and quark temperature domains, respectively. The interaction of quark with meson resonances is identified as the reason for this profile and order of magnitude of $\eta/s(T)$. Similarly, in Refs. [220–222], the interaction of pion/kaon/nucleon with other meson/baryon resonances are identified as the reason for the decreasing profile and low order of magnitude of $\eta/s(T)$ within the hadronic temperature range.

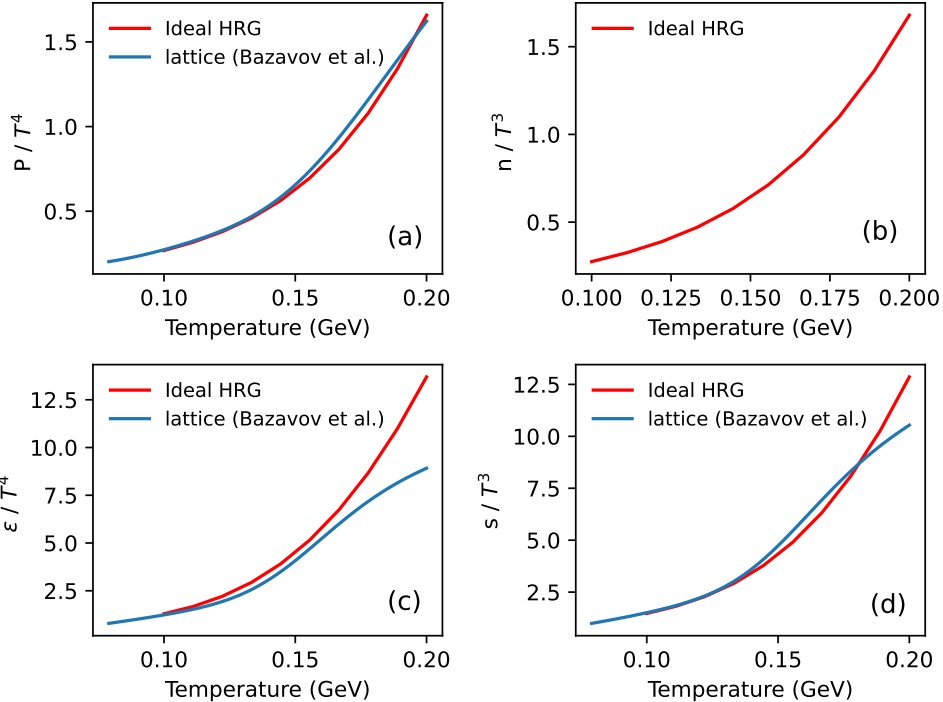

**Figure 2.** (Color online) normalized (**a**) pressure, (**b**) number density, (**c**) energy density, and (**d**) entropy density as a function of temperature in the ideal HRG model. The results are compared with the lattice QCD data from Refs. [251].

In the latest advancements, as outlined in work by Bernhard et al., the understanding of transport coefficients in relativistic heavy ion collisions has taken a significant leap forward [253–255]. Departing from earlier studies that provided approximate constraints, this work by Bernhard et al. employed sophisticated methods, including Bayesian parameter estimation. The study presented the most precise estimates to date for key properties of the QGP and introduced a versatile methodology applicable to various collision models and experimental data of charged particle yields, mean transverse momentum, and anisotropic flow harmonics. Their study suggested a reduced range of $\eta/s \approx \frac{1}{4\pi} - \frac{3}{4\pi}$. If we equivalently map this band via the RTA point of view, then a massless QGP with a very small constant relaxation time range $\tau_c = 0.3$–0.7 fm can be expected, which is certainly a strongly coupled QGP (sQGP) picture.

## 7. Discussions on Bulk Viscosity, Electrical Conductivity, and Thermal Conductivity

In Figure 3, we present plots depicting the temperature dependence of $\zeta/s$ from various models. Specifically, we include results from the linear sigma [215,256], NJL [211], EHRG [227], and Chiral perturbation [257] models, as reported by Chakraborty et al. (or Dobado et al.), Marty et al., Kadam et al., and Frail et al., respectively. These model

calculations collectively exhibit a decreasing trend in $\zeta/s$ with increasing temperature, observed consistently up to $T = 0.17$ GeV. The same trend for the $\zeta/s$ is also observed in Mitra et al. [258], where the authors calculated bulk viscosity of pion gas by including medium effects through $\rho$ and $\sigma$ meson exchange in the $\pi - \pi$ scattering cross-section. On the other hand, Singha et al. [216] predicted the ratio $\zeta/s$ to increase up to 0.17 GeV with the use of the Polyakov-quark–meson model. In contrast to the $\eta/s$ plot in Figure 1, we see that in almost all the models (except ref. [211] ), the $\zeta/s$ peaks around $T = 0.20$ GeV, gradually decreasing and becoming zero at the high-temperature domain, where the conformal limit of QCD is expected to reach. The minimum of shear viscosity to entropy density $\eta/s$ or the maximum of the bulk viscosity to entropy density $\zeta/s$ around $T = 0.20$ GeV obtained from the various model calculations and shown in Figures 1 and 3, respectively, may be taken as a signature of the quark–hadron phase transition. Owing to this fact, these (normalized) transport coefficients may be considered as alternative order parameters of the quark–hadron phase transition. A proper investigation of the temperature profile of these transport coefficients along these lines may shed some light on the order of phase transition and the position of critical temperature.

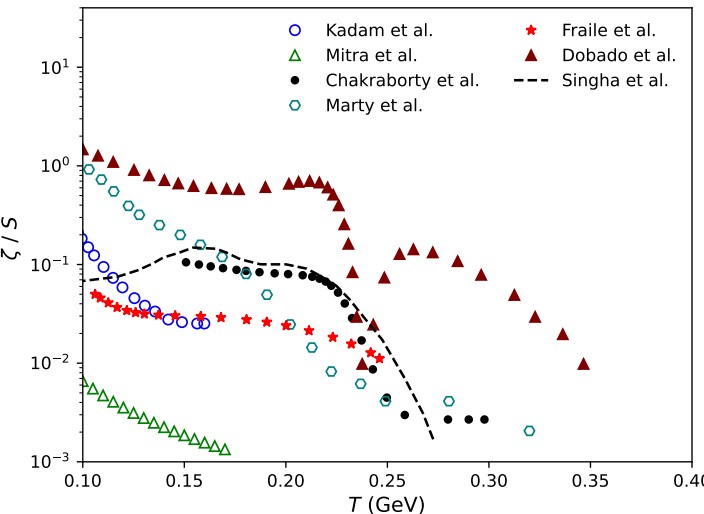

**Figure 3.** (Color online) variation of the bulk viscosity to entropy density ratio ($\zeta/s$) with temperature, comparing different model calculations. Linear sigma, NJL, EHRG, and Chiral perturbation models.

Similarly, in Figure 4 we display the normalized electrical conductivity $\sigma/T$ derived from various model calculations across a range of temperatures. Lee et al. [259] utilized a spectral function approach, while Puglisi et al. [260] employed a Quasi-particle RTA (QP RTA) approach, with both demonstrating an increasing trend in the low-temperature range ($T \sim 0.1$ to 0.2 GeV). Conversely, the results from Cassing et al. [218] using the Parton-hadron-string dynamics (PHSD) model, Marty et al. [211] (using the NJL model), and Frail et al. [219] (using ChPT) exhibit a decreasing trend within this temperature range (up to $T \sim 0.2$ GeV). As temperatures rise, all model calculations demonstrate a gradual increase in $\sigma/T$, reaching a constant value in the conformal regime. Notably, the results from Cassing et al. (PHSD) and Marty et al. (NJL) nearly overlap each other, while the results obtained from Greif et al. [261] using the Boltzmann Approach to Multi-Parton Scatterings (BAMPS) and Puglisi et al. using QP RTA align for most temperature values. Additionally, we have presented the LQCD calculations of electrical conductivity by Amato et al. [262]; in comparison, this data lies below the results of other model calculations. However, the trend observed in the LQCD calculations aligns with that of other model calculations.

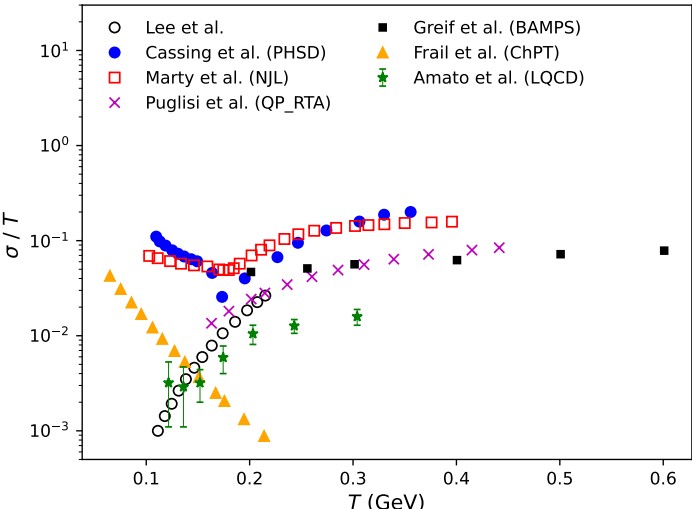

**Figure 4.** (Color online) variation of the normalized electrical conductivity ($\sigma/T$) with temperature, comparing results from different models.

In Figure 5, we illustrate the variation of normalized thermal conductivity $\kappa/T^2$ with respect to temperature. We compare the thermal conductivity obtained from two distinct models: the NJL model [263] and the Quasi-particle model (QP) [263]. The results from the NJL model exhibit a notable pattern: at low temperatures, the thermal conductivity remains relatively constant with temperature, followed by a drop near $T \approx 0.2$ GeV, and then a rapid increase above this temperature threshold. Conversely, the results from the Quasi-particle model show a monotonic increase in thermal conductivity for temperatures $T > 0.2$ GeV.

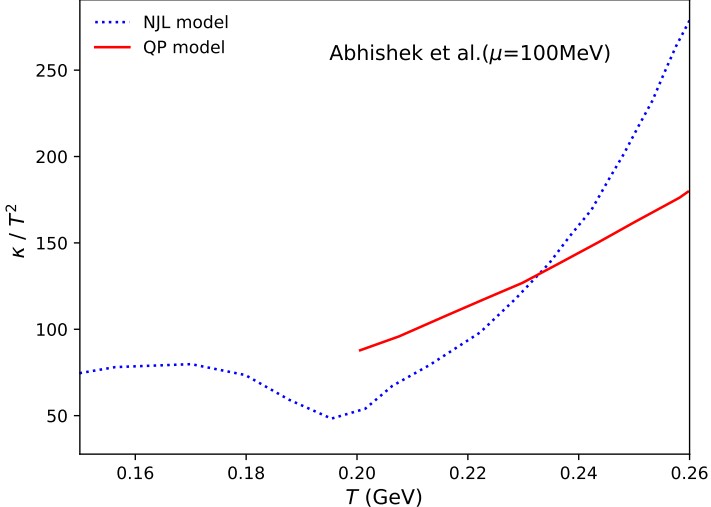

**Figure 5.** (Color online) variation of the normalized thermal conductivity ($\kappa/T^2$) with temperature from two different models: NJL model and Quasi-particle model (QP).

Readers can notice that most of the transport coefficient data, discussed via graphs and tables, are generated by using the basic RTA expression, which roughly carries two components: the thermodynamics part and the relaxation time part. Now, the temperature profile and magnitude of transport coefficients become different because these two components in different model calculations are different. The present review has tried to cover many model calculations cultivated in the last two decades to estimate transport coefficients of quark and hadronic matter. However, it also failed to cover many important model calculations. For example, the Color String Percolation model (CSPM) [264–266] is

one of them. The CSPM is a QCD-inspired model where the interactions between partons in HIC are described by extended colored strings joining the colliding partons. In the CSPM model, one can estimate the thermodynamic and transport properties of the matter formed in HIC. The results obtained from CSPM match with other models and LQCD up to a reasonable extent [264–266].

At the end, we should mention another well-practiced formalism: the Green-Kubo formalism, which is not covered here. Its detailed description can be found in Refs. [217,257,267–269]. Grossly, it is a completely different way of looking at the transport quantities. It was first proposed as a quantum mechanical treatment [267], and later its framework was extended to a quantum field theoretical treatment [217,257,268,269]. If we consider our medium constituents as quantized fields at finite temperature and express different dissipating quantities in terms of those fields, then the dissipation or transport coefficients can be expressed in terms of a two-point function of those fields, which actually interprets the transport probability of fields from one point to another point.

## 8. Summary

We have reviewed the traditional theory of relativistic fluid dynamics, the Boltzmann equation-based kinetic theory, and the calculation of transport coefficients in CEA. Instead of giving the details of the multitude of methods used to derive the relativistic fluid dynamics and transport coefficients from the BTE, we focused on the perturbative technique known as CEA. For the benefit of readers unfamiliar with hydrodynamics, we developed the theory of ideal relativistic fluid dynamics and dissipative fluid dynamics pedagogically. We tried to arrange the section on relativistic fluid dynamics (Section 2) in a way that would also be useful for setting the stage for the readers to read more recent advancements in theory, like first-order casual hydrodynamics. We developed the section on the Boltzmann equation in an instructive way to show how the usual macroscopic conservation laws follow from microscopic conservation laws. In the section on entropy production (Section 3.2), we proved the familiar law of increase in entropy with the help of BTE. We distinguished the global equilibrium distribution from the local equilibrium distribution, which we believe caused confusion many times among the new readers of kinetic theory. We dedicated a section to the local equilibrium thermodynamics (Section 3.3) to prove the thermodynamic identities that are often used in the kinetic theory. In the section on solving BTE with CEA (Section 4), we showed how the dissipative flows come into play when the system goes out of equilibrium. We explicitly demonstrated how one ends up replacing the temporal derivatives from the RHS of BTE with the spatial derivatives for the consistency of the theory. We then used the theory developed in Section 4 to solve for the transport coefficients in the $2 \longleftrightarrow 2$ collision kernel and relaxation type models. We see that contrary to the Marle and Anderson–Witting model, the BGK model allows one to choose one matching condition out of five. Moreover, we discussed the numerical values of shear viscosity to entropy density of the fluid formed in HIC as an application of the methods developed in the article. In Section 6, we gave a brief on the literature where the shear viscosity to entropy density ratio has been calculated for the fluid formed in HIC. We put stress on the importance of different model calculations over the result of perturbative QCD that has taken place in history to calculate the shear viscosity to entropy ratio. Unlike the perturbative QCD calculations, the model calculations predict shear viscosity to entropy density close to the KSS bound, which agrees with the experimental results of HIC. We briefly addressed the potential sources of the low shear viscosity to entropy ratio: the resonance type interaction, the finite size effect, and the effect of the magnetic field. We also made a final estimation of the shear viscosity to entropy ratio by using the formulas derived in this review in CEA for the Anderson–Witting model. We observed that the theoretical curves for $\eta/s$ remain within $\frac{1}{4\pi} - \frac{10}{4\pi}$, which can be realized as Anderson–Witting type RTA-based $\eta/s$ expressions for massless QGP by varying the relaxation time from 0.41 to 3.94 fm. In terms of the recently understood range $\eta/s \approx \frac{1}{4\pi} - \frac{3}{4\pi}$, we can expect a massless QGP with a very small constant relaxation time range $\tau_c$ = 0.3–0.7 fm, which reflects a strongly coupled

nature of RHIC and LHC matter. Furthermore, we reviewed other transport coefficients, including bulk viscosity, electrical conductivity, and normalized thermal conductivity, compiling results from earlier studies.

**Author Contributions:** Conceptualisation, A.D., N.P., A.C. and S.G.; methodology, A.D. and S.G.; validation, A.D., N.P., A.C. and S.G.; formal analysis, A.D. and N.P.; investigation: A.D., N.P., A.C. and S.G.; writing—original draft preparation, A.D. and N.P.; writing—review and editing, A.C. and S.G.; supervision, A.C. and S.G. All authors have read and agreed to the published version of the manuscript.

**Funding:** This research received no external funding.

**Data Availability Statement:** Data are contained within the article.

**Acknowledgments:** A.D. and N.P. gratefully acknowledge the Ministry of Education (MoE), Govt. of India. The authors thank Snigdha Ghosh for providing helpful material and knowledge on HRG model calculations.

**Conflicts of Interest:** The authors declare no conflict of interest.

## Notes

[1]  A Fluid can consist of multiple particle species. Even for a single species at sufficiently high energies, one may have the corresponding antispecies viz, for $e^-$, the pair production can create $e^+$. The stress–energy tensor and the other charge and particle flows can be described accordingly [92,230,231]. We will ignore such scenarios and stick with a single species fluid throughout this section.

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
