# Peer review of "Transport Coefficients of Relativistic Matter: A Detailed Formalism with a Gross Knowledge of Their Magnitude"

_universe, doi:10.3390/universe10030132_

Round 1

Reviewer 1 Report

Comments and Suggestions for Authors

The present paper serves as a review for the transport coeffecients of quark-gluon plasma and hadron-resonance gas formed in relativistic heavy-ion collisions at RHIC and LHC energies. The authors spent the major content on the formalism of relativstic hydrodynamics and Boltzmann kinetic equations, as well as the Chapman-Enskog approximation and the relaxation-time approximation, but less on the results of the obtained transport coefficients so far. The most rigorous Green-Kubo's method has not even been mentioned. It is also desirable to discuss the accuracy/validity of the transport coefficients from different methods. One should also discuss the limitation of the hydrodynamics, since it is not valid for a too large shear viscosity. As one of the most interesting topic on the transport coefficient, the minimum of the ratio of the shear viscosity to the entropy density (and the maximum of the ratio of the bulk viscosity to the entropy density) around the phase transition is not explicitly mentioned or properly discussed. The authors have derived the expressions for different transport coefficients, but only results of the shear viscosity have been discussed in Sec. 6. The authors may also consider to divide some of the long paragraphs into shorter ones, for better structures of the content.

Comments on the Quality of English Language

English is mostly OK. Minor modications on the language and typos are needed.

Author Response

Referee report universe-2852010

Review Report

The present paper serves as a review for the transport coefficients of quark-gluon plasma and hadron-resonance gas formed in relativistic heavy-ion collisions at RHIC and LHC energies. The authors spent the major content on the formalism of relativistic hydrodynamics and Boltzmann kinetic equations, as well as the Chapman-Enskog approximation and the relaxation-time approximation, but less on the results of the obtained transport coefficients so far. The most rigorous Green-Kubo's method has not even been mentioned. It is also desirable to discuss the accuracy/validity of the transport coefficients from different methods. One should also discuss the limitation of the hydrodynamics, since it is not valid for a too large shear viscosity. As one of the most interesting topic on the transport coefficient, the minimum of the ratio of the shear viscosity to the entropy density (and the maximum of the ratio of the bulk viscosity to the entropy density) around the phase transition is not explicitly mentioned or properly discussed. The authors have derived the expressions for different transport coefficients, but only results of the shear viscosity have been discussed in Sec. 6. The authors may also consider to divide some of the long paragraphs into shorter ones, for better structures of the content.

Author’s point-by-point response

  1. The present paper serves as a review for the transport coefficients of quark-gluon plasma and hadron-resonance gas formed in relativistic heavy-ion collisions at RHIC and LHC energies.

Reply: We thank the referee for carefully going through the article and providing valuable comments for the betterment of our work.

—-------------------------------------------------------------------------------------------------------------------------------------

  1. The authors spent the major content on the formalism of relativistic hydrodynamics and Boltzmann kinetic equations, as well as the Chapman-Enskog approximation and the relaxation-time approximation, but less on the results of the obtained transport coefficients so far.

Reply: We agree with this point. Thus, in the last section, named, “Discussions on bulk viscosity, electrical conductivity, and thermal conductivity,” we added a comparison of the numerical results of bulk viscosity to entropy density, electrical conductivity, and thermal conductivity calculated from various models (line no. 1075 to 1118). 

—------------------------------------------------------------------------------------------------------------------------------------

  1. The most rigorous Green-Kubo's method has not even been mentioned. It is also desirable to discuss the accuracy/validity of the transport coefficients from different methods. One should also discuss the limitation of the hydrodynamics, since it is not valid for a too large shear viscosity.

Reply: We agree with the referee that we have not mentioned Green-Kubo’s method of obtaining transport coefficients in this review. The original article was planned for a comprehensive review of the Chapman-Enskog method of obtaining transport coefficients from the Boltzmann transport equation in Relaxation time approximation. Nevertheless, we feel that the readers should be aware of the existence of this rigorous technique. Therefore,  we briefly discussed Green-Kubo’s method of obtaining transport coefficients in lines 1132  to 1140.

—------------------------------------------------------------------------------------------------------------------------------------     4. As one of the most interesting topic on the transport coefficient, the minimum of the ratio of the shear viscosity to the entropy density (and the maximum of the ratio of the bulk viscosity to the entropy density) around the phase transition is not explicitly mentioned or properly discussed.

Reply: We thank the referee for pointing this out.  In the section, named, “Discussions on bulk viscosity, electrical conductivity, and thermal conductivity,” we have added discussions on the maximum of bulk viscosity to entropy ratio and the minimum of the shear viscosity to entropy density around the phase transition in lines 1088 to 1094, “ The minimum of shear viscosity to entropy density /s or the maximum of the bulk viscosity to entropy density /s around T=0.20 GeV obtained from the various quantum field theoretical models and shown in Fig.1 and  Fig.3, respectively, may be taken as a signature of the quark-hadron phase transition. Owing to this fact, these (normalized) transport coefficients may be considered as alternative order parameters of the quark-hadron phase transition. A proper investigation of the temperature profile of these transport coefficients along these lines may shed some light on the order of phase transition and the position of critical temperature. ”

—-----------------------------------------------------------------------------------------------------------------------------------

  1. The authors have derived the expressions for different transport coefficients, but only the results of the shear viscosity have been discussed in Sec. 6. 

Reply: We have added the discussion on the other transport coefficients like bulk viscosity, electrical conductivity, and thermal conductivity in the section, named, “Discussions on bulk viscosity, electrical conductivity, and thermal conductivity”.

—-----------------------------------------------------------------------------------------------------------------------------------

  1. The authors may also consider to divide some of the long paragraphs into shorter ones, for better structures of the content.

Reply: We are thankful to the referee for this suggestion. In the introduction, we split several longer paragraphs into shorter ones for better content structure.

—-----------------------------------------------------------------------------------------------------------------------------------

Summary of changes

  1. Apart from the above-mentioned changes in the manuscript, we also made some grammatical corrections like punctuation errors.
  2. We also paraphrased some sentences for clarity and a better understanding of the readers. The paragraph (lines 231 to 248) in the introduction has been rewritten to reduce the similarity rate following the editorial board's suggestion.
  3. In line number 439, we redefine the heat flow vector correctly with the inclusion of number density in the denominator. The error caused by the wrong definition of heat flow propagated to several places in the article; therefore, we appropriately changed the heat flow-related expression in line no. 455, 479, 810, 812 (Eq. 155), 842 (Eq. 171),  843 (Eq. 175).

The modifications in the updated manuscript are indicated using text highlighted in red. 

Reviewer 2 Report

Comments and Suggestions for Authors

The paper is an interesting review  of transport coefficients extended to macroscopic and microscopic descriptions based in relativistic fluid dynamics and kinetic theory respectively. It is also discussed the relaxion time approximation in models of the Bolzmann transport equations to join the above mentioned descriptions. Finally it is studied the shear and the bulk viscosity over entropy density ratio comparing different approaches in relation with RHIC and LHC data. The paper is well written and will be appreciate by the scientific community due to the overall compact view of the field and for this reason deserves the paper should be published but before it could be improved including the following points.

  Concerning the shear and bulk viscosity in the paper are shown different results of models, however for completness it would be convenient thto include others related to dynamical models, for instance the models studied in references J.Dias de Deus et al Phys Rev C 93 024915 (2016) J.Alvarado Garcia et al Phys Rev D 108 114002(2023), D.Sahu et al arXiv 2006.04185 or the included in X.G.Ding et al arXiv 2401.02293. Also some comment concerning on the relaxion time in line with the study of M.Rybczynski Phys Rev D 103 11 (2021) should be convenient.

  For sake of completness should be very convenient to include comments or studies on other transport coeeficients as thermal conductivity, electrical conductivity,... 

Author Response

Referee report universe-2852010

Review Report

The paper is an interesting review of transport coefficients extended to macroscopic and microscopic descriptions based in relativistic fluid dynamics and kinetic theory respectively. It is also discussed the relaxation time approximation in models of the Bolzmann transport equations to join the above mentioned descriptions. Finally it is studied the shear and the bulk viscosity over entropy density ratio comparing different approaches in relation with RHIC and LHC data. The paper is well written and will be appreciate by the scientific community due to the overall compact view of the field and for this reason deserves the paper should be published but before it could be improved including the following points.

  Concerning the shear and bulk viscosity in the paper are shown different results of models, however for completeness it would be convenient to include others related to dynamical models, for instance the models studied in references J.Dias de Deus et al Phys Rev C 93 024915 (2016) J.Alvarado Garcia et al Phys Rev D 108 114002(2023), D.Sahu et al arXiv 2006.04185 or the included in X.G.Ding et al arXiv 2401.02293. Also some comment concerning on the relaxation time in line with the study of M.Rybczynski Phys Rev D 103 11 (2021) should be convenient.

  For sake of completeness should be very convenient to include comments or studies on other transport coefficients as thermal conductivity, electrical conductivity,... 

Author’s point-by-point response                                                                                                                       1. The paper is an interesting review of transport coefficients extended to macroscopic and microscopic descriptions based in relativistic fluid dynamics and kinetic theory respectively. It is also discussed the relaxation time approximation in models of the Bolzmann transport equations to join the above mentioned descriptions. Finally it is studied the shear and the bulk viscosity over entropy density ratio comparing different approaches in relation with RHIC and LHC data. The paper is well written and will be appreciate by the scientific community due to the overall compact view of the field and for this reason deserves the paper should be published but before it could be improved including the following points.

Reply: We are thankful to the referee for going through the article thoroughly and providing valuable comments to enrich the content of the article.

—---------------------------------------------------------------------------------------------------------------------------                      2. Concerning the shear and bulk viscosity in the paper are shown different results of models, however for completeness it would be convenient to include others related to dynamical models, for instance the models studied in references J.Dias de Deus et al Phys Rev C 93 024915 (2016) J.Alvarado Garcia et al Phys Rev D 108 114002(2023), D.Sahu et al arXiv 2006.04185 or the included in X.G.Ding et al arXiv 2401.02293. Also some comment concerning on the relaxation time in line with the study of M.Rybczynski Phys Rev D 103 11 (2021) should be convenient.

Reply: We thank the referee for introducing us to these important references. The transport coefficients studied under the framework of the Color String Percolation Model (CSPM) in the Refs:  J.Dias de Deus et al. Phys Rev C 93 024915 (2016), J.Alvarado Garcia et al. Phys Rev D 108 114002(2023), D.Sahu et al. arXiv 2006.04185 have been included in the article in lines 1126 to 1131, “For example, the Color String Percolation model (CSPM) [269-271] is one of …..”. Similarly, the review article ‘X.G.Ding et al. arXiv 2401.02293’ on the shear viscosity of the nucleonic matter has been added in the introduction section in line 198, “ A comprehensive knowledge on shear viscosity of nucleonic matter can be found in the Ref. [112]”. We also added comments on the relaxation times of the multiparticle system in the introduction section in lines 199 to 204, “As an aside comment, we should also mention the time evolution of a multiparticle system is associated with….”.

—----------------------------------------------------------------------------------------------------------------------------

  1. For sake of completness should be very convenient to include comments or studies on other transport coeeficients as thermal conductivity, electrical conductivity,... 

Reply: We have included studies on bulk viscosity, electrical conductivity, and thermal conductivity in the section “Discussions on bulk viscosity, electrical conductivity, and thermal conductivity” of the article.

—---------------------------------------------------------------------------------------------------------------------------

Summary of changes                                                                                                                                            1. 1. Apart from the above-mentioned changes in the manuscript, we also made some grammatical corrections like punctuation errors.                                                                                                                                2. We also paraphrased some sentences for clarity and a better understanding of the readers. The paragraph (lines 231 to 248) in the introduction has been rewritten to reduce the similarity rate following the editorial board's suggestion.                                                                                                                                     3. In line number 439, we redefine the heat flow vector correctly with the inclusion of number density in the denominator. The error caused by the wrong definition of heat flow propagated to several places in the article; therefore, we appropriately changed the heat flow-related expression in line no. 455, 479, 810, 812 (Eq. 155), 842 (Eq. 171),  843 (Eq. 175).

The modifications in the updated manuscript are indicated using text highlighted in red. 

Round 2

Reviewer 1 Report

Comments and Suggestions for Authors

The authors have addressed my comments and made proper changes to the manuscript. I recommend it for publication in Universe.

Comments on the Quality of English Language

English language is mostly fine.

Reviewer 2 Report

Comments and Suggestions for Authors

The authors have inclused my suggestions